# Mechano-osmotic signals control chromatin state and fate transitions in pluripotent stem cells

Kaitlin P. McCreery [1,13], Aki Stubb[1,2,13], Rebecca Stephens[3],
Nadezda A. Fursova[4], Andrew Cook[3], Kai Kruse [5], Anja Michelbach[1],
Leah C. Biggs[1,2], Adib Keikhosravi[6], Sonja Nykänen[2,7], Roosa Pulkkanen[2,7],
Christel Hydén-Granskog[8], Jizhong Zou[9], Jan-Wilm Lackmann [10],
Carien M. Niessen [11], Sanna Vuoristo[2,7,12], Yekaterina A. Miroshnikova[3,14] ✉ &
Sara A. Wickström [1,2,12,14] ✉

Acquisition of specific cell shapes and morphologies is a central component of cell fate transitions. Although signalling circuits and gene regulatory networks that regulate pluripotent stem cell differentiation have been intensely studied, how these networks are integrated in space and time with morphological changes and mechanical deformations to control state transitions remains a fundamental open question. Here we focus on two distinct models of pluripotency, preimplantation inner cell mass cells of human embryos and primed pluripotent stem cells, to discover that cell fate transitions associate with rapid, compaction-triggered changes in nuclear shape and volume. These phenotypical changes and the associated active deformation of the nuclear envelope arise from growth factor signalling-controlled changes in cytoskeletal confinement and chromatin mechanics. The resulting osmotic stress state triggers global transcriptional repression, macromolecular crowding and remodelling of nuclear condensates that prime chromatin for a cell fate transition by attenuating repression of differentiation genes. However, while this mechano-osmotic chromatin priming has the potential to accelerate fate transitions and differentiation, sustained biochemical signals are required for robust induction of specific lineages. Our findings uncover a critical mechanochemical feedback mechanism that integrates nuclear mechanics, shape and volume with biochemical signalling and chromatin state to control cell fate transition dynamics.

The deployment of transcriptional programmes during early embryogenesis is tightly coordinated with morphological transformations of cells to control the maintenance of pluripotency and commitment to the first cell lineages[1]. Pluripotency is a transient developmental feature, but can be captured in in vitro cultured human pluripotent stem (hPS) cells, including embryonic stem (hES) and induced pluripotent stem (hiPS) cells. These cells can be maintained in states of naive or primed pluripotency, resembling the late blastocyst inner cell mass

(ICM), or postimplantation epiblast, respectively[2,3]. Pluripotency can be seen as a continuum of state transitions[4], during which hPS cells undergo global changes in transcriptional programmes, triggered by context-dependent growth factor signals[5–9].

During the transition from pluripotency to specific cell lineages, hPS cells also undergo distinct morphological and mechanical changes[10–12]. Studies in other systems show that morphological and volumetric changes can also regulate gene expression through

---

mechanosensitive signalling pathways such as the YAP pathway[13–15]. Besides activating intracellular signalling, extrinsic mechanical forces and osmotic pressures can also directly deform the cell nucleus, driving the adaptation of nuclear morphology and genome architecture[16–20]. Beyond these mechanosensitive effects on transcription, the role of mechanical properties of the nucleus and chromatin in gene regulation is less understood. On the scale of the entire nucleus, chromatin behaves like an elastic solid, but on the scale of chromatin domains it displays liquid-like properties[21–24]. These rheological properties influence the dynamics of molecular processes—for example, by altering chromatin fibre interactions driven by chromatin remodellers, which can in turn regulate genome activity[25,26]. On a larger scale, mechanisms such as remodeller- and ATP-dependent chromatin stirring may facilitate protein access to dense heterochromatic regions, although the physiological relevance remains unclear[27]. Thus, how cell shape transitions, nuclear mechanical and volumetric properties are integrated during lineage transitions to influence chromatin and gene expression presents a fundamental open question.

Here, we use hPS cells to investigate how morphological and osmotic changes are integrated with biochemical signalling and gene regulatory networks to gate cell fate transitions. We find that cell fate transitions in the ICM of human preimplantation embryos and three-dimensional (3D) blastoid embryo models are associated with a decrease in nuclear volume, accompanied by a transient osmotic stress response. Mechanistic studies in primed hiPS cells reveal that removal of growth factors that maintain pluripotency generates mechanical and osmotic stresses that act in concert to modulate nucleoplasmic viscosity and nuclear stiffness. Consequently, macromolecular crowding and biomolecular condensate remodelling prime chromatin for a cell fate transition. We propose that mechano-osmotic reprogramming of the nuclear environment tunes differentiation efficiency by lowering the energy barrier for cell fate transitions.

## Results

### Exit from pluripotency is associated with mechano-osmotic remodelling of the nucleus

The first embryonic cell lineage segregation to either trophoblast or ICM is observed when the embryo proceeds from the morula to early blastocyst stage[28–30]. The ICM then segregates into epiblast and hypoblast where the epiblast expresses pluripotency-associated transcription factors OCT4 and NANOG, whereas the hypoblast expresses SOX17, GATA6, GATA4 and PDGFRA[29,31–34]. To obtain insights into morphological transitions that accompany early human development, we quantified nuclear shapes during ICM lineage transitions. We obtained surplus human embryos (five embryos) donated for research. These embryos were morphologically staged, after which whole-mount immunostainings were performed of NANOG and GATA6, together with DAPI and Lamin-B1 to mark chromatin and the nuclear lamina (Fig. 1a). Using 3D segmentation, we observed that the nuclei of GATA6-high cells had reduced volumes and increased surface-to-volume ratios compared with NANOG-high cells, indicative of flattening or deformation (Fig. 1b). Intriguingly, we noted the presence of actin structures that correlated with nuclear deformation (Fig. 1c).

We next asked if the observed nuclear shapes were relevant to cell fate transitions and which stimuli could be driving these changes. For this, we generated 3D blastoids from naive hES cells[35]. Similarly to human embryos, the emerging GATA6-positive cells within the blastoid ICM displayed reduced nuclear volumes. In addition, they showed activation of the osmosensitive kinase p38 mitogen-activated protein kinase (MAPK)[36,37] as an indicator of osmotic stress (Fig. 1d).

To determine if mechano-osmotic nuclear transitions occur as a cause or consequence of exit from pluripotency, we turned to cellular models of primed pluripotency. Morphometric analyses revealed that differentiation of hiPS cells (Allen Institute) from primed pluripotency into the three germ layers was accompanied by reduction in nuclear volume and flattening (Fig. 1e–g and Extended Data Fig. 1a–d), paralleling observations in mouse ES cells exiting naive pluripotency[38].

To understand the source of mechano-osmotic stress, we plated a mixture of endogenously tagged SOX2 and LMNB1 hiPS cell reporter lines on two-dimensional (2D) micropatterns (also termed 2D gastruloids)[39] to generate sparse mosaicism for tracking and resolving single nuclei or cells for segmentation, and triggered differentiation by removing ROCK inhibitor and adding BMP4 (Fig. 1h,i). Live imaging revealed that 6–8 h after removing ROCK inhibitor/adding BMP4, before lineage selection, the hiPS cell colony displayed collective compaction (Fig. 1i,j and Supplementary Video 1). This was followed by reflattening, and finally the emergence of the expected radial differentiation pattern[39] (Fig. 1h,i). Importantly, strongest compaction was accompanied by transient nuclear deformation (Fig. 1i,j and Supplementary Video 1). Given that mechanosensitive transcription factors aid in embryonic cell fate specification[12,20,38,40,41], and reported nuclear deformation as the critical factor activating YAP[42], we monitored

**Fig. 1 | Exit from primed pluripotency is associated with mechano-osmotic remodelling of the nucleus. a**, Representative images (from five embryos) and quantification of nuclear volume in human preimplantation stage embryos stained for DAPI, Gata6, Nanog and LaminB1. **b**, Nuclear volume of Gata6-high cells in the ICM (scale bars, 50 and 5 μm; $n$ = 20 (Gata high), 16 (Nanog high) nuclei pooled across 5 embryos; mean ± s.d.; Mann–Whitney). **c**, Representative images of human preimplantation stage embryos stained for DAPI, Nanog and F-Actin (phalloidin). Note actin-rich bleb-like structures with corresponding nuclear deformation (scale bars, 20 and 10 μm; images representative of 3 embryos). **d**, Representative images and quantification of nuclear volume and phosphorylated p38 (p-p38) in Gata6-positive and Gata6-negative ICM cells from human blastoids generated from naive hiPS cells stained for Gata6, Oct3/4 and pp38 (scale bar, 100 μm; $n$ = 6 blastoids representing 33 (Gata6 high) and 76 (Gata6 low) nuclei for volumes and 26 (Gata6 high) and 126 (Gata6 low) for p-p38 intensity, respectively; paired $t$-test). **e**, Representative top views ($x$–$y$), 3D reconstructions and cross-sections ($z$), of Sox2-GFP-tagged hiPS cells undergoing ectodermal differentiation for the indicated timepoints (representative of 3 independent experiments; scale bars, 15 μm). **f**, Quantification of nuclear height from hiPS cells undergoing ectodermal differentiation for the indicated timepoints ($n$ = 3 independent experiments with 360 ($t$ = 0), 591 ($t$ = 8), 656 ($t$ = 24), 1,086 ($t$ = 48) total nuclei/timepoint; Kruskal–Wallis/Dunn's). **g**, Quantification of nuclear volume from hiPS cells undergoing trilineage differentiation for the indicated timepoints ($n$ = 3 independent experiments with 903, 946, 1,771 and 1,636 (ectoderm 0, 8, 24 and 48 h, respectively); 334, 437, 816 and 741 (mesoderm 0, 8, 24 and 48 h, respectively); and 393, 416, 709 and 835 (endoderm 0, 8, 24 and 48 h, respectively) total nuclei/condition; Kruskal–Wallis/Dunn's). **h**, Representative immunofluorescence images of Sox2-GFP-tagged hiPS cells on 2D micropatterns treated with BMP4 for the indicated timepoints and stained for Brachyury and Nanog. Note radial pattern of differentiation at 48 h (scale bars, 100 μm; $n$ = 3 independent experiments). **i,j**, Representative snapshots (**i**) and quantification of nuclear deformation and Sox2 intensity dynamics (**j**) from live imaging videos of mosaic micropatterns with Sox2-GFP and LaminB1-RFP hiPS cells treated with BMP4 for the indicated timepoints. Note transient compaction of colony, accompanied by nuclear deformation before appearance of radial differentiation pattern (scale bars, 100 μm; $n$ = 9 gastruloids pooled across 4 independent experiments; mean ± s.d.). **k,l**, Representative immunofluorescence images (**k**) and quantification (**l**) of LaminB1-RFP-tagged hiPS cells on 2D micropatterns treated with BMP4 for the indicated timepoints of maximal colony compaction and stained for p-p38 and YAP. Note transient YAP and p38 activation at pattern centres as well as sustained YAP and p38 activation within edge cells (scale bars, 100 μm; $n$ = 9 (9 h)/10 (rest) 2D gastruloids pooled across 3 independent experiments with 1,278, 1,918 and 3,203 (6 h centre, mid and edge, respectively); 1,249, 1,881 and 3,135 (7 h centre, mid and edge, respectively); 1,388, 2,155 and 3,590 (8 h centre, mid and edge, respectively); 1,363, 2,054 and 3,421 (9 h centre, mid and edge, respectively); and 1,260, 1,898 and 3,750 (10 h centre, mid and edge, respectively) total nuclei/condition; minimum-to-maximum box plots show 75th, 50th and 25th percentiles; ANOVA/Dunnett's). a.u., arbitrary units; diff, differentiation; nuc, nucleus; cyto, cytoplasm.

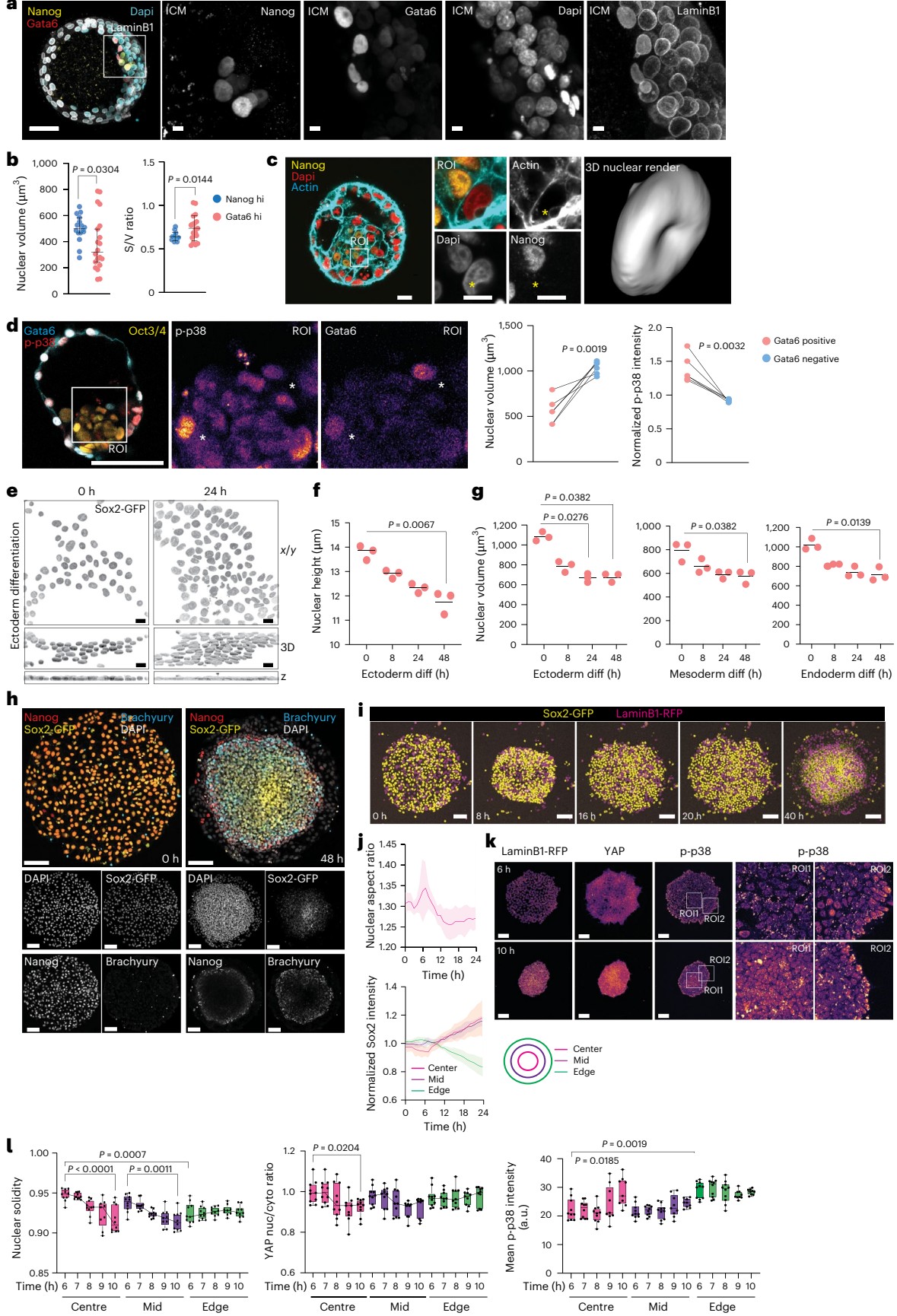

nuclear YAP levels during the compaction of the colony, and observed high YAP activity in early phases of contraction, followed by inactivation (Fig. 1k,l). Compaction was further associated with activation of the osmosensitive kinase p38 MAPK at the centre of the patterns, where changes in nuclear shapes were most substantial (Fig. 1k,l). The edge cells showed sustained deformation nuclei and sustained patterns of YAP and p38 activation (Fig. 1k,l). To confirm that colony compaction caused these responses, we enhanced contractility using Calyculin A[43] and observed increased nuclear deformation, volume loss and p38 activation (Extended Data Fig. 1e,f).

Collectively, these experiments demonstrated that pluripotency exit and cellular compaction is characterized by transient nuclear deformation, volume loss and a mechano-osmotic stress response.

## Mechano-osmotic nuclear remodelling is a rapid response to growth factor removal

To understand the temporal dynamics and molecular mechanisms of mechano-osmotic nuclear changes preceding pluripotency exit, we analysed the immediate response of nuclear volume to removal of FGF2 and TGF-β1 that maintain primed pluripotency in hiPS cells[44]. Replacing the pluripotency-maintenance medium with basal medium lacking these growth factors resulted in rapid reduction of nuclear volume within 15 min of media change, despite unchanged media osmolarities (Fig. 2a and Extended Data Fig. 2a). This effect was mainly driven by FGF2 removal, as adding FGF2 to the basal medium prevented nuclear volume loss, while adding TGF-β1 had a less prominent impact (Fig. 2a). Consistent with observations in the differentiation experiments (Fig. 1), removal of pluripotency-maintaining growth factors triggered a subtle but robust activation of p38, and this activation was counteracted by adding FGF2 into the medium (Extended Data Fig. 2b).

To investigate the mechanisms of this rapid, minute-scale nuclear volume change triggered by growth factor removal, we analysed nuclear envelope mechanics by quantifying nuclear envelope fluctuations using fast imaging of LaminB1-RFP hiPS cells[45,46]. Nuclear envelope fluctuations increased immediately following the removal of pluripotency-maintaining growth factors, appearing within minutes of the medium exchange (Fig. 2b and Supplementary Video 2). Similarly to the decrease in nuclear volume, nuclear fluctuations were controlled by growth factors, specifically FGF2 (Fig. 2b and Supplementary Video 2).

Given the rapid timescale and known interactions between growth factor signalling and cytoskeletal remodelling[47], we hypothesized that nuclear envelope fluctuations resulted from the perinuclear cytoskeleton actively deforming the nucleus. Indeed, live imaging revealed two key actin structures: a taut perinuclear actin ring that encapsulates the nuclear envelope, and dynamic intercellular cavities resembling 'microlumens' that actively deform the nucleus (Fig. 2c and Supplementary Video 3). Occasionally, membrane blebs, frequently associated with mitotic cells, also correlated with nuclear deformations (Fig. 2c and Supplementary Video 3).

Consistent with active forces driving nuclear fluctuations, enhancing contractility using Calyculin A resulted in amplified nuclear envelope fluctuations (Fig. 2d and Supplementary Video 4). However, while complete actin cytoskeleton depolymerization with Cytochalasin D reduced fluctuations, combined disruption of microtubules and F-actin unexpectedly accelerated them (Fig. 2d and Supplementary Video 4). This indicated that the cytoskeleton plays dual roles—both actively deforming and confining the nucleus—and suggested the presence of cytoskeleton-independent processes. We investigated whether internal forces from chromatin contribute to these fluctuations, as previously suggested[45]. We depleted cellular ATP to block energy-dependent processes including chromatin remodelling and found substantially reduced nuclear envelope dynamics (Fig. 2d and Supplementary Video 4). Collectively, these findings supported a model where fluctuations reflect the dynamic force balance between cytoskeletal and intranuclear processes.

To examine the relationship between cytoskeletal confinement and intranuclear forces, we used a compression bioreactor to confine hiPS cell colonies to a height of 5 μm, resulting in nuclear flattening with a 30–40% decrease in nuclear height (Fig. 2e). We found that, while mechanical compression altered nuclear fluctuations, their dynamics were strongly dependent on the growth factor signalling. In pluripotency-promoting media, compression slowly attenuated nuclear envelope fluctuations and increased tautness. By contrast, in basal media lacking pluripotency growth factors, we observed a biphasic response: an initial rapid decrease in fluctuations with increased nuclear wrinkling, followed by amplified nuclear envelope fluctuations over longer timescales (Fig. 2e,f). Intriguingly, growth factors also controlled the rates of nuclear volume loss triggered by extrinsic compression. In basal media, hiPS cell nuclei exhibited rapid volume loss followed by a quick recovery. By contrast, nuclei in pluripotency media showed a gradual volume decrease that peaked after 30 min of confinement (Fig. 2g). These distinct volume change kinetics closely paralleled patterns in nuclear fluctuations. This suggested that osmotic forces could regulate nuclear fluctuations, where volume loss would attenuate fluctuations and volume recovery accelerate them. To directly test this, we induced hypertonic stress, which triggered rapid nuclear volume loss and wrinkling (Extended Data Fig. 2c). As predicted, hypertonic treatment temporarily ceased nuclear fluctuations regardless of the biochemical environment. Upon return to isotonic conditions, the fluctuations resumed, with a more substantial recovery observed in basal media (Extended Data Fig. 2d). Collectively, this indicated that nuclear fluctuations mirror the dynamics of osmotic changes within the cell.

As the results implicated both active cytoskeletal confinement and intranuclear osmotic changes in determining nuclear dynamics, we determined nuclear and chromatin composite stiffness by atomic force microscopy (AFM)-mediated force indentation spectroscopy[23]. Removal of pluripotency factors triggered nuclear stiffening within 5 min of perturbation (Fig. 2h). This effect was controlled by growth factors: adding FGF2 to basal medium prevented stiffening, while TGF-β1 had milder effect, and combining both restored the pluripotent nuclear mechanical state (Fig. 2h). Similar nuclear stiffening was observed when nuclear volume loss was induced by hypertonic shock (Extended Data Fig. 2e), suggesting that nuclear stiffening could be triggered by osmotic stress. Nuclear stiffening was also observed in cells where actin was depolymerized using Cytochalasin D (Extended Data Fig. 2f), confirming that growth factor removal impacted nuclear stiffness directly and not through modulating actomyosin organization. We asked whether this rapid stiffening was due to changes in nucleoplasmic or chromatin viscosity caused by osmotic stress. To measure the rheological properties of the nucleoplasm, we utilized nuclear genetically encoded multimeric nanoparticles (nucGEMs)[48]. Analysis of the mean square displacement of nucGEMs showed reduced diffusion and increased confinement in basal conditions compared with pluripotency medium (Fig. 2i–k and Supplementary Video 5), indicative of increased viscosity and macromolecular crowding in response to osmotic stress.

To determine whether YAP activity was regulated by nuclear tautness or flattening or by osmotic stress, we compared YAP dynamics between hyperosmotic shock and compression. Live imaging of hiPS cells with endogenously Halo-tagged YAP revealed a salt-and-pepper-like pattern of active YAP in the nucleus at steady state (Fig. 2l and Supplementary Video 6). Hypertonic stress did not substantially alter this localization pattern but rather triggered YAP relocalization to cell–cell borders (Extended Data Fig. 2g), indicating nuclear entry of YAP is not directly controlled by nuclear volume loss. By contrast, compression in pluripotency growth factors increased YAP nuclear entry (Fig. 2m and Supplementary Video 6), consistent with previous reports implicating nuclear deformation as a critical signal for YAP activation[42,49]. Interestingly, the same mechanical

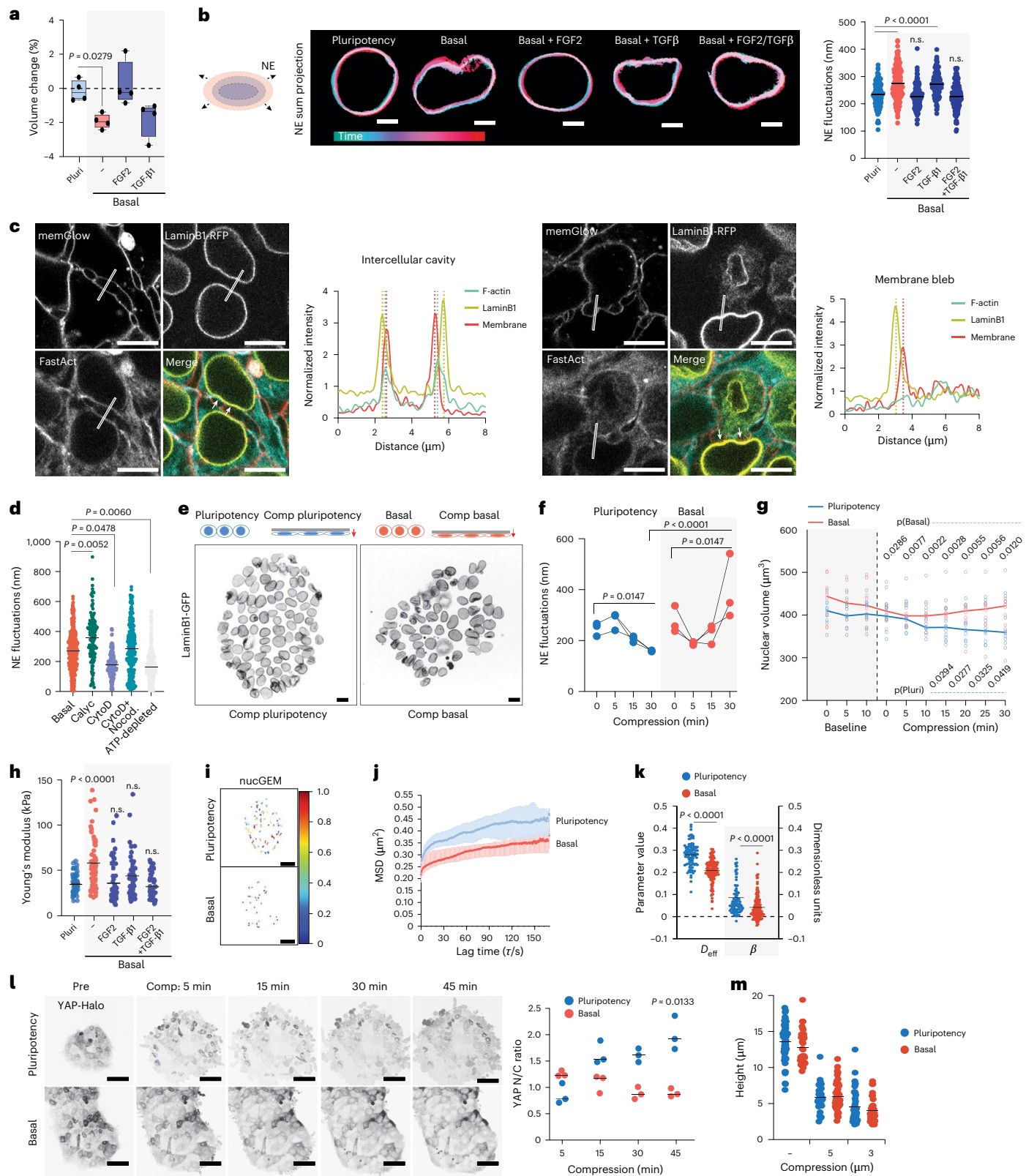

manipulation in the absence of these growth factors failed to strongly activate YAP, indicating that growth factor-controlled nuclear mechanics gate YAP activation (Fig. 2l,m). Consistently with such thresholding activity, subjecting hiPS cells to greater deformation (3 μm instead of 5 μm) resulted in YAP activation in the absence of growth factors (Extended Data Fig. 2h). Importantly, this effect was specific to YAP as compression decreased nuclear levels of GFP fused to a nuclear localisation signal (NLS-GFP) used as the control (Extended Data Fig. 2i).

Collectively, these experiments demonstrated the mechano-osmotic changes observed upon exit from pluripotency are directly controlled by pluripotency-maintaining growth factors. Transient nuclear deformation and volume loss result from a composite effect of active forces from the perinuclear cytoskeleton and intranuclear osmotic changes characterized by increased chromatin viscosity and macromolecular crowding. The osmotic changes manifest as altered force balances across the nuclear envelope, leading to fluctuations,

**Fig. 2 | Mechano-osmotic nuclear remodelling is a rapid response to FGF2 removal. a**, Quantification of change in nuclear volume upon exposure to culture medium/growth factors indicated (*n* = 4 independent experiments with 953 (Pluripotency; Pluri), 1,157 (Basal), 851 (FGF2) and 882 (TGF-β1) nuclei per condition; minimum-to-maximum box plots show 75th, 50th and 25th percentiles; Kruskal–Wallis/Dunn's). **b**, Representative projections of nuclear envelope fluctuations as a function of time upon pluripotency factor removal and adding back specific growth factors. Note that removal of pluripotency factors triggers fluctuations that can be rescued by adding back TGF-β1 and FGF2 (scale bars, 5 μm; *n* = 301 (Pluri), 268 (Basal), 320 (Basal + FGF), 382 (Basal + TGF-β1) and 386 (Basal + FGF2/TGF-β1) nuclei pooled across 3 independent experiments; ANOVA/Dunnett's). **c**, Representative snapshots and line scans of live imaging videos of LaminB1-RFP-tagged hiPS cells in basal medium, stained with FastAct and memGlow to label actin and plasma membrane, respectively. Left: perinuclear actin rings surrounding nuclei and intercellular cavities corresponding nuclear deformation. Right: blebs derived from a mitotic cell deforming the nucleus of a neighbouring cell (scale bars, 10 μm; images representative of 5 videos). **d**, Quantification of nuclear fluctuations from cells in basal medium with or without inhibitor treatments as indicated (*n* = 812 (Basal), 267 (Calyculin A (Calyc)), 357 (CytochalasinD (CytoD)), 522 (CytoD + Nocodazole (Nocod.)) and 998 (ATP-depleted) nuclei per condition pooled across 3 independent experiments; ANOVA/Dunnett's). **e,f**, A schematic of the experimental outline, representative images (**e**) and quantification (**f**) of nuclear envelope fluctuations in cells compressed (Comp) in pluripotency or basal medium for timepoints indicated. Note decreased fluctuations in pluripotency condition and an increase in basal medium (scale bars, 10 μm;

*n* = 3 independent experiments with 318, 337, 355 and 207 (Pluripotency 0, 5, 15 and 30 min, respectively) and 350,188, 320 and 151 (Basal 0, 5, 15 and 30 min, respectively) total nuclei per condition; ANOVA/Fischer's). **g**, Quantification of nuclear volume dynamics from of Sox2-GFP-tagged hiPS cells live imaged directly after a media change into pluripotency or basal medium, followed by compression. Line represents median volume and individual dots are average colony volumes at indicated timepoints (*n* = 10 colonies per condition pooled across 6 independent experiments). **h**, AFM force indentation experiments of iPS cell nuclei within 20 min of media switch. Note increased elastic modulus of cells in basal media conditions, restored by adding FGF2 (*n* = 69 (Pluri), 71 (Basal), 76 (Basal + FGF2), 85 (Basal + TGF-β1) and 74 (Basal + FGF + TGF-β1) nuclei pooled across 5 independent experiments; Kruskal–Wallis/Dunn's). **i**, Representative tracks of nucGEM particles. Colours represent average rate of diffusion per tracked particle (scale bars, 5 μm). **j,k**, Quantification of mean squared displacement (MSD) versus lag time (tau τ/s per nucGEM particle (**j**) and nucGEM diffusion ($D_{eff}$) and diffusivity exponent β (**k**) (*n* = 260 (Pluri) and 370 (Basal) cells pooled across 4 independent experiments; mean ± s.d.; Kruskal–Wallis/Dunn's). **l,m**, Representative snapshots of live imaging and quantification of HALO-tagged endogenous YAP localization (**l**) and nuclear height (**m**) in cells compressed to 5 μm height in pluripotency or basal medium. Note YAP nuclear entry in pluripotency condition but not in basal medium upon compression (scale bars, 30 μm; **l**, *n* = 3 independent experiments with 127 (Pluri) and 101 (Basal) total cells per condition; **m**, *n* = 56 cells (Pluri uncompressed), 42 (Basal uncompressed), 34 (5 μm Pluri), 47 (5 μm Basal), 37 (3 μm Pluri) and 41 (5 μm Basal) cells pooled across 3 independent experiments; ANOVA/Friedman).

whereas mechanical changes—possibly related to nuclear envelope tension—trigger YAP activity.

## Osmotic component of nuclear flattening primes chromatin for spontaneous differentiation

We next asked whether the observed mechano-osmotic changes in the nucleus impact cell fate transitions. To test this, we flattened cells in pluripotency medium for 5 or 30 min using the compression bioreactor (Fig. 3a) and characterized immediate cellular responses and potential heterogeneity by measuring genome-wide chromatin accessibility and transcription (single-cell multiome assay for transposase-accessible chromatin using sequencing (ATAC-seq) and RNA sequencing (RNA-seq); 10x Genomics platform). We also examined the long-term reversibility of potential changes by analysing a recovery condition 24 h after the 30 min compression. After 5 min of compression, we detected changes in chromatin accessibility and gene expression that became more prominent after 30 min (Fig. 3b, Extended Data Fig. 3a and Supplementary Table 1). This response was robust and stereotypic, as indicated by a cluster consisting almost exclusively of 30 min compressed cells predominantly in the ATAC-seq dataset (Fig. 3b), whereas the 24 h recovery condition clustered with uncompressed cells.

Overall chromatin accessibility reduced, with 5,131 regions losing accessibility and only 157 gaining accessibility at 30 min. To elucidate regulatory circuits between chromatin changes, transcription factors

and gene expression, we applied SCENIC+ analyses[50] to detect integrated gene regulatory networks after 5 or 30 min of compression and after 24 h recovery. The coregulator of the mechanosensitive transcription factor YAP, TEAD1 and the mechanosensitive POU2F1 (Oct1)[15] were upregulated regulons at 5 min of compression (Fig. 3c), while regulatory networks downstream of master pluripotency transcription factors, including SOX2, SOX4 and TCF4, and key master regulators of stress responses JUND, ATF4 and NRF2 peaked slower after 30 min of compression (Fig. 3c). Overall, chromatin accessibility and gene expression changes appeared largely reversible as the 24 h recovery condition closely resembled the control condition (Fig. 3b,c). Notably, no induction of apoptosis/necrosis genes, effects on cell cycle or altered regulation of nutrient and hypoxia-sensitive genes were observed (Extended Data Fig. 3b,c), indicating that compression did not trigger damage or cell death, influence cell cycling or prevent cell access to nutrients and oxygen.

To investigate the relationship between nuclear deformation and pluripotency, we used chromVAR[51] to predict transcription factor 'activity' based on the enrichment of binding motifs for increasingly accessible chromatin regions following compression-driven nuclear flattening. The strongest signature came from TEAD1–4-targeted motifs, as well as the SOX, POUF and SIX families of transcription factors involved in gastrulation, indicative of chromatin priming towards exit from pluripotency (Extended Data Fig. 3d). Differential gene expression analyses confirmed that genes involved in regulating

**Fig. 3 | Nuclear flattening primes chromatin for spontaneous differentiation. a**, Representative top views, side views and 3D reconstructions of LaminB1-RFP-tagged hiPS cells subjected to compression (scale bars, 10 μm; images representative of six independent experiments). **b**, UMAP of scRNA- and scATAC-seq from hiPS cells subjected to compression for timepoints indicated. **c**, A heatmap of predicted regulons enriched in compressed cells from SCENIC+ analyses of the multiome data. **d**, A schematic of experimental outline for genome-wide mapping of H3K27ac changes. **e,f**, Heatmap (**e**) and metaplot (**f**) analysis of mean H3K27ac levels at active promoters and predicted active enhancer regions. Note reduction in H3K27ac enrichment at promoters across all conditions and at enhancers in cells compressed in basal medium or exposed to hypertonic shock. **g**, UpSet plot showing an overlap of enhancers decommissioned in compression and hypertonic shock conditions. **h**, Venn diagram and Reactome pathway enrichment of compression-specific and shared

decommissioned enhancers as defined in **g**. **i**, A schematic of the experimental outline for the quantification of the nascent transcriptome. 4sU, 4-thiouridine. **j**, Quantification of RNA synthesis across conditions from TTseq. Note reduced synthesis across all conditions compared with pluripotency medium condition (*n* = 3 biological replicates per condition). **k**, Quantification of total changes in nascent RNA production across conditions relative to the pluripotency medium condition (*n* = $\log_2$FC of 19,288 genes computed from 3 biological replicates; Tukey's box plots show 75th, 50th and 25th percentiles). **l**, A heatmap of *z* scores from altered nascent RNA levels of relevant transcripts from TTseq quantified by DESeq2. Note increased levels of IEGs specifically in cells compressed in pluripotency medium while key pluripotency and growth factor regulators are repressed. Comp, compression; Hyper, hypertonic; Rec, recovery; pluri, pluripotency.

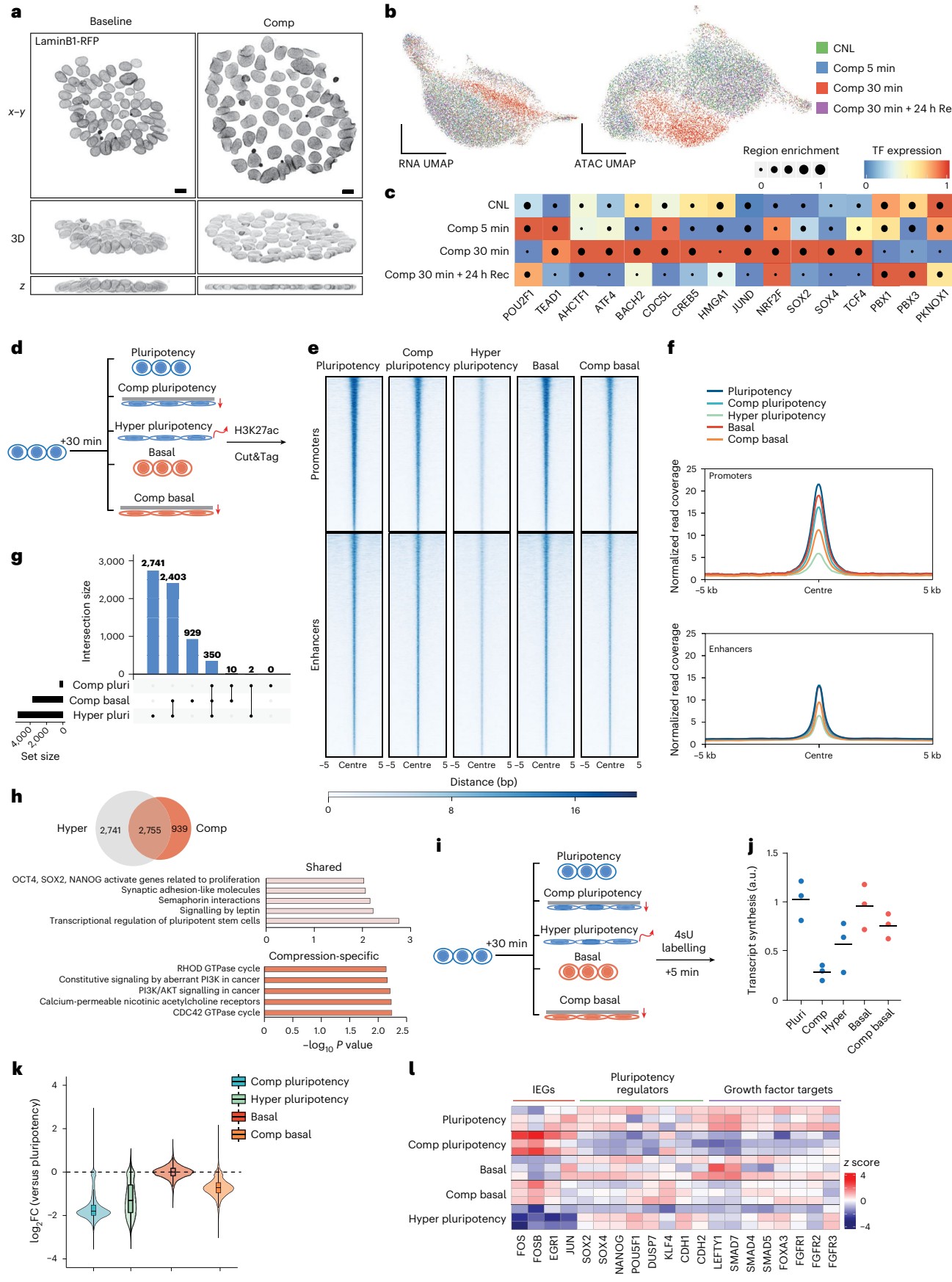

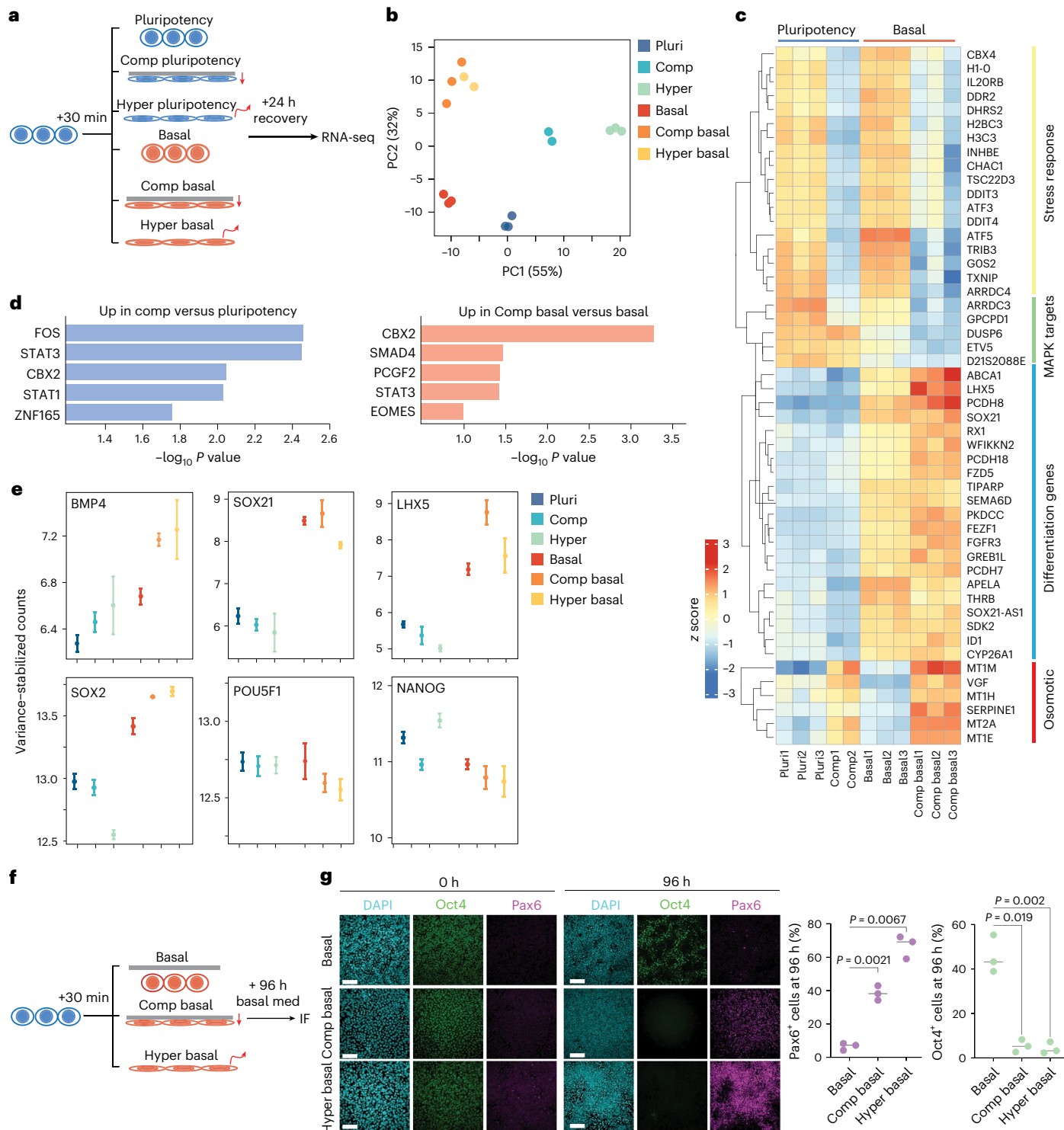

**Fig. 4 | Mechano-osmotic signals control kinetics of lineage commitment.** **a,b**, Schematic of the experimental outline (**a**) and PCA plot (**b**) of bulk RNA-seq in cells subjected to compression (Comp) or hypertonic (Hyper) shock and recovery in the indicated media conditions. Note the divergence in transcriptomic responses to hypertonic shock and mechanical compression in pluripotency medium, and their convergence in basal medium. **c**, A heatmap of the top variable genes from bulk RNA-seq in conditions indicated. Note the increase in differentiation gene expression in cells compressed in basal medium. **d**, Transcription factor binding enrichment analysis from genes upregulated in the bulk RNA-seq for the indicated conditions. **e**, Representative examples of gene expression changes across the conditions. Note increased expression of differentiation genes in basal compressed and hypertonic shock conditions ($n = 2$ biological replicates (Comp)/3 biological replicates rest, pooled; mean ± s.d.). **f,g**, Schematic of experimental outline (**f**), representative images and quantification (**g**) of hiPS cells immunostained for Oct4 and Pax6 after compression or hypertonic shock. Note the increased differentiation in compressed cells or cells exposed to hypertonic shock (scale bars, 75 μm; $n = 3$ independent experiments with 900 (control); 1,190 (compression); 949 (hypertonic) total nuclei per condition; ANOVA/Kruskal–Wallis). PC, principal component; basal med, basal media; pluri, pluripotency.

the actomyosin cytoskeleton including known YAP target genes (*AMOTL2*, *TAGLN* and *CCN2*), and genes involved in heat stress and osmotic stress (*HSPA1A*, *HSPA1B* and *FOS*) were upregulated at 30 min (Extended Data Fig. 3e). However, in contrast to chromatin profiling, differential gene expression analysis further revealed that expression of genes involved in pluripotency such as *SOX2*, *OCT4* and *DUSP7* were largely unchanged or even mildly upregulated upon 30 min of compression (Extended Data Fig. 3e,f). Thus, whereas the chromatin profile indicated priming chromatin towards exit from pluripotency, activity of predicted transcription factors such as YAP/TEAD and RNA profiling described a more complex transcriptional state without a strong signature of loss of pluripotency.

As nuclear flattening by mechanical compression triggered chromatin priming towards lineage transition but no sustained fate transition, we reasoned that pluripotency-promoting growth factors were preventing exit from pluripotency. To test this, we quantified genome-wide changes in promoter and enhancer states by profiling the epigenetic mark of active enhancers, H3K27ac, in pluripotency or basal media with or without compression for 30 min (Fig. 3d and Extended Data Fig. 4a). To separately evaluate the impact of osmotic stress, we included a hypertonic shock condition. Intriguingly, removal of pluripotency factors (basal medium) or compression in pluripotency medium triggered a decrease in H3K27ac at promoters but had no strong effect on enhancers (Fig. 3e,f and Extended Data Fig. 4a). By contrast, compression in basal medium caused a strong reduction of H3K27ac both at promoters and enhancers, similar to hypertonic shock (Fig. 3e,f). The decommissioned enhancers showed high overlap, with many enhancers affected by compression in basal media conditions also changed in hypertonic shock conditions (Fig. 3g,h). Annotating the enhancers by assigning them to the nearest gene revealed that the shared enhancer changes were enriched for key pluripotency genes such as *SOX2*, *LIN28A* and *ZIC3* (Fig. 3h and Supplementary Table 2). By contrast, compression-specific decommissioned enhancers were enriched for cytoskeletal regulators and PI3Kinase signalling, whereas hyperosmotic-specific enhancers showed enrichment for FOXO targets (Fig. 3h, Extended Data Fig. 4b and Supplementary Table 2).

The strong effects of compression in decommissioning promoters and enhancers, similar to hypertonic stress, suggested a global impact on transcription. To quantify this and understand the immediate transcriptional responses to mechanical deformation in different contexts, we analysed the nascent transcriptome (transient transcriptome sequencing, TTseq[52]) after 30 min exposure to pluripotency-promoting culture conditions or basal medium, with or without compression, and in response to hypertonic shock as control for osmotic regulation (Fig. 3i). Quantification of RNA synthesis from nascent RNA labelled during the last 10 min of perturbations revealed reduced levels of global transcription upon compression and hypertonic shock (Fig. 3j,k). This transcriptional repression was strongest in compression in pluripotency medium. Further analysis of differential gene expression between compressed conditions revealed decreased expression of more than

15,000 genes in the compressed pluripotency condition compared to pluripotency medium control and over 8,000 genes compared with compression in basal medium, including mediators of differentiation (Extended Data Fig. 5a). Meanwhile, although transcription was strongly repressed in cells compressed in pluripotency medium, classical differential gene expression analysis of the nascent transcriptome revealed strong activation of immediate early genes (IEGs; for example, JUN, FOS and EGR1), a class of genes that are rapidly and transiently induced in response to stress or growth factor stimuli[53–55] (Fig. 3l, Extended Data Fig. 5b and Supplementary Table 3). This was consistent with the multiome analyses showing increased expression of JUN/FOS targets in this condition (Extended Data Fig. 3e). IEG induction was less apparent upon 30 min compression in basal medium (Fig. 3l and Extended Data Fig. 5c) pointing to biochemical context dependency of nuclear mechanotransduction in hiPS cells.

The notion of pluripotency-promoting growth factors gating the transcriptional response to mechanical flattening was supported by the kinetics of osmosensitive p38 MAPK. While p38 was strongly activated by compression in basal medium at 5 min and downregulated at 30 min, coinciding with transcriptional repression, p38 activation and transcriptional repression only became apparent in pluripotency medium at 30 min of compression, the timepoint of nascent RNA profiling (Extended Data Fig. 5d). Slower p38 activation was consistent with the kinetics of nuclear volume loss that peaked at 30 min in this condition (Fig. 2g). Importantly, adding FGF2 into the basal medium suppressed p38 activation by compression, whereas TGF-β1 did not have a strong effect, confirming that FGF2 signalling controls the mechano-osmotic properties of the nucleus and its osmotic response to deformation (Extended Data Fig. 5e).

Collectively these data reinforced that mechanical deformation of the nucleus has two components: a mechanical component that impacts YAP activity and cytoskeletal and extracellular matrix gene expression, and an osmotic stress component sufficient to induce a chromatin state priming towards exit from primed pluripotency in hiPS cells. Furthermore, pluripotency growth factor-mediated signals determine the mechano-osmotic state of the nucleus. In pluripotency medium, osmotic chromatin priming is weaker and transient, and while promoters are strongly impacted by compression, enhancers remain largely unchanged, locking existing cellular fates. Removal of pluripotency factors enhance the osmotic component of deformation to efficiently remodel enhancers.

## Mechano-osmotic signals control kinetics of lineage commitment

To directly address the role of mechano- and osmosensitive enhancer remodelling on long-term transcriptional memory and lineage commitment, we performed bulk RNA-seq 24 h after recovery from 30 min compression, either in pluripotency maintenance conditions or in basal medium, comparing these scenarios with hypertonic shock (Fig. 4a). Principal component analysis revealed that either transient

**Fig. 5 | Osmotic pressure controls CBX2 condensation to gate gene repression. a**, Heatmap and Euclidian distance dendrogram of differentially abundant phosphosites quantified by mass spectrometry in cells subjected to compression (Comp) or hypertonic (Hyper) stress. **b**, Distance-based clustering of phosphosites and GO-term analyses show changes specific or common to the specific stresses. **c**, Example heatmaps of differentially abundant phosphoproteins from **b**. **d**, Representative snapshots of live imaging and quantification of CBX2 condensation dynamics from hiPS cells with an endogenously tagged *CBX2* allele. Note the rapid dissolution and re-establishment of condensates upon removal of pluripotency (pluri) factors or exposure to axial compression or hypertonic shock. Arrows mark dissolving condensates; dotted arrows mark newly formed condensates (scale bars, 20 μm; *n* = 136 (Pluri), 172 (Basal), 99 (Comp); 146 (Hyper) total nuclei tracked over time and pooled across 3 independent experiments; mean ± s.e.m.; two-way-ANOVA/ Tukey's). **e**, Heatmap and Euclidian distance dendrogram of differential CBX2

occupancy quantified by CUT&Run in cells subjected to removal of pluripotency factors (basal), compression (comp) or compression in basal medium, normalized to pluripotency condition. Asterisks mark transcription factors that control differentiation. **f**, Reactome analysis of genes in clusters 1 and 4 implicate metal-binding genes with reduced CBX2 occupancy in both basal medium and basal medium compression condition, whereas differentiation genes show reduced CBX2 in compression in basal medium. **g**, Representative tracks of genes with altered CBX2. **h**, Model of how intranuclear and cytoskeletal forces influence hiPS cell exit from pluripotency. Under conditions with pluripotency growth factors (GFs), nuclear mechanics are maintained and differentiation is prevented under volumetric stress, restoring pluripotency gene expression. In the absence of pluripotency GFs, osmotic stress leads to nuclear envelope fluctuations and CBX2 condensation, priming chromatin for a cell state transition. This ultimately causes derepression of CBX2 target genes, facilitating exit from pluripotency.

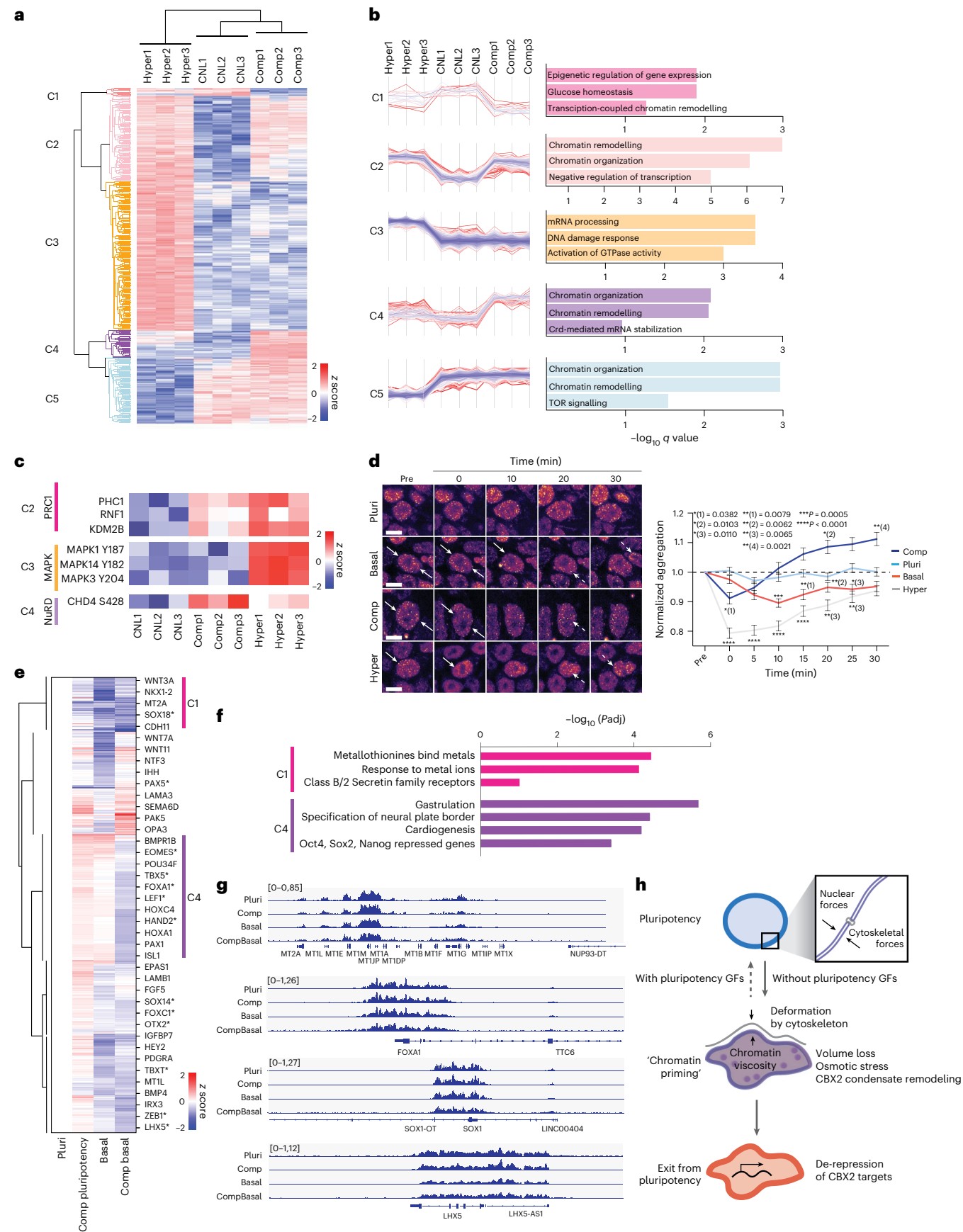

hypertonic shock or compression triggered distinct long-term transcriptional responses under pluripotency conditions. By contrast, basal media conditions promoted a transcriptional signature of spontaneous differentiation, and compression and hypertonic shock in basal media conditions led to a highly correlated transcriptional state (Fig. 4b and Supplementary Table 4). This further supported that growth factor signalling gates the balance between the mechanical and osmotic components of compression. Furthermore, in the pluripotency condition, the long-term transcriptional response to transient compression was less pronounced (65 significantly upregulated genes in pluripotency versus 179 significantly upregulated genes in basal media conditions; $P < 0.05$, log$_2$fold change (FC) >1) showing a gene expression signature of FOS and STAT3 transcriptional targets and genes involved in regulation of cytoskeleton and lipid composition (Fig. 4c–e and Extended Data Fig. 6a), consistent with the weaker effect of this perturbation on enhancers. By contrast, transient compression in basal media led to upregulation of a gene regulatory network of 'non-lineage-specific' differentiation and downregulation of pluripotency genes compared with basal media alone (Fig. 4c–e and Extended Data Fig. 6a,b). A similar activation of differentiation gene signatures was observed in the hypertonic condition in basal medium (Extended Data Fig. 6b). The osmosensitive metallothionine gene family[56] was most upregulated in cells transiently compressed in basal media conditions (Fig. 4c), further indicating the long-term transcriptional effects of compression resemble hypertonic shock. Collectively, gene activation profiles indicated that withdrawal of pluripotency factors, together with nuclear volume loss, led to acceleration of differentiation.

To assess whether osmotic-stress-mediated induction of differentiation gene expression was sufficient to influence long-term lineage progression, we induced spontaneous differentiation of hiPS cells by maintaining them in basal medium after 30 min compression or hypertonic shock (Fig. 4f). Strikingly, transient 30 min compression or hypertonic shock in basal media accelerated spontaneous ectodermal-like differentiation after 96 h (Fig. 4g). Consistent with the notion that the osmotic component of deformation is driving the effect, exposing compressed cells to a brief (10 min) hypo-osmotic shock attenuated the effect of compression in enhancing spontaneous differentiation (Extended Data Fig. 6c). In line with the transcriptome analyses, transient 30 min compression in pluripotency medium did not enhance and even delayed biochemically induced differentiation into ectoderm (Extended Data Fig. 6d). Taken together, hyperosmotic stress, in the absence of pluripotency factors, accelerates spontaneous exit from pluripotency. By contrast, osmotic chromatin priming in the presence of pluripotency factors is not capable of driving cell state transitions and even has a mildly reinforcing effect on the preexisting cell state.

## Osmotic pressure controls CBX2 condensation to gate gene repression

We next sought to unravel the distinct mechanisms by which mechanical and osmotic stress gate chromatin priming towards exit or sustained pluripotency maintenance. To compare these stimuli, we quantified the phosphoproteome of cells subjected to 5 min of hypertonic shock or compression in pluripotency conditions, where the mechanical stress response dominated the osmotic stress response. Hierarchical clustering of significantly altered phosphosites revealed clusters either shared by or specific to either stimulus (Fig. 5a). A small group of phosphosites specifically regulated by compression were related to chromatin remodelling, and compression had a strong impact on the phosphorylation of the central component of the NuRD chromatin remodelling complex, CHD4 (cluster 4; Fig. 5b,c and Supplementary Table 5). CHD4 controls transcriptional repression of specific genomic loci and is under the control of YAP[57]. Interestingly, phosphosites upregulated by both compression and hyperosmotic shock were related both to chromatin remodelling and negative regulation of transcription (cluster 2;

Fig. 5b,c and Supplementary Table 5). Specifically, phosphorylation of Polycomb repressive complex 1 (PRC1)[58] components were upregulated (Fig. 5c). Finally, phosphosites upregulated specifically in response to hypertonic shock were activators of GTPase activity (mainly cytoskeletal proteins such as Vimentin, Pak1, Myo9B and Dock2) and MAPK1, MAPK3 and MAPK14, corresponding to activation of ERK1/2 and p38 MAPKs (cluster 3; Fig. 5c and Supplementary Table 5). Taken together, these results highlight distinct (YAP) and overlapping (PRC1) pathways activated by compression and hyperosmotic shock, and suggest potential regulatory mechanisms of chromatin priming.

We asked whether differential YAP and ERK activation observed by compression versus osmotic stress would explain the enhanced spontaneous differentiation observed in the absence of pluripotency factors. To this end, we addressed the long-term transcriptional response of withdrawal of pluripotency factors, combined with transient compression and hypertonic shock (30 min followed by 24 h recovery) on differentiation gene expression. These experiments confirmed that the osmotic stress response (indicated by metallothionein gene induction) and spontaneous differentiation were strongly triggered by compression in basal medium. Inhibition of ERK and YAP did not prevent spontaneous differentiation, but instead enhanced expression of differentiation genes, especially in the context of compression in pluripotency medium (Extended Data Fig. 7a). This effect is consistent with the established role of ERK and YAP in alleviating osmotic and mechanical stress, respectively[59,60]. Thus, these experiments suggested ERK and YAP activation function to mitigate mechano-osmotic stress and, thus, their downregulation accelerates exit from pluripotency.

As ERK and YAP signalling were not sufficient to explain why osmotic stress triggers chromatin priming for pluripotency exit, we returned to examine chromatin regulators phosphorylated in these conditions. Focusing on protein complexes co-regulated in the phosphoproteome, we turned to the PRC1 pathway, supported by the enrichment of target genes of its central component CBX2 in the 24 h recovered transcriptome in response to compression in basal medium (Fig. 4d). As serine phosphorylation is associated with propensity to form condensates[61] and macromolecule concentration is probably altered by changes in nuclear volume and nucleoplasm crowding and viscosity[62], we analysed CBX2 condensation in response to compression and upon removal of pluripotency factors or induction of hypertonic stress. To this end, we generated an endogenously tagged CBX2-GFP allele for live imaging of CBX2 dynamics. Live imaging revealed that removal of pluripotency factors, axial compression and hypertonic treatment triggered dynamic remodelling of CBX2 condensates, characterized by dissolution of large condensates, followed by reassembly (Fig. 5d and Supplementary Video 7). While these changes were consistent across all manipulations, the dynamics and magnitude of the changes were stress-specific. Hypertonic stress had the strongest effect in the initial condensate dissolution, whereas compression showed rapid recovery and individual condensates were larger than in the pluripotency medium control. Immunofluorescence staining of untagged, endogenous CBX2 confirmed the localization pattern of CBX2 in nuclear condensates and supported the live imaging data with increased condensation and condensate redistribution (Extended Data Fig. 7). In response to compression, remodelled condensates frequently localized to the nuclear periphery (Extended Data Fig. 7b).

The dynamic behaviour of condensate dissolution and subsequent reassembly suggested potential derepression of CBX2/PRC1 target genes. To test this and the long-term effects of mechano-osmotic CBX2 condensation, we profiled CBX2 occupancy genome-wide 24 h after removal of pluripotency-promoting factors with or without an initial 30-min pulse of compression. As expected, CBX2 was found occupying differentiation genes as well as known PRC1/2 targets[63] (Extended Data Fig. 7c and Supplementary Table 6). While removal of pluripotency factors reduced CBX2 occupancy at differentiation

genes and transcription factors regulating differentiation programmes (Fig. 5e, marked by asterisks), the reduction was strongest when a pulse of axial compression was applied (Fig. 5e,f). Consistent with the reversibility of chromatin changes observed in the ATAC-seq, compression in pluripotency medium had only a minor effect on CBX2 occupancy long term (Fig. 5e,f). By contrast, upon compression in the basal medium CBX2 occupancy was substantially reduced at differentiation genes and genes repressed by pluripotency factors such as *FOXA1*, *SOX1* and *LHX5*, and *HOX* genes involved in differentiation (Fig. 5e–g). The osmoresponsive metallothionein gene cluster was occupied by CBX2, and occupancy was moderately reduced by compression in pluripotency medium, but more strongly affected in the basal medium or upon compression in the basal medium (Fig. 5e–g). Finally, to confirm the causative role of CBX2 loss on exit from pluripotency, we depleted CBX2 from hiPS cells using CRISPR interference (CRISPRi)[64] (Extended Data Fig. 7d). As predicted, depletion of CBX2 accelerated spontaneous differentiation of hiPS cells in basal medium (Extended Data Fig. 7e), indicating that CBX2 activity maintains pluripotency or prevents differentiation.

Collectively these experiments demonstrated that mechano-osmotic forces trigger redistribution of CBX2 condensates and genome-wide occupancy, derepressing genes involved in buffering against osmotic stresses as well as differentiation genes.

## Discussion

The highly regulated 3D organization of genomes arises from interactions across scales, from DNA loops to chromatin domains and higher-order compartments. Despite the dynamics and stochasticity of transcription and variability of chromatin architecture at the single-cell level, these processes are tightly regulated and gate cell fate transitions. The dynamic, probabilistic properties of genome organization have raised questions on how cell states are generated and maintained among single cells and how cell populations are able to coordinately transition between states[65,66]. Here, we demonstrate that rapid growth-factor-driven mechano-osmotic remodelling of chromatin architecture and nuclear mechanics influences lineage transition in pluripotent cells. We observe that, in addition to controlling specific gene expression programmes on longer time scales, removal of pluripotency growth factors mediates rapid, minute-scale changes in the mechano-osmotic state of the nucleus, characterized by nuclear deformation and decreased nuclear volume, leading to increased nucleoplasm viscosity and macromolecular crowding. We propose that the osmotic component of nuclear deformation enhances accessibility of differentiation genes, and this transient and reversible chromatin remodelling, while not sufficient to facilitate cell fate conversion, lowers the energy barrier for a signalling factor-driven cell state change and possibly synchronizes chromatin state of a heterogeneous cell population to be equally receptive for the specific signalling factors (Fig. 5h).

Our data further indicate that nuclear deformation triggered by mechanical compression has two components: a mechanical component that triggers specific enhancer changes and transcriptional responses probably mediated by mechanosensitive transcription factors including YAP. The second component is osmotic stress arising from nuclear volume loss, triggering a transcriptional reset and chromatin remodelling. Our work identifies the PRC1–CBX2 axis as a mechano-osmotically sensitive chromatin regulator. This is consistent with previous work showing that components of the CBX2–PRC1 complex form condensates that colocalize with chromatin and with genes relevant to their function[67–69], as well as the importance of condensate formation as a mechanism to buffer cells from osmotic pressure and heat stress[61].

While a large number of studies have implicated both growth factor signalling and mechanical forces regulate gene expression, the epigenome, and cell states[15,70–72], it has remained unclear how forces interplay with biochemical signals to determine transcriptional outcomes. This study provides a foundation for understanding the role of mechanochemical feedback loops, where biochemical signals are capable of altering mechanical properties of the nucleus to change chromatin states, while extrinsic forces have the capacity to alter how biochemical signals are interpreted by cells to impact the kinetics of cellular state transitions.

## Online content

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

¹Department of Cell and Tissue Dynamics, Max Planck Institute for Molecular Biomedicine, Münster, Germany. ²Stem Cells and Metabolism Research Program, Faculty of Medicine, University of Helsinki, Helsinki, Finland. ³Laboratory of Molecular Biology, National Institute of Diabetes and Digestive and Kidney Diseases, National Institutes of Health, Bethesda, MD, USA. ⁴Systems Biology of Gene Expression, National Cancer Institute, National Institute of Health, Bethesda, MD, USA. ⁵Bioinformatics Service Unit, Max Planck Institute for Molecular Biomedicine, Münster, Germany. ⁶High-Throughput Imaging Facility, National Cancer Institute, National Institute of Health, Bethesda, MD, USA. ⁷Gynecology and Obstetrics, Clinicum, University of Helsinki, Helsinki, Finland. ⁸Helsinki University Hospital, Reproductive Medicine Unit, Helsinki, Finland. ⁹iPSC Core, National Heart, Lung, and Blood Institute, National Institutes of Health, Bethesda, MD, USA. ¹⁰CECAD Research Center, University of Cologne, Cologne, Germany. ¹¹Department Cell Biology of the Skin, Cologne Excellence Cluster on Cellular Stress Responses in Aging Associated Diseases (CECAD), Center for Molecular Medicine Cologne, University Hospital Cologne, University of Cologne, Cologne, Germany. ¹²Helsinki Institute of Life Science, Biomedicum Helsinki, University of Helsinki, Helsinki, Finland. ¹³These authors contributed equally: Kaitlin P. McCreery, Aki Stubb. ¹⁴These authors jointly supervised this work: Yekaterina A. Miroshnikova, Sara A. Wickström. ✉e-mail: kate@mpi-muenster.mpg.de; sara.wickstrom@mpi-muenster.mpg.de

## Methods

### Human embryos and ethical issues

Collection and experiments on human embryos were approved by the Helsinki University Hospital Ethics Committee (diary number HUS/1069/2016). Research permission was approved by the Helsinki University Hospital Research Committee. Couples that had been treated for infertility at the Helsinki University Hospital Reproductive Medicine unit were offered a possibility to donate their cryopreserved embryos to research after termination of infertility treatments and expiration of embryo preservation timespan. Human surplus blastocysts were donated for research with an informed consent, and patients understood that donating embryos to research is voluntary. Patients were informed about the research and research methods that were used to perform the experiments. Patients were not compensated for embryo donations, and they were offered counselling. Human embryos were not created for research, nor were they manipulated for research purposes. The culture of embryos was terminated at blastocyst stage, that is, several days before the expected primitive streak formation or onset of gastrulation. We strictly follow local legislation, ethical guidelines and regulations, as well as principles laid out in the International Society for Stem Cell Research guidelines.

Day 5 and day 6 blastocysts (4AA, 4AB or 4BA) vitrified with either Kitazato Vitrification Media or Cryotech Vitrification Solutions (110, Cryotech) were warmed with Thawing Media. Cryotech strips (Cryotech) were quickly immersed in Thawing Solution at 37 °C for 1 min. The blastocysts were gently aspirated and transferred to Diluent Solution at room temperature for 3 min. Subsequently, blastocysts were transferred to Washing Solution (WS) 1 for 5 min and then to the surface of WS2 and after sinking to the bottom; this procedure was repeated, after which they were transferred to culture media (GTL, Vitrolife) and cultured in a GERI incubator for 15–16 h. All thawing solutions were from Kitazato.

### Naive hES cell culture and blastoids

Naive H9 hES cells (WA09, Wicell), were converted from primed to naive pluripotency using NaïveCult Induction Kit (StemCell Technologies), and cultured on mitotically inactivated CF-1 mouse embryonic fibroblasts (Gibco) in NaïveCult Expansion Medium (StemCell Technologies) in humidified 5% $O_2$, 5% $CO_2$ at 37 °C. Y-27632 (10 μM; Selleckchem) was added for the first 24 h.

Blastoids were generated according to established protocols[73]. Naive H9 hES cells were dissociated with Accutase (Gibco), inactivated in 0.1% bovine serum albumin (BSA)–DMEM–F12, passed through a 40-μm cell strainer (Corning) and centrifuged at 300$g$ for 5 min. Cells were plated onto 0.1% gelatin-coated dishes in NaïveCult Expansion Medium with 10 μM Y-27632 and incubated at 5% $O_2$, 5% $CO_2$ at 37 °C for 90 min. Non-adherent cells were collected, filtered through a cell strainer, resuspended in N2B27 basal medium and plated onto Aggrewell 400 (StemCell Technologies) plates at $6 \times 10^4$ cells per well in 500 μl N2B27 with 10 μM Y-27632, 0.3% BSA. Medium was changed 24 h after plating (day 1), and 500 μl of 2× PALLY medium was added to each well. On day 2, half of the medium was replaced with fresh 1× PALLY medium. On day 3, 500 μl of the medium was removed and 800 μl of LY medium was added. On day 4, 800 μl of the medium was replaced with fresh LY medium. PALLY medium consisted of N2B27 basal medium supplemented with 1 μM PD0325901 (MedChemExpress), 1 μM A83-01 (MedChemExpress), 10 ng ml$^{-1}$ LIF (Qkine), 3 μM lysophosphatidic acid (Tocris), 10 μM Y-27632 and 1× penicillin–streptomycin. LY medium consisted of N2B27 basal medium supplemented with 500 nM LPA and 10 μM Y-27632 (Selleckchem).

### hiPS cells

hiPS cells were from Allen Cell Collection (Coriell). Sox2-GFP-tagged cells were used for experiments unless indicated otherwise; LaminB1-RFP and CRISPRi lines were used wherever indicated. Cells were cultured on Matrigel-coated plates in mTeSR (StemCell technologies) medium and placed in E6 basal medium (Gibco) at the onset of experiments wherever indicated, and 5 ng ml$^{-1}$ TGF-β1 and 100 ng ml$^{-1}$ bFGF (both from StemCell Technologies) were supplemented to basal media where indicated.

Sox2-GFP hiPS cells were differentiated into mesoderm, endoderm and ectoderm lineages according to the STEMdiff Trilineage Differentiation protocol (StemCell Technologies, #05230). In brief, cells were seeded on Matrigel-coated plates $5 \times 10^4$ cells cm$^{-2}$ for mesoderm induction or $2 \times 10^5$ cells cm$^{-2}$ for endoderm and ectoderm induction, and were treated the following day with their respective STEMdiff media until analysed.

### 2D gastruloids

Micropatterns measuring 600 μm were generated using ultraviolet lithography (Alveole Primo) on plastic eight-well dishes (Ibidi) according to the manufacturer's protocol and coated with Matrigel (1:100). Cells were dissociated with Accutase (Thermo Fischer), pelleted by centrifugation, resuspended in mTeSR1 with 10 μM Y-27632 (StemCell Technologies) and plated at 400,000 cells per well. After 1 h, the patterns were rinsed and incubated in mTeSR1 + 10 μM Y-27632 medium for 2 h, after which Y-27632 was removed. Two hours later, 50 ng ml$^{-1}$ BMP4 (Miltenyi Biotech) was added.

### Halotag-YAP1 and CBX2-GFP knock-in hiPS cells

For YAP1, 0.6 pmol pUC57-Halotag-N-YAP1 plasmid (GenScript; 400 bp/ea homology arms flanking YAP1 (NM_001130145.3) start codon and Halotag coding sequence followed by GGSGGS linker) was mixed with 12.2 pmol Alt-R HiFiCas9 v3 (IDT #1081061) and 17.5 pmol single guide RNA (sgRNA; Synthego,) to transfect 226,000 hiPS cells using Nucleofector 4D 16-well cuvette, 20 μl P3 Primary Cell Nucleofector Solution (Lonza #V4XP-3032), programme CA-137. After nucleofection, the hiPS cells were cultured in rhLaminin-521 (Thermo #A29249)-coated 24-well plates in StemFlex medium (Thermo #A3349401) plus 1 μM HDR enhancer v2 (IDT #10007921) and 1× RevitaCell (Thermo #A2644501) at 32 °C overnight. Medium was subsequently changed to StemFlex plus 0.5× RevitaCell. After 3 days, cells were moved to 37 °C with StemFlex medium change every 2 days. After 16 days, cells were labelled with 1 μM Halotag Oregon Green ligand (Promega #G2802) and flow cytometry sorted (BD FASCMelody) based on fluorescence as single cells into Matrigel-coated 96-well plates. To confirm two copies of allele and no vector integration, droplet digital (dd) PCR (Bio-Rad QX200) with specific primers and probes for Halotag, AmpR (vector backbone) and reference hRPP30 was performed, further confirmed by genomic PCR of a 2.5-kb fragment, followed by Sanger sequencing.

For the mEGFP-CBX2 reporter, 400 bp/ea homology arms flanking the CBX2 (NM_005189.3) start codon were used, into which the mEGFP coding sequence was inserted, followed by a GGSGGS linker sequence (GenScript). The Kozak sequence in CBX2 was changed from GGCAGC to GGCACC to prevent CRISPR–Cas9 recutting. Then, 5 μg donor plasmid was mixed with 120 pmol Alt-R HiFiCas9 v3 (IDT #1081061) and 200 pmol sgRNA (Synthego) to transfect 800,000 hiPS cells as described above, and cultured and sorted as described above. To confirm genotype, sorted clones were first screened by 5'-junction and 3'-junction PCRs, which were further confirmed by Sanger sequencing. Clones with correct junction PCRs were then confirmed by ddPCR with one or two copies of mEGFP knock-in allele and no vector integration using mEGFP, AmpR and reference hRPP30 assay (dHsaCP1000485, Bio-Rad #10031243). Fifteen out of 32 screened clones that had one copy of mEGFP and zero copies of AmpR were selected as heterozygous mEGFP-CBX2 knock-in clones. The non-knock-in alleles in these clones were further screened by wild-type allele PCR and Sanger sequencing. For all primers and probes, see Supplementary Table 7.

### Immunofluorescence and confocal microscopy

Human blastocyst stage embryos were fixed in 3.8% paraformaldehyde for 15 min at room temperature, washed three times in washing buffer

(0.1% Tween20–Dulbecco's phosphate-buffered saline (DPBS)) and permeabilized in 0.5% Triton-X-100–DPBS for 15 min at room temperature. After washing the embryos in washing buffer, embryos were incubated in Ultra Vision Protein block (Thermo Fisher Scientific) for 10 min at room temperature, followed by incubation with primary antibodies at 4 °C overnight. After three washes in washing buffer, secondary antibodies (1:500 in washing buffer) were added for 2 h at room temperature. Embryos were washed and counterstained with DAPI (1:500) and imaged in DPBS either on optical-grade plastic μ-slide eight-well chambers (Ibidi) or on glass-bottom dishes (Mattek).

Cells were fixed in 4% paraformaldehyde, permeabilized with 0.3% Triton X-100 in phosphate-buffered saline (PBS) and blocked in 5% BSA. Samples were subsequently incubated overnight in primary antibody in 1% BSA/0.3% Triton X-100/PBS, followed by PBS washed and incubation with secondary antibody in 1% BSA/0.3% Triton X-100/PBS. Cells were imaged directly after staining in PBS or mounted in Elvanol.

The following antibodies were used: OCT3/4 (Santa-Cruz Biotechnology, sc-5279; 1:1,000), Brachyury (R&D Systems, AF2085; 1:1,000), GATA6 (AF1700, RnD Systems; 1:200), NANOG (D73G4, Cell Signaling Technologies; 1:200), LAMINB1 (66095-1-Ig, Proteintech; 1:200), Pax6 (Invitrogen, #42-6600; 1:1000), SOX1 (R&D Systems, AF3369; 1:200), SOX7 (R&D Systems, AF1924; 1:1,000), YAP1 (Santa Cruz sc-101199; 1:200), p38 MAPK phosphoThr180/Tyr182 (Cell Signaling Technologies; 1:800) and CBX2 (Thermo Fisher PA-582812; 1:800). Alexa Fluor 488-, 568-, 594- and 647-conjugated secondary antibodies (all from Invitrogen) were used at 1:500 dilution.

Fluorescence images were collected by laser scanning confocal microscopy (LSM980; Zeiss) with Zeiss ZEN Software (Zeiss ZEN v.3.7), or with Andor Dragonfly 505 spinning disk confocal (Oxford Instruments) equipped with 488-nm and 546-nm lasers, and Zyla 4.2 sCMOS camera using 40×, 63× or 100× immersion objectives and Fusion software (v.2.3.0.44, Andor). Images were acquired at room temperature using sequential scanning of frames of 1-μm confocal planes (pinhole 1).

### Live imaging

Samples were mounted on a Andor Dragonfly spinning disc confocal (Oxford Instruments) equipped with an environment chamber (5% $CO_2$ and 37 °C). Images were acquired with Fusion software (v.2.3.0.44, Andor). For nuclear volume measurements LaminB1-RFP hiPS cells were live imaged for 10 min (growth factor experiment) or 40 min (cell compression experiments) acquiring full z-stacks from the same colony for each timepoint using a 63× water immersion objective. For 2D gastruloid micropatterns, cells were imaged with a 20× air objective, the z-step was 0.75 μm and images were captured every 1 h. For nuclear envelope fluctuations, cells were imaged with a 63× oil immersion objective using a high frame rate acquisition mode (150 ms per frame) for 5 min. For CBX2 dynamics, mEGFP-CBX2 hiPS cells were imaged with a 63× water immersion objective, focusing on single focal plane at the widest point of the nuclei. Images were obtained at at 5 min per frame.

### Segmentation and image analysis

**Nuclear volume quantification.** Volumes were quantified using ImageJ[74]. Four-dimensional live imaging videos were filtered using a 3D median filter and subsequently bleach corrected using simple ratio correction. Individual nuclei were identified manually and marked with an oval selection stored to a region of interest (ROI) manager. Seeds were then expanded using the Limeseg-plugin[75]. The segmentation was manually supervised to ensure complete segmentation.

For human embryos, the segmentation was performed similarly with the following modifications. Fixed immune-stained embryos were imaged using a 40× oil immersion objective, and cells were identified as ICM on the basis of Nanog expression. ICM cells with high expression of Nanog or Gata6 were then manually identified and segmented on the basis of LaminB1 signal. For volume calculation in three-lineage differentiation experiments, Sox2-GFP-expressing hiPS cells were plated on glass-bottom 35-mm gridded bottom dish (Ibidi) to image the same colonies of cells across multiple timepoints. A complete z-stack was imaged using a Nikon eclipse Ti2 inverted microscope mounted with a CSU-W1 spinning disk microscope (60× oil immersion lens, numerical aperture (NA) 1.49) immediately before starting the differentiation protocol. The same colonies were imaged 8 h, 24 h and 48 h after the addition of differentiation media. Then, 3D segmentation of nuclei (outlined by Sox2-GFP expression) was performed using a custom Cellpose (v.2.2.2) model[76,77] to generate 3D masks. The 3D masks were converted to 3D ROIs using the 3D Suite plugin[78], after which nuclear volumes was quantified from 3D ROIs using the same plugin.

**Nuclear envelope fluctuations.** Images were corrected for bleaching (Bleach Correction function of ImageJ) and linear rotational drift (Stackreg; ImageJ). After corrections, the nuclear edge position was recorded as a function of time at different positions along the NE by drawing a line perpendicular to NE at multiple locations per nucleus. Fluctuations were calculated by measuring the standard deviation of the position of the NE from its mean position[46]. Actinomycin D (10 μM)/2-desoxyglucose (5 mM), Cytochalasin D (200 nM), Calyculin A (5 nM) and Nocadozole (400 nM; all from Sigma) were added where indicated. SPY650-FastAct (SpiroChrome) and Mem-Glow 488 (Cytoskeleteton) were used to visualize actin and the plasma membrane.

**CBX2 cluster analysis.** Images were obtained using Zeiss LSM980 plus or Airy Scan imaging mode and subsequent deconvolution. Individual cell nuclei were segmented from the images using custom Cellpose (v.2.2.2) model[76,77]. CBX2 clusters were detected with Cellpose using a custom-trained model. The number of aggregates per cells were counted using ImageJ. For localization at the nuclear periphery nuclei Cellpose masks were dilated 400 nm isotropically and eroded 20 times for 150 nm to generate thin consecutive masks using ImageJ. The thin masks were used to measure spatially resolved mean intensities of CBX2 and DAPI from the nuclear periphery from background-subtracted images. Each measurement was then divided by the mean total nuclear intensity of CBX2.

For condensate detection, live imaging videos were preprocessed with the Noise2Void noise reduction algorithm[79] and segmented using a Cellpose custom-trained model. The nuclear outlines were tracked on the basis of the overall CBX2 nuclear signal using TrackMate Cellpose adaptation[80]. For relative aggregation, CBX2 unprocessed videos were processed with ImageJ bleach correction plugin using the histogram matching algorithm. Relative aggregation was calculated for each tracked cell from background-subtracted videos by measuring the summed grey values within the condensate area and dividing them by the summed grey values of the entire nuclear area.

**Quantification of nuclear intensities of transcription factors.** Mean intensities were measured from images using a nuclear mask generated from LaminB1-RFP, DAPI or Sox2-GFP using ImageJ[74]. The N/C ratio for YAP staining was calculated for each cell by dividing the nuclear mean intensity by the cytoplasmic mean intensity.

### Mechano-osmotic perturbations

Compression was performed with a modified version of a previously published cell confiner system[81]. In brief, suction-cup-bound coverslips were applied to 2D iPS cell colonies using a controlled pressure pump (Elveflow). Compression height was controlled by 5-μm polystyrene bead spacers.

For the western blotting, phosphoproteomics and sequencing experiments cells were axially compressed using polystyrene block custom manufactured to fit 6-cm or 10-cm cell culture dishes. The blocks were preheated to 37 °C and placed on cells, ensuring a homogeneous layer of culture medium between the block and the cell layer.

Hyperosmotic shock was induced using 0.5 M sucrose in media. For CBX2 live imaging in hypertonic conditions, hypertonic buffer was diluted 1:20. Hypo-osmotic shock was induced using preheated mQH$_2$O that was added to the medium to achieve a final 1:4 mixture.

## AFM

AFM force spectroscopy measurements were performed using JPK NanoWizard 2 (Bruker Nano) mounted on a Nikon Eclipse Ti inverted microscope and operated with JPK SPM Control Software v5. Measurements were performed at 37 °C. Triangular non-conductive silicon nitride cantilevers (MLCT, Bruker) with a nominal spring constant of 0.01 N m$^{-1}$ were used for force spectroscopy measurements with the probe tip positioned directly over the nucleus. For all indentations, forces of up to 3 nN were applied and the velocity of indentation was kept constant at 2 µm s$^{-1}$. Force maps of 4 µm$^2$ with resolution 2 × 2 pixels were used to perform technical replicates of nuclei indentation, and all valid curves were analysed. Before fitting the Hertz model to obtain Young's Modulus (Poisson's ratio of 0.5), the offset was removed from the baseline signal, the contact point was identified and cantilever bending was subtracted from each force curve. All analysis was performed with JPK Data Processing Software (Bruker).

## nucGEMS

hiPS cells were transfected using X-tremeGENE (Sigma) according to the manufacturer's instructions. Colonies were imaged 36 h after transduction on a Nikon Eclipse Ti Eclipse microscope mounted with a Yokogawa CSU-W1 spinning disk unit using 405-nm and 488-nm lasers, a 63X CFI Apo 60×/NA 1.49/0.12 total internal reflection fluorescence microscopy objective and an ET525/36 m emission filter (Chroma). Images were acquired using 80% power from single focal plane at 100-ms intervals, 1 × 1 binning and 16-bit pixel depth. The High-Throughput Image Processing Software platform[82] was used to track particles, and custom Python pipelines were used to calculate aggregate mean square displacement, effective diffusivity and the diffusive exponent[83]. Single-particle tracks were generated with the Napari GEMspa plugin[84].

## Multiome sequencing and analysis

Cells were compressed as described above, after which single nuclei were processed using the Chromium Next GEM Single Cell Multiome ATAC + Gene Expression kit. Library preparation were performed according to the manufacturer's protocol. Two biological replicates were prepared for each condition and sequenced using the Chromium Single Cell Multiome platform (10x Genomics). All replicates were quality controlled and analysed separately to ensure reproducibility, after which all conditions were pooled and analysed together, treating each cell in the total pool as a biological replicate. Initial transcript count and peak accessibility matrices were obtained with Cell Ranger Arc (v.1.1.2).

Raw counts were subsequently imported into Python (3.8) as AnnData (0.8.0) objects. Cells with more than 25% mitochondrial RNA content were removed. Doublet prediction on single-cell (sc)RNA-seq data was performed using scrublet (0.2.3). Raw scRNA-seq counts were normalized using scran (1.22.1) with Leiden clustering input at resolution 0.5. scRNA-seq and scATAC-seq data were integrated as MuData objects (0.2.2) using Muon (0.1.2). Raw scATAC-seq counts were filtered for noise, and subsequently Term Frequency–Inverse Document Frequency was transformed using muon.atac.pp.tfidf with a scale factor of 10,000.

scRNA-seq data were further processed using scanpy (1.8.2): for 2D embedding, the expression matrix was subset to the 2,000 most highly variable genes (sc.pp.highly_variable_genes, flavour 'seurat'). The top 50 principal components were calculated and served as the basis for $k$-nearest-neighbour calculations (sc.pp.neighbors, n_neighbors = 30), which were used as input for uniform manifold approximation and projection (UMAP; https://doi.org/10.48550/arXiv.1802.03426) layout (sc.tl.umap, min_dist = 0.3).

scATAC-seq data dimensionality was reduced to 50 components using muon.atac.tl.lsi, and embedded in a 2D UMAP (sc.tl.umap) on the basis of $k$-nearest-neighbour calculation (sc.pp.neighbors, n_neighbors = 20).

Gene signatures for cell cycle[85], apoptosis (BCL2L1, CASP9, CYCS, IL1A, PIK3CG, TNFRSF10D, FADD, BIRC3 and FAS), starvation (ATG5, ATG12, ULK2, GABARAPL1, PRKAA2, BNIP3, TRIB3, DDIT3, HSPA5 and SERPINB3), hypoxia (HIF1A, HIF3A, EPAS1, ARNT, PCNA, ADM, CCND1, CA9 and GLUT1) and necrosis (BIRC3, FAS, DNM1L, GSDME, IPMK, MLKL, RBCK1, TICAM1 and YBX3) were scored using the scanpy 'sc.tl.score_genes' function. Pychromvar (0.0.4) was used to interrogate transcription-factor accessibility. Pseudobulk differential expression analysis was performed with pyDESeq2 (0.4.8). Active enhancers and gene regulatory networks were inferred using SCENIC+[50] (pyscenic 0.11.2) following documentation. In brief, topic modelling was performed with 16 topics after evaluating different Latent Dirichlet allocation models, and binarized using the Otsu method. Condition annotations were added to the metadata alongside topics. Accessibility matrices were imputed and normalized, and highly variable features and condition-specific accessibility were calculated. Custom motif and ranking databases were created using 'create_cisTarget_databases' based on the observed scATAC-seq peaks. Motifs were obtained from the SCENIC+ 'v10nr_clust_public' collection. After running SCENIC+, eRegulons underwent standard filtering and scoring.

## H3K27ac Cut&Tag and analysis

Cut&Tag was performed using Complete CUT&Tag-IT Assay Kit (Active Motif). Three biological replicates were collected from 10-cm dishes by brief incubation at 37 °C in 0.05 M EDTA in Hank's Balanced Salt Solution and attached to Concanavalin-A-conjugated magnetic beads (ConA beads, Polysciences) before overnight binding of primary antibody H3K27ac (Abcam, ab4729). Samples were then incubated with secondary antibody and CUT&Tag-IT Assembled pA-Tn5 Transposomes, after which tagmented DNA was purified, PCR amplified using a i5/i7 indexing primers, cleaned using SPRI beads and pooled into an equimolar library for sequencing on an Illumina HiSeq4000 sequencing platform.

Unmapped paired-end reads were trimmed to remove adapters and poor-quality sequences using fastp v.0.23.2 (--detect_adapter_for_pe). Paired-end reads were mapped to GRCh38 and *Drosophila melanogaster* (GCF_000001215.4_release_6_plus_iso1_mt) reference genomes using bwa-mem2 v.2.2.1 with default settings. To define active enhancers and promoters, we performed peak calling using the callpeak function ('-fBAMPE -ghs -q1e-5 --keep-dup all --nomodel') from MACS2 (2.2.7.1)[86]. PCR duplicates were removed using sambamba (v.1.0.1)[87], and the IgG-negative control from the CBX2 Cut&Run experiment was used for estimating background. The final set of H3K27ac-enriched genomic regions included a union of peaks from all experimental conditions that were detected in at least two biological replicates, with peaks closer than 1 kb merged together and peaks overlapping with the ENCODE blacklisted genomic regions discarded[88]. All H3K27ac peaks were further classified into active promoters ($n = 10,728$) and putative active enhancers ($n = 15,531$) based on the overlap with transcription start site (TSS) ± 1 kb regions of hg38 UCSC refGene and protein-coding GENCODE v38 genes. To link putative active enhancers with their potential target genes, we used the closest function from BEDTools (v.2.31.1)[89] to find the nearest UCSC refGene gene (TSS ± 1 kb region).

Following PCR duplicate removal, biological replicates (two replicates for pluripotency and hypertonic stress conditions, three for all others) from the same condition were spike-in normalized and merged together[90] and then used to generate coverage tracks using bamCoverage from deepTools (v.3.5.4)[91] ('-bs 1 –ignoreDuplicates'). BigwigCompare from deepTools ('--skipZeroOverZero --operation log2 --fixedStep -bs 50') was used to generate the differential H3K27ac enrichment tracks. To perform metaplot and heatmap analysis of the mean read density at regions of interest, we used computeMatrix and plotProfile/plotHeatmap from the deepTools suite.

Significant changes (p-adj <0.05, |FC| >1.5) in H3K27ac enrichment at active enhancers following different treatments were identified using the DESeq2 R package (v.1.42.1)[92,93]. In brief, multiBamSummary from deepTools (--outRawCounts) was used to obtain counts of hg38 mapped reads at target regions of interest across different conditions, following PCR duplicate removal. Read counts from the spike-in dm6 genome at dm6 refGene genes were used to calculate size factors for spike-in calibrated DESeq2 analysis[90]. To visualize an overlap of active enhancers decommissioned following compression (a union of peaks showing a significant loss of H3K27ac in the compressed pluripotency and basal conditions, $n = 3,694$) and hyperosmotic shock ($n = 5,496$), we used UpSetR (v.1.4.9) and VennDiagram (v.1.7.3) R packages. Gene Ontology (GO) term analyses were carried out using Enrichr[94].

### TTseq and analysis
Cells were labelled with 500 µM 4-thiouridine for the last 10 min of the indicated 30-min treatments, after which cells were collected and lysed in TRIzol. ERCC spike-ins (00043, 00170 and 00136) were added to lysates, after which RNA was isolated, fragmented and biotinylated with EZ-link HPDP-biotin (Thermo Fischer). Biotinylated nascent RNAs were purified with streptavidin-conjugated magnetic beads (µMACS; Miltenyi), and libraries of total and biotinylated RNA were prepared with Illumina TruSeq total RNA kit. Libraries were quantified using the KAPA Library Quantification Kit and sequenced with Illumina NextSeq 500 using the High Output Kit v.2.5 (150 cycles, Illumina) for 2 × 75-bp paired-end reads. Trimmed reads were aligned to the human genome assembly (hg38) using STAR2.4.2[95]. For coverage profiles and visualization, reads were uniquely mapped, destranded, antisense corrected and normalized with size factors calculated from DESeq2[92].

To quantify changes, spike-in calibrated differential analysis was performed[92]. DESeq2 with size factors estimated using ERCC spike-ins (00043, 00170 and 00136) was used for differential expression. Raw read counts following PCR removal were obtained for a set of hg38 MANE RefSeq genes (v.1.3)[10] using the featureCounts function ('isPairedEnd = TRUE, countReadPairs = TRUE, useMetaFeatures = TRUE, strandSpecific = 1') from the Rsubread package (v.2.16.1). log$_2$FCs following shrinkage using the original DESeq2 estimator[7] were visualized using custom R scripts and ggplot2.

### Bulk RNA-seq and analysis
Total RNA was isolated using the NucleoSpin RNA Plus kit (Macherey-Nagel). After quantification and quality control using Agilent 2200 TapeStation, total RNA amounts were adjusted and libraries were prepared using the TruSeq Stranded Total RNA kit with Ribo-zero gold rRNA depletion (Illumina). RNA-seq was carried out on Illumina NextSeq 500 using the High Output Kit v.2.5 (150 cycles, Illumina) for 2 × 75-bp paired-end reads.

Raw FASTQ files were adapter-trimmed and quality-filtered using fastp (0.23.2) using default settings and the '−detect_adapter_for_pe' flag. Filtered reads were mapped to the GRCh38 reference genome merged with ERCC92 spike-in sequence reference using STAR (2.7.10a) (--outSAMstrandField intronMotif --outFilterIntronMotifs RemoveNoncanonical --outFilterMultimapNmax 1 --winAnchorMultimapNmax 50 --outFilterType BySJout --alignSJoverhangMin 8 --alignSJDBoverhangMin 1 --outFilterMismatchNmax 999 --outFilterMismatchNoverReadLmax 0.04 --alignIntronMin 20 --alignIntronMax 1500000 --alignMatesGapMax 1500000 --chimSegmentMin 15 --chimOutType WithinBAM --outSAMattributes All). Gene expression counts were generated from mapped reads using Gencode v35 and ERCC92 annotations with htseq-count (2.0.3).

PCA and differential gene expression analysis were performed in R (4.1.2) using DESeq2 (1.34.0) assuming negative binomial distribution of read counts (Wald test). GO term and transcription factor binding analyses were carried out using Enrichr/ChEA3[94].

### Phosphoproteomics and analysis
hiPS cells in pluripotency media were subjected to compression or hypertonic shock for 5 min, after which cell lysates were collected in 6 M guanidinium chloride buffer supplemented with 5 mM Tris(2-carboxyethyl)phosphine, 10 mM chloroacetamide in 100 mM Tris-HCl. Samples were then boiled at 95 °C for 10 min, sonicated at high performance for 10 cycles (30 s on/off) using Bioruptor Plus Ultrasonicator (Diagenode) and spun down for 20 min at room temperature at 20,000g. Supernatants were then trypsin-gold digested (Promega Corp., V5280) overnight at 37 °C. Digested samples were acidified and peptides were cleaned with custom-packed C18-SD Stage Tips. Eluted peptides were vacuum dried at 30 °C, and phosphopeptides were enriched using the 3 mg/200 ml Titansphere Phos-TiO kit (GL Sciences) according to the manufacturer's instructions. After elution, phosphopeptides were vacuum dehydrated for 2 h at 30 °C and cleaned with custom-packed C18-SD. Ultimate 3000 ultra-high-performance liquid chromatography in conjunction with high-pH reversed-phase chromatography was used to separate and fractionate a total of 1 mg of tryptic peptides, from which eight fractions were collected.

All samples were analysed on a Q Exactive Plus Orbitrap mass spectrometer that was coupled to an EASY nLC (both Thermo Scientific). Peptides were loaded with solvent A (0.1% formic acid in water) onto an in-house packed analytical column (50 cm, 75 µm inner diameter, filled with 2.7-µm Poroshell EC120 C18, Agilent). Peptides were chromatographically separated at a constant flow rate of 250 nl min$^{-1}$ using the following gradient: 3–5% solvent B (0.1% formic acid in 80 % acetonitrile) within 1.0 min, 5–30% solvent B within 121.0 min, 30–40% solvent B within 19.0 min, 40–95% solvent B within 1.0 min, followed by washing and column equilibration. The mass spectrometer was operated in data-dependent acquisition mode. The MS1 survey scan was acquired from 300–1,750 $m/z$ at a resolution of 70,000. The top ten most abundant peptides were isolated within a 1.8-Th window and subjected to Higher-energy collisional dissociation (HCD) fragmentation at a normalized collision energy of 27%. The AGC target was set to $5 × 10^5$ charges, allowing a maximum injection time of 55 ms. Product ions were detected in the Orbitrap at a resolution of 17,500. Precursors were dynamically excluded for 25.0 s.

Raw data were processed with MaxQuant (v.2.4.0)[96] using default parameters against the Uniprot canonical Human database (UP5640, downloaded 4 January 2023) with the match-between-runs option enabled between replicates and phosphorylation at S, T and Y defined as variable modifications. Results were loaded into Perseus (v.1.6.15)[96], and contaminations and insecure identifications were removed. Remaining identified phosphosites were filtered for data completeness in at least one replicate group, and remaining data were normalized by median subtraction for each individual samples. After sigma-downshift imputation using standard settings, significantly changed sites were identified using both one-way analysis of variance (ANOVA) and false discovery rate-controlled $t$-tests and annotated using PhosphoSitePlus[97]. One-dimensional enrichments were performed on site abundance differences. ANOVA-significant site abundances were $z$-scored and hierarchically clustered using standard settings. GO term analyses were carried out using Enrichr[94].

### Western blotting
Cells were rinsed in PBS, suspended in lysis buffer (50 mM Tris-HCl buffer (pH 8.0), containing 150 mM NaCl, 1% Triton X-100, 0.05% sodium deoxycholate, 10 mM EDTA, protease and phosphatase inhibitors) and cleared by centrifugation. The lysates were then reduced in Laemmli sample buffer at 95 °C, separated by polyacrylamide gel electrophoresis in the presence of SDS and transferred onto polyvinylidene fluoride membranes. Membranes were blocked with 5% milk powder in Tris-buffered saline containing 0.05% Tween (TBS-Tween) for 1 h at room temperature, after which primary antibodies were added in 5% BSA, TBS-Tween and incubated overnight at 4 °C. The membranes were

subsequently washed in TBS-Tween, after which secondary horseradish peroxidase-conjugated antibodies (Bio-Rad) were added in 5% milk powder in TBS-Tween and incubated for 30 min at room temperature. Antibody binding was detected by chemiluminescence (Immobilon Western, Millipore) using the Bio-Rad ChemiDoc Imaging System. Antibodies used: LaminB1 (Cell Signaling 9087; 1:1,000), p38 MAPK phosphoThr180/Tyr182 (Cell Signaling 4511, 1:1,000), p38 MAPK (Cell Signaling 9212, 1:1,000), p44/42 MAPK phosphoErk1/2 (Cell Signaling 4376, 1:2,000), p44/42 MAPK Erk1/2 (Cell Signaling 4695, 1:1,000) and RNApol2 PS2 (Abcam ab5095; 1:5,000).

### RT–qPCR
Cells were treated as indicated with media conditions, compression, YAP inhibitor verteporfin (200 nM) and ERK inhibitor (1 μM). RNA was isolated using the NucleoSpin RNA plus Kit (Macherey-Nagel). RNA quality was assessed using the high-sensitivity RNA Screen Tape Analysis (TapeStation, Agilent), and subsequently, 500 ng of RNA was reverse transcribed applying the SuperScript IV VILO Master Mix (Thermo Fisher Scientific) following the manufacturer's instructions. External RNA Controls Consortium RNA Spike-in Mix (Thermo Fisher Scientific) was added to samples as reference to ensure validity of normalization. PCR was performed with PowerUp SYBR Green Mastermix (Thermo Fisher Scientific) using QuantStudio5 Real-Time PCR System (Thermo Fisher Scientific). Gene expression was quantified using the $\Delta\Delta$Ct method with normalization to LaminB1. For primers, see Supplementary Table 7.

### CBX2 Cut&Run and analyses
Cells were compressed from 30 min in basal or pluripotency medium followed by 24 h recovery in the same media condition. Uncompressed cells in the respective media were used as controls (two biological replicates per condition). CBX2 cut&run was performed using the CUTANA ChIC Cut&Run Kit (EpiCypher) using the manufacturers' instructions. In brief, 50,000,000 cells per replicate were collected and absorbed with activated ConA beads, followed by incubation with 0.5 μg of CBX2 antibody (mAb #25069, Cell Signaling) overnight at 4 °C. Chromatin was then digested and released, followed by DNA purification and library preparation using the CUTANA Cut&Run Library Prep Kit (EpiCypher). After library quantification and quality control using Agilent 2200 TapeStation, DNA sequencing was carried out on Illumina NextSeq 500 using the High Output Kit v2.5 (150 cycles, Illumina) for 2×75-bp paired-end reads.

Raw FASTQ files were adapter-trimmed and quality-filtered using fastp (0.23.2) using default settings and the '−detect_adapter_for_pe' flag. Filtered reads were mapped to the GRCh38 and *Escherichia coli* (ASM886v2) reference genome using bwa-mem2 (2.2.1). Mapped reads were quality-filtered using a cut-off of 3. Scaling-factor normalized tracks were generated according to Zheng et al. (https://doi.org/10.17504/protocols.io.bjk2kkye). Regions were blacklisted using published CUT&RUN blacklists[98].

Coverage profiles for IgG and CBX2 were generated using MACS2 (2.2.9.1). Normalization to IgG was performed using MACS2 'bdgcmp' and the 'qpois' method, and peak calls were obtained using MACS2 'bdgbroadcall' (-c 5 -C 2 -g 1000 -G 2000 --no-trackline). Peaks were filtered for noise using reproducibility (called in >1 sample of the same condition), minimum peak size (>1,000 bp), and minimum mean CBX2 signal in one condition (>0.075).

Aggregate statistics for each sample were calculated on replicate-merged tracks for peaks overlapping known genes (Gencode v44). Mean peak signal was scaled to the pluripotency medium as reference condition, and $\log_2$-transformed for symmetry. Hierarchical clustering was performed using the Ward method and Euclidian distance metric (scipy 1.11.4); clusters were chosen with a tree distance cut-off of 7.

### CRISPRi-mediated silencing of CBX2
One microgram of Alt-R CRISPR sgRNA (IDT; 5′-CCCGGCAGCCAGCCC GACCG) targeting the CBX2 promoter was mixed with $1 \times 10^5$

WTC-dCas9-TagBFP-KRAB hiPS cells (Coriell #AICS-0090-391) and electroporated using the Neon Transfection System (Thermo Fisher Scientific), using a regime of 1,400 V, 10 ms and two pulses. After electroporation, hiPS cells were cultured in Matrigel coated plate with mTeSR medium plus 25 μM Y-27632 ROCK inhibitor (Tocris #1254). After 24 h, the medium was changed to mTeSR without Y-27632 with fresh medium change daily. Ninety-six hours after sgRNA delivery, a subset of cells was fixed and immunostained for CBX2 to assess silencing efficiency. The remaining cells were transferred to E6 basal medium and cultured for an additional 96 h with daily medium changes.

### Statistics and reproducibility
Statistical analyses were performed using GraphPad Prism software (GraphPad, v.9) or in R (v.4.2.2). Statistical significance was determined by the specific tests indicated in the corresponding figure legends. Only two-tailed tests were used. In all cases where a test for normally distributed data was used, normal distribution was confirmed with the Kolmogorov–Smirnov test ($\alpha = 0.05$). All experiments presented in the Article were repeated in at least three independent replicates. No statistical method was used to predetermine sample size. No data were excluded from the analyses. The experiments were not randomized as groups were assigned by treatments. As data analyses where automated, investigators were not blinded for outcome assessment.

### Reporting summary
Further information on research design is available in the Nature Portfolio Reporting Summary linked to this article.

## Data availability
Sequencing datasets are available at GEO (GSE268091). Proteomic datasets are available at PRIDE (PXD052588). All raw data that support the conclusions are available from the authors on request. Source data are provided with this paper.

## Code availability
All analysis scripts and custom code are available at WickstromLab via GitHub at https://github.com/WickstromLab.

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

## Acknowledgements

We thank T. Misteli and I. Bedzhov for critical reading of the manuscript, N. Jabado for advice on CBX2, M. Geis and H. Lee for help with segmentation, H. vom Bruch, C. Ortmeier and S. Heising for expert technical assistance, F. Bertillot for advice on image analyses, the Max Planck Institute BioOptics and Sequencing core facilities and the CCR/NCI High-Throughout Imaging Facility, for support with experiments. Human embryos were imaged at the Biomedicum Helsinki Imaging Unit supported by HiLIFE. Multiome sequencing was performed at the Institute for Molecular Medicine Finland FIMM Genomics unit supported by HiLIFE and Biocenter Finland. This work was supported by Instrumentarium Science Foundation and Academy of Finland fellowships (to A.S.), the Intramural Research Program of the National Institute of Diabetes (NIH) and Digestive and Kidney Diseases (NIDDK; to Y.A.M.), the Sigrid Juselius Foundation (to S.V. and S.A.W.), Helsinki Institute of Life Science (to S.V. and S.A.W.) and Academy of Finland Research Fellowship (353549 to S.V.), Academy of Finland Center of Excellence BarrierForce and R'Life Programme consortium NucleoMech (to S.A.W.), German Research Foundation (DFG) FOR 5504 and the Max Planck Society (to S.A.W.).

## Author contributions

K.P.M. and A.S. designed and performed experiments and analysed data. R.S. and A.C. performed and analysed differentiation experiments. K.K. performed and supervised sequencing data analysis. J.-W.L. performed and analysed proteomics experiments. A.M. performed experiments. C.M.N. provided conceptual advice and supervised the proteomics analyses. N.A.F. performed TTseq and H3K27Ac Cut&Tag analyses. L.C.B. supported sequencing experiments. C.H.-G. collected, warmed and cultured the human embryos. J.Z. designed and constructed Halo-tagged lines with Y.A.M. A.K. aided in nucGEM mean square displacement analyses. S.N., R.P. and S.V. designed and performed all studies on human embryos and blastoids. YAM supervised the study, performed all sequencing experiments, designed and performed experiments and analysed data. S.A.W. conceived and supervised the study, designed experiments, analysed data and wrote the paper. All authors commented and edited the manuscript.

## Funding

## Competing interests

The authors declare no competing interests.

## Additional information

**Extended data** is available for this paper at https://doi.org/10.1038/s41556-025-01767-x.

**Correspondence and requests for materials** should be addressed to Yekaterina A. Miroshnikova or Sara A. Wickström.

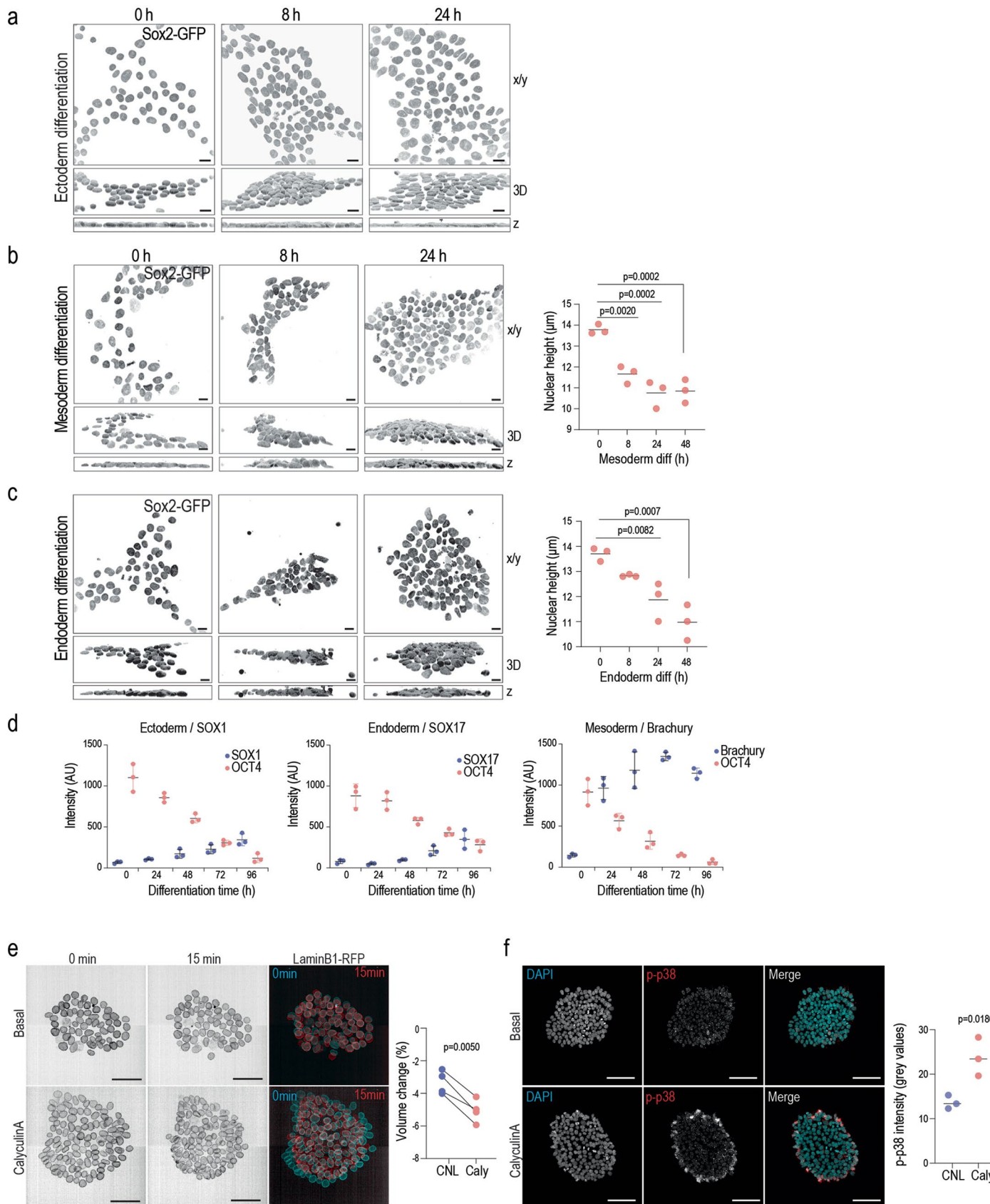

**Extended Data Fig. 1 | See next page for caption.**

**Extended Data Fig. 1 | Mechano-osmotic changes of the nucleus accompany exit from pluripotency.** (a-c) Representative top views (x-y), 3D reconstructions and cross sections (z), and quantification of nuclear height from of Sox2-GFP-tagged hiPSCs undergoing ectodermal (**a**), mesodermal (**b**), or endodermal (**c**) differentiation for the indicated time points (scale bars 15 μm; n = 3 independent experiments with 348, 496, 841, 765 (mesoderm 0, 8, 24, 48 h, respectively); 435, 661, 800, 835 (ectoderm 0, 8, 24, 48 h, respectively) total cells/ condition; ANOVA/ Dunnett's). (**d**) Quantification of lineage marker intensity from immunofluorescence stainings in hiPSCs undergoing ectodermal (left), endodermal (middle), or mesodermal (right) differentiation for the indicated time points (scale bars 15 μm; n = 3 independent experiments with 5093, 1257, 3913, 8290, 11345 (ectoderm SOX1); 1257, 3255, 5751, 8290, 11309 (ectoderm OCT4); 1362, 3967, 4680, 6542, 8450 (Endoderm OCT4); 1362, 3967, 4680, 6542, 6410 (endoderm SOX17); 841, 3089, 4897, 5368, 6189 (mesoderm Brachyury); 842, 3089, 4897, 5368, 6186 (mesoderm OCT4) total cells/0, 24, 48, 72, 96 h conditions, respectively; mean±SD). (**e**) Representative snapshots from live imaging data and quantification of nuclear volume from LaminB1-RFP hiPSCs treated with Calyculin A for 15 min. Note compaction of colony and nuclear deformation at colony edges (scale bars 50 μm; n = 4 independent experiments with 546 (DMSO) and 682 (Calyculin A) total nuclei/condition; Student's t-test). (**f**) Representative images and quantification of p-p38 from hiPSCs treated with Calyculin A for 15 min. Note increased p38 activation at colony edges (scale bars 100 μm; n = 3 independent experiments with 14 (DMSO) and 17 (Calyculin A) total colonies/condition; Student's t-test). Source numerical data are available in source data.

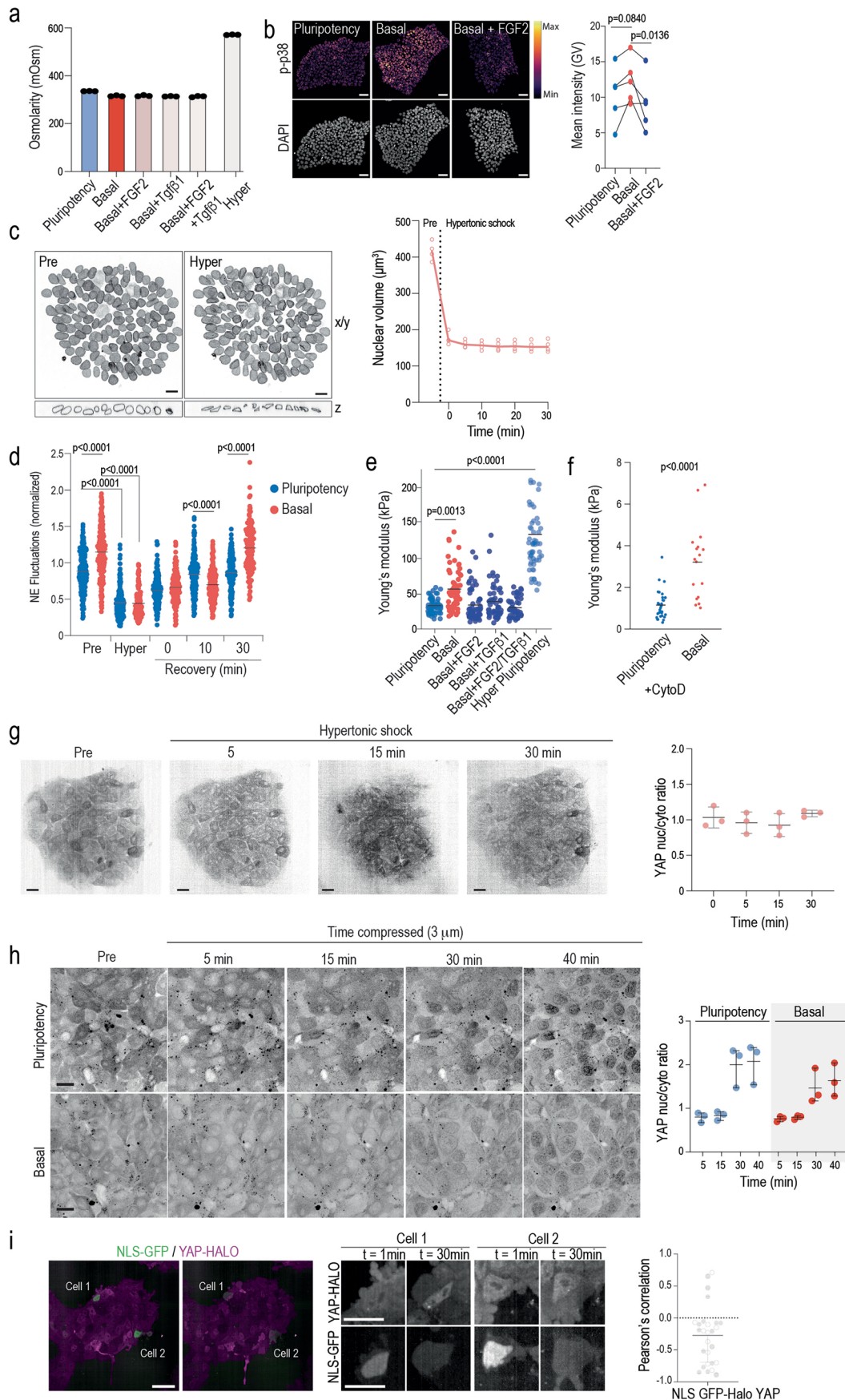

**Extended Data Fig. 2 | See next page for caption.**

**Extended Data Fig. 2 | Mechanical signals control YAP activity while osmotic signals control nuclear rheology.** (**a**) Osmolarity measurements of the various media conditions (mean ± SD; n = 3 independent measurements). (**b**) Representative images and quantification of p-p38 from hiPSCs 5 min after exchanging pluripotency maintenance medium with pluripotency, basal or basal + FGF2 medium for 5 min. Note moderate but consistent activation of p38 activation in cells in basal but not in basal + FGF2 medium (scale bars 50 μm; n = 5 independent experiments; RM-ANOVA/ Fischer's). (**c**) Representative snapshots of live imaging (x/y), optical cross sections (z), and quantifications from LaminB1-RFP-tagged hiPSCs before (Pre) and after (Hyper) hypertonic shock. Note progressive decline in nuclear volume. Line represents median volume and individual dots are average colony volumes at indicated timepoints (scale bars 10 μm; n = 3 independent experiments with 80, 166, 137 nuclei/experiment tracked over the time; Mann-Whitney). (**d**) Quantification of nuclear envelope fluctuations in cells exposed to hypertonic shock and subsequent washout in pluripotency or basal medium for time points indicated. Note attenuations of fluctuations upon hypertonic shock in both conditions and recovery to more abundant fluctuations in basal medium (n = 428 (Pluri Baseline), 331 (Pluri Hyper 30), 338 (Pluri Rec 0), 400 (Pluri Rec 10), 372 (Pluri Rec 30), 466 (Basal Baseline), 307 (Basal Hyper 30), 392 (Basal Rec 0), 424 (Basal Rec 10), 317 (Basal Rec 30) cells pooled across 3 independent experiments; Kruskal-Wallis/Dunn's). (**e**) AFM

force indentation experiments of iPS cell nuclei within 20 min of media switch or hypertonic shock. Note data is reproduced from Fig. 1m but with additional condition of hypertonic shock (n = 69 (Pluripotency), 71 (Basal), 76 (Basal+FGF2), 85 (Basal+TGF-β1), 74 (Basal+FGF + TGF-β1), 60 (Hyper Pluripotency) nuclei pooled across 5 independent experiments; ANOVA/Kruskal-Wallis). (**f**) AFM force indentation experiments of iPS cell nuclei treated with Cytochalasin D in pluripotency or basal medium (n = 14 and 17 nuclei for Pluripotency and Basal conditions, respectively, pooled across 3 independent experiments). (**g**) Representative snapshots and quantification of live imaging of YAP-Halo-tag hiPSCs during hypertonic shock (scale bars 10 μm; n = 3 independent experiments with 132 total cells tracked across time). (**h**) Representative snapshots of live imaging and quantification from YAP-Halo-tag hiPSCs during 3 μm compression in pluripotency or basal medium. Note comparable activation of YAP in both conditions (scale bars 30 μm; n = 3 independent experiments representing 255 (Basal), 201 (Pluripotency) cells/condition tracked across time). (**i**) Representative snapshots of live imaging and quantification from YAP-Halo-tag hiPSCs transfected with NLS-EGFP and compressed to 3 μm height. Note anticorrelated dynamics of YAP and EGFP where YAP nuclear localization is enhanced upon compression whereas EGFP is not (scale bars 50 μm (left panel), 30 μm (right panel); n = 31 cells pooled across 3 independent experiments). All error bars mean±SD. Source numerical data are available in source data.

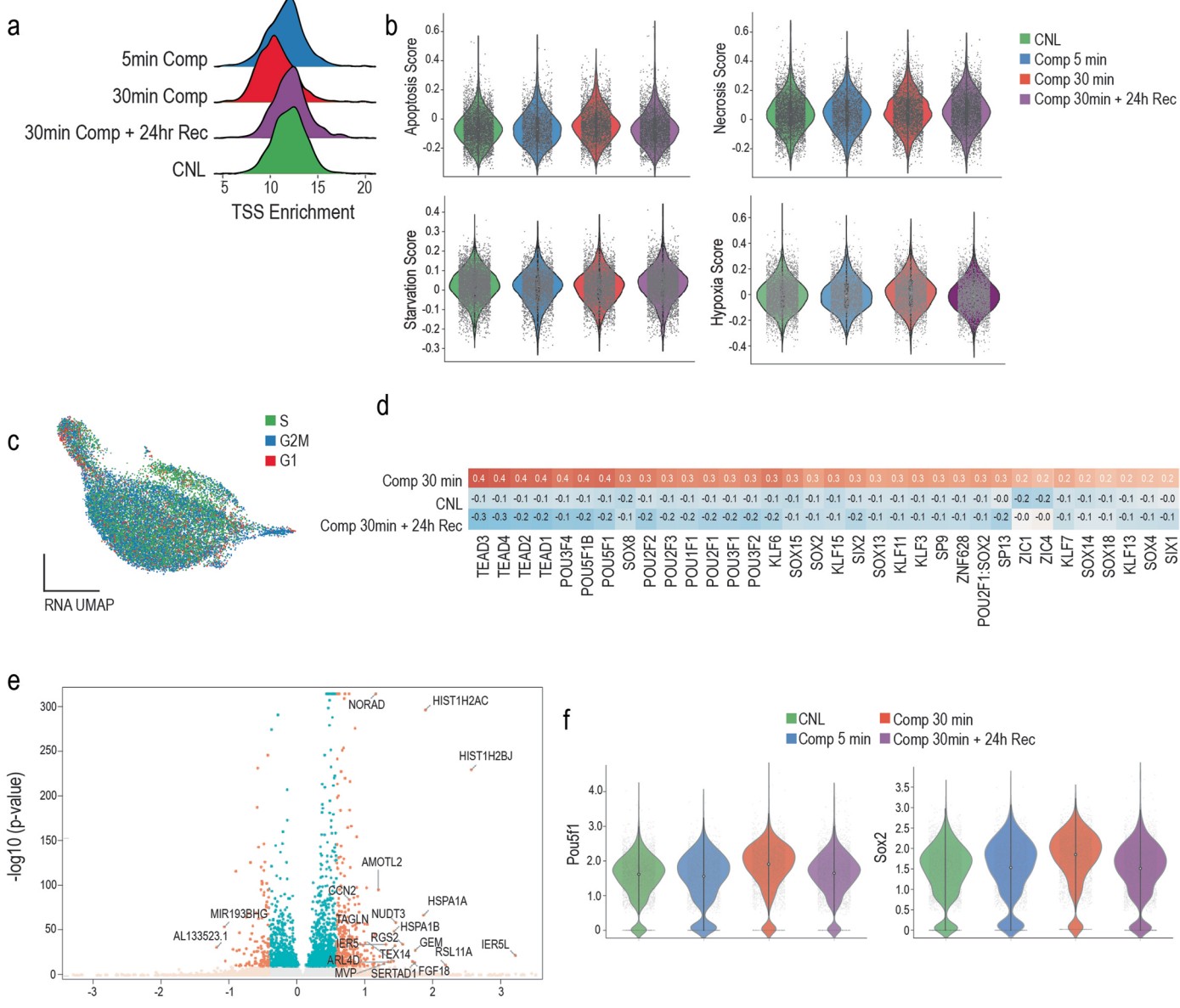

**Extended Data Fig. 3 | Analyses of transcriptional and chromatin changes in single cells in response to mechano-osmotic stimuli. (a)** Quantification of open chromatin from scATACseq of hiPSCs compressed (Comp) for 5 or 30 mins with or without 24 h recovery (Rec) as well as uncompressed controls (CNL). Note reduced accessibility at transcription start sites (TSS) at 30 min compression. **(b)** Violin plots of computed apoptosis, necrosis, starvation and hypoxia scores from single cell RNAseq across all conditions. **(c)** UMAP of cell cycle stage distribution from single cell RNAseq across all conditions. **(d)** Heatmap of ChromVAR analysis from scATACseq of hiPSCs compressed for 30 mins with or without 24 h recovery. **(e)** Volcano plot of differential gene expression between 30 min compression and CNL. **(f)** Violin plots of *POU5F1* and *SOX2* gene expression across all conditions (Wald/IHW).

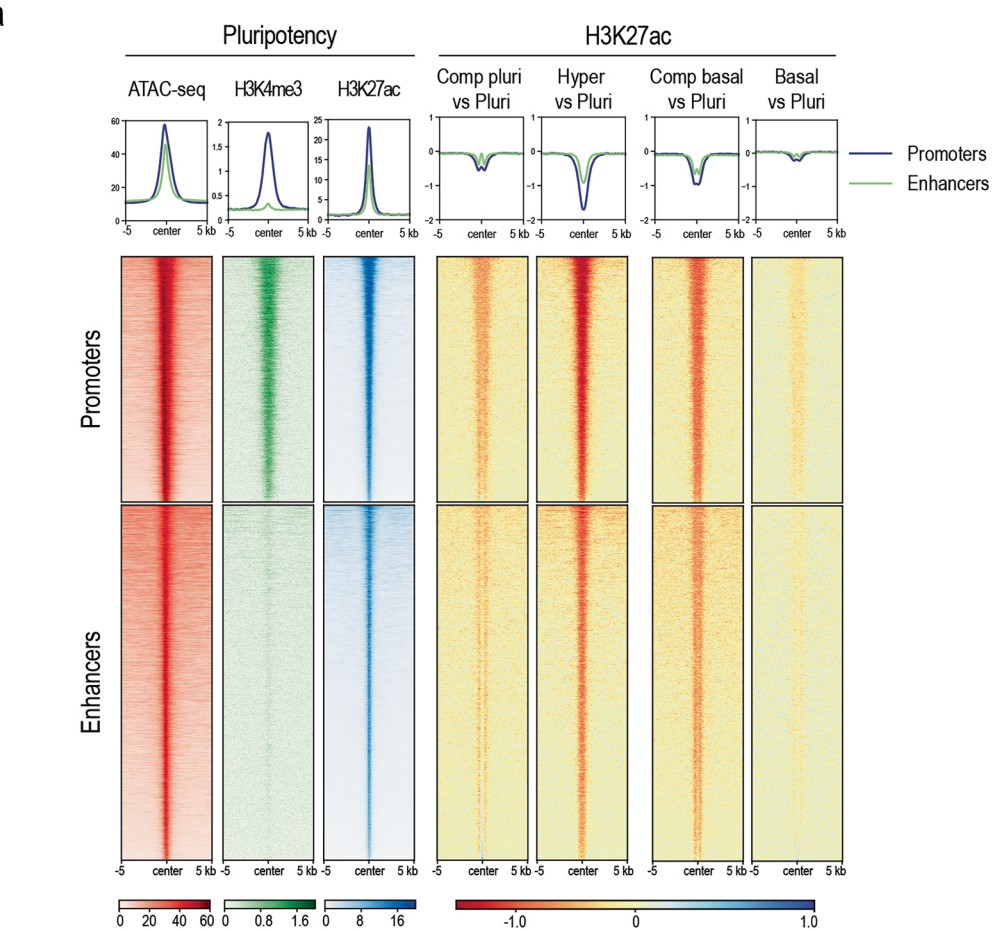

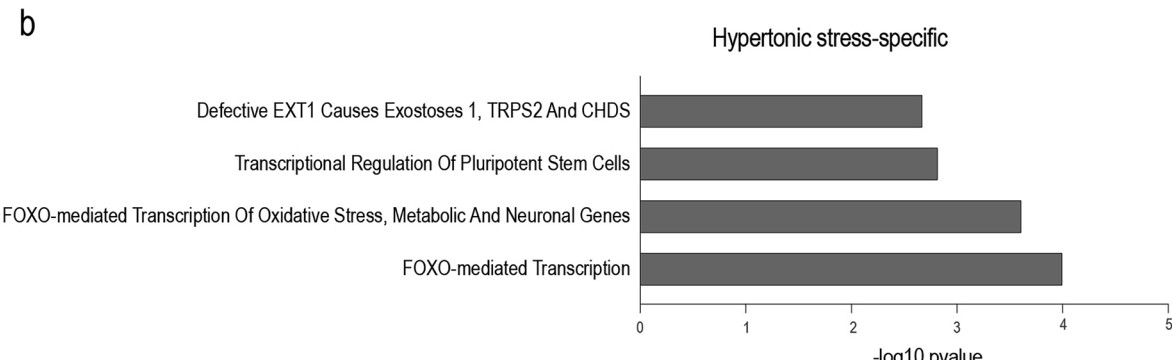

**Extended Data Fig. 4 | Changes in enhancer and promoter activity in response to mechano-osmotic stimuli. (a)** Heatmaps of pseudo-bulk ATAC-seq, H3K4me3 and H3K27ac Cut & Tag at annotated H3K27ac peaks, further classified into active promoters and putative active enhancers, illustrating expected enrichment of histone modifications and chromatin accessibility at promoters and enhancers.

Log2 fold change (FC) heatmaps highlight H3K27ac changes at active promoters and enhancers across different conditions compared to the pluripotency medium condition. **(b)** Reactome pathway enrichment of decommissioned enhancers specific to hypertonic shock (Fisher's exact /Benjamini-Hochberg). Source numerical data are available in source data.

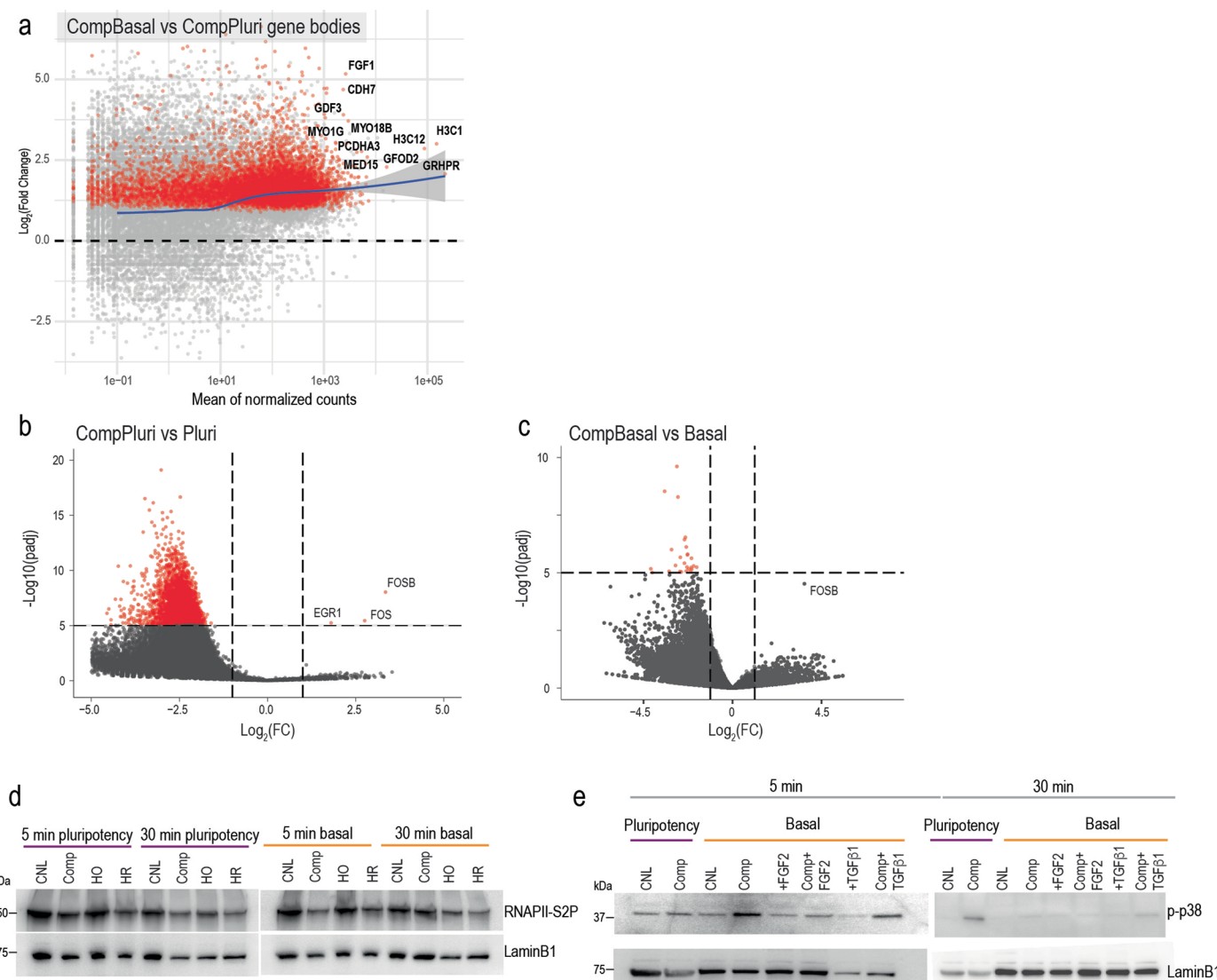

**Extended Data Fig. 5 | Changes in transcription in response to mechano-osmotic stimuli.** (**a**) Spike-in normalized MA plot contrasting CompBasal to CompPluri nascent transcript states. Loess local regression line in blue. Selected statistically significant upregulated genes are labeled. (**b, c**) Spike-in normalized volcano plot contrasting CompPluri to Pluri (**b**) and CompBasal to Basal (**d**) conditions (Wald/IHW). (**d**) Representative western blots from RNAPII-S2P from cells exposed to compression or hypertonic/hypotonic shocks in the media compositions and time points indicated. Note decrease in RNAPII-S2P at 30 min compression in pluripotency medium but already at 5 min in basal medium (n = 3 independent experiments). (**e**) Representative western blots from phosphorylated p38 (pp38) from cells exposed to compression in the media compositions and time points indicated. Note increase pf pp38 at 30 min compression in pluripotency medium but already at 5 min in basal medium and suppression of compression mediated activation of p38 by addition of FGF2 but not with TGF-β1 into basal medium (n = 3 independent experiments). Hypertonic (HR) and hypotonic (HO) shocks are used as controls for **c**, **d**. Unprocessed blots are available in source data.

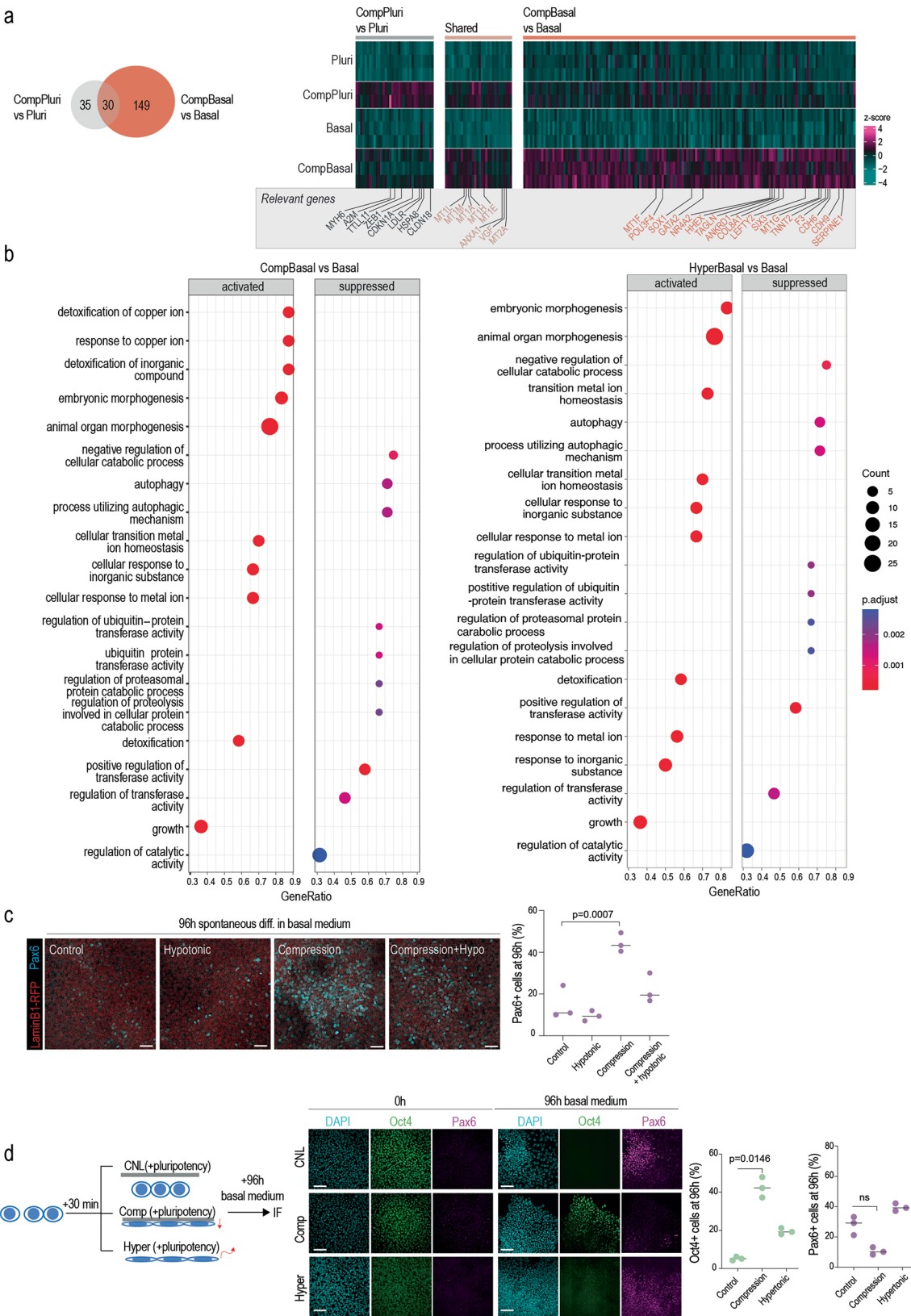

**Extended Data Fig. 6 | See next page for caption.**

**Extended Data Fig. 6 | Long-term transcriptional changes in response to mechano-osmotic stimuli.** (**a**) Venn diagram and Z-score heatmap analyses of specific and overlapping genes from bulk RNA sequencing in 24 h upregulated in response to compression in basal medium or pluripotency medium when compared to their respective media conditions. Note that compression in basal medium leads to more genes being upregulated and the specific genes representing differentiation genes. Compression in pluripotency medium leads to upregulation of cytoskeletal and stress genes. (**b**) Gene set enrichment analyses of differentially expressed genes from bulk RNA sequencing in 24 h basal medium versus 30 min compression + 24 h recovery in basal medium (left panel) and 24 h basal medium versus 30 min hypertonic shock + 24 h recovery in basal medium (right panel). Note enrichment of gene sets involved in morphogenesis and metal ion homeostasis in both conditions. (**c**) Representative images of Pax6 staining and quantification of LaminB1-RFP hiPSCs exposed to 30 min basal medium (Control), 30 min basal medium + 10 min hypotonic shock (Hypotonic), 30 min compression (Compression) and 30 min compression +10 min hypotonic shock (Compression+Hypo) in basal medium followed by 96 h of spontaneous differentiation in basal medium. Note delayed differentiation in both shock conditions (scale bars 50 μm; n = 3 independent experiments with 33844 (Basal), 34959 (Hypo), 35860 (Comp), 27615 (Comp+Hypo) total cells/condition; ANOVA/ Dunnett's). (**d**) Schematic representation of experimental outline, representative images of Oct4 and Pax6 staining and quantification of cells exposed to 30 min compression or 30 min hypertonic shock in pluripotency medium for 96 h. Note delayed differentiation in both shock conditions (scale bars 75 μm; n = 3 independent experiments with 673 (control), 564 (Compression), 421, 832 (Hypertonic) total cells/condition; Kruskal-Wallis/Dunn's, ns=not significant). Source numerical data are available in source data.

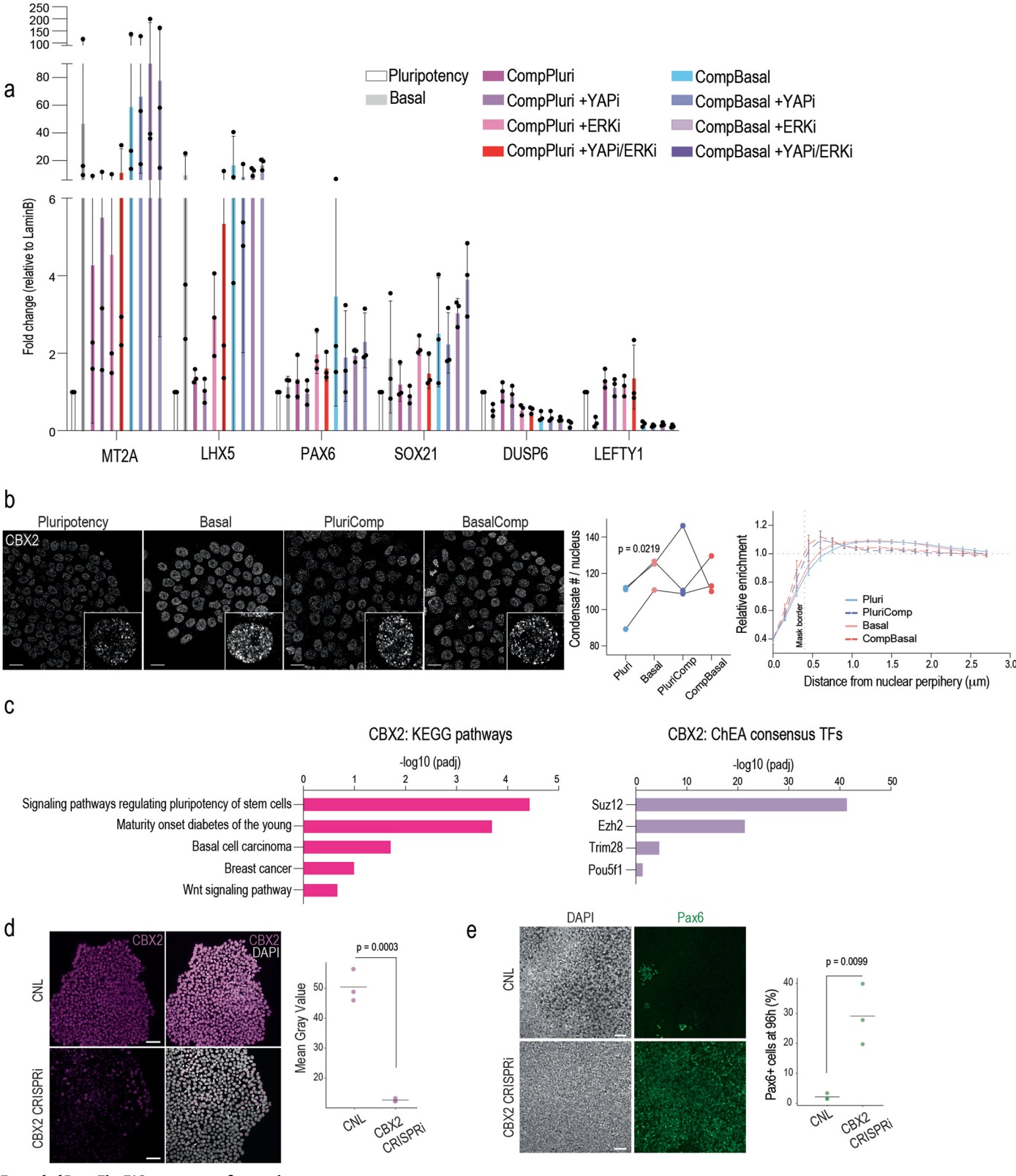

**Extended Data Fig. 7 | See next page for caption.**

**Extended Data Fig. 7 | Analyses of CBX2 condensation dynamics and its role in pluripotency exit.** (**a**) RT-qPCR from cells exposed to 30 min compression in indicated media conditions followed by 24 h recovery in the same media conditions with and without YAP and ERK inhibitors as indicated. Note increased metallothionine (MT2A) and differentiation gene (LHX5, PAX6, SOX21) expression in basal medium and 30 min compression in basal medium conditions (CompBasal) and lack of rescue with YAP/ERK inhibition. Cells compressed in pluripotency medium (CompPluri) show increased differentiation gene expression upon inhibition of YAP/ERK (n = 3 independent experiments; mean ±SD). (**b**) Representative images and quantification of CBX2 clustering at the nuclear periphery (scale bars 5 μm; left graph: n = 3 independent experiments with 1450 (Pluripotency), 1091 (Basal), 1181 (PluriComp), 1093 (BasalComp) total nuclei/condition; Student's t-test; right graph: n = 5 independent

experiments with 100 nuclei/condition; mean ±SD). (**c**) KEGG pathway analysis (left) and CheEA consensus Transcription Factor (TF) prediction (right) from CBX2 peaks differentially occupied in basal/compression in basal medium conditions (Fisher's exact /Benjamini-Hochberg). (**d**) Representative images and quantification of CBX2 levels after CRISPRi depletion (scale bars 100 μm; n = 3 independent experiments with 6539, 5597 total cells for CNL and CBX2 CRISPRi conditions, respectively; Student's t-test). (**e**) Representative images and quantification of Pax6 levels after CBX2 CRISPRi depletion and 96 h of spontaneous differentiation in Basal medium (scale bars 100 μm; n = 3 independent experiments with 21562, 25843 total cells per CNL, CBX2 CRISPRi conditions, respectively; Student's t-test). Source numerical data are available in source data.

# Reporting Summary

## Statistics

For all statistical analyses, confirm that the following items are present in the figure legend, table legend, main text, or Methods section.

| n/a | Confirmed | |
|---|---|---|
| ☐ | ☒ | The exact sample size ($n$) for each experimental group/condition, given as a discrete number and unit of measurement |
| ☐ | ☒ | A statement on whether measurements were taken from distinct samples or whether the same sample was measured repeatedly |
| ☐ | ☒ | The statistical test(s) used AND whether they are one- or two-sided *Only common tests should be described solely by name; describe more complex techniques in the Methods section.* |
| ☒ | ☐ | A description of all covariates tested |
| ☐ | ☒ | A description of any assumptions or corrections, such as tests of normality and adjustment for multiple comparisons |
| ☐ | ☒ | A full description of the statistical parameters including central tendency (e.g. means) or other basic estimates (e.g. regression coefficient) AND variation (e.g. standard deviation) or associated estimates of uncertainty (e.g. confidence intervals) |
| ☐ | ☒ | For null hypothesis testing, the test statistic (e.g. $F$, $t$, $r$) with confidence intervals, effect sizes, degrees of freedom and $P$ value noted *Give P values as exact values whenever suitable.* |
| ☒ | ☐ | For Bayesian analysis, information on the choice of priors and Markov chain Monte Carlo settings |
| ☒ | ☐ | For hierarchical and complex designs, identification of the appropriate level for tests and full reporting of outcomes |
| ☒ | ☐ | Estimates of effect sizes (e.g. Cohen's $d$, Pearson's $r$), indicating how they were calculated |

*Our web collection on statistics for biologists contains articles on many of the points above.*

## Software and code

Policy information about availability of computer code

| Data collection | Andor Fusion software (spinning disc confocal microscopy, version 2.3.0.44 ) |
|---|---|
| | Leica Application Suite X (confocal microscopy, version 2.0.0.14332) |
| | JPK SPM Control Software (version 5) |
| | Nikon Software (NIS-Elements AR 5.41.01) |
| | Zeiss ZEN Software (Zeiss ZEN version 3.7), |

| Data analysis | Zeiss ZEN Software (Zeiss ZEN version 3.7) |
|---|---|
| | Cellpose (version 2.2.2) |
| | Fiji (version 2.0.0) |
| | JPK Data Processing Software (Bruker Nano, version 5) |
| | Python (3.8) |
| | R (v4.2.2) |
| | Cell Ranger Arc (v.1.1.2) |
| | scanpy (v1.8.2) |
| | SCENIC+ (pyscenic v0.11.2) |
| | FastP (v0.23.2) |
| | bwa-mem2 (v.2.2.1) |
| | sambamba (v 1.0.1) |
| | deepTools (v 3.5.4) |
| | DESeq2 (v1.34.0) |
| | Perseus (v1.6.15) |

scipy (v1.11.4)
MaxQuant (v2.4.0)
Zeiss ZEN Software (Zeiss ZEN version 3.7)
GraphPad Prism software (GraphPad, version 9)

For manuscripts utilizing custom algorithms or software that are central to the research but not yet described in published literature, software must be made available to editors and reviewers. We strongly encourage code deposition in a community repository (e.g. GitHub). See the Nature Portfolio guidelines for submitting code & software for further information.

## Data

Policy information about availability of data

All manuscripts must include a data availability statement. This statement should provide the following information, where applicable:
- Accession codes, unique identifiers, or web links for publicly available datasets
- A description of any restrictions on data availability
- For clinical datasets or third party data, please ensure that the statement adheres to our policy

Sequencing datasets are available at GEO
Project accession: GSE26809:

Proteomic datasets are available at PRIDE
Project accession: PXD052588

Data will be freely available upon publication. All other data supporting the findings of this study are available from the corresponding author on reasonable request.

## Human research participants

Policy information about studies involving human research participants and Sex and Gender in Research.

| Reporting on sex and gender | N/A |
|---|---|
| Population characteristics | N/A |
| Recruitment | N/A |
| Ethics oversight | N/A |

Note that full information on the approval of the study protocol must also be provided in the manuscript.

# Field-specific reporting

Please select the one below that is the best fit for your research. If you are not sure, read the appropriate sections before making your selection.

☒ Life sciences          ☐ Behavioural & social sciences          ☐ Ecological, evolutionary & environmental sciences

For a reference copy of the document with all sections, see nature.com/documents/nr-reporting-summary-flat.pdf

# Life sciences study design

All studies must disclose on these points even when the disclosure is negative.

| Sample size | Sample size was determined based on previous experience, published literature, or to specific requirements of a given technique. Sample size for each experiment is indicated in figure legends |
|---|---|
| Data exclusions | Sequencing results were filtered by quality using fastp and files with low quality were excluded from subsequent analyses. |
| Replication | All experiments were performed using at least three biological replicates. Number of replicates for each experiment is indicated in the corresponding figure legend. Several steps were taken to ensure the reproducibility of experimental findings and key results were confirmed using complementary experimental approaches. |
| Randomization | Samples were not randomized, randomization was not relevant as samples were grouped according to treatment. |

| Blinding | Blinding was not used. This was not meaningfuly as automated software algorithms were used for unbiased quantification of staining intensities and sequencing data. |
| --- | --- |

# Reporting for specific materials, systems and methods

We require information from authors about some types of materials, experimental systems and methods used in many studies. Here, indicate whether each material, system or method listed is relevant to your study. If you are not sure if a list item applies to your research, read the appropriate section before selecting a response.

## Materials & experimental systems

| n/a | Involved in the study |
| --- | --- |
| ☐ | ☒ Antibodies |
| ☐ | ☒ Eukaryotic cell lines |
| ☒ | ☐ Palaeontology and archaeology |
| ☒ | ☐ Animals and other organisms |
| ☒ | ☐ Clinical data |
| ☒ | ☐ Dual use research of concern |

## Methods

| n/a | Involved in the study |
| --- | --- |
| ☒ | ☐ ChIP-seq |
| ☒ | ☐ Flow cytometry |
| ☒ | ☐ MRI-based neuroimaging |

## Antibodies

| Antibodies used | OCT3/4 (Santa-Cruz Biotechnology, sc-5279; 1:1000), Brachyury (R&D Systems, AF2085; 1:1000), GATA6 (AF1700, RnD Systems; 1:200), NANOG (D73G4, Cell Signaling Technologies; 1:200), LAMINB1 (66095-1-Ig, Proteintech; 1:200), LaminB1 (Cell Signaling 9087; 1:1000), Pax6 (Invitrogen, #42-6600; 1:1000), SOX1 (R&D Systems, AF3369; 1:200), SOX7 (R&D Systems, AF1924; 1:1000), YAP1 (Santa Cruz sc-101199; 1:200), CBX2 (Thermo Fisher PA-582812; 1:800), p38 MAPK phospho Thr180/Tyr182 (Cell signaling 4511, 1:1000), p38 MAPK (Cell Signaling 9212, 1:1000), p44/42 MAPK phospho-Erk1/2 (Cell Signaling 4376, 1:2000), p44/42 MAPK Erk1/2 (Cell Signaling 4695, 1:1000), RNApol2 PS2 (Abcam ab5095; 1:5000), H3K27ac (Abcam, ab4729). Alexa Fluor 488, 568, 594 and 647 conjugated antibodies (all from Invitrogen) were used as secondary antibodies at 1:500 dilution. |
| --- | --- |
| Validation | All antibodies are well characterized and widely used in the literature. They were applied according to datasheet instructions or previously published protocols with respect to staining and Western Blotting. |

## Eukaryotic cell lines

Policy information about cell lines and Sex and Gender in Research

| Cell line source(s) | Allen Institute LMNB1 mTagRFP (AICS-0034 cl.62); Allen Institute SOX2 mEGFP (AICS-0074 cl.26); dCas9-KRAB TagBFP (AICS-0090 cl.391); dCas9-TagBFP-KRAB Halotag-YAP1 knockin,  dCas9-TagBFP-KRAB CBX2-GFP knockin |
| --- | --- |
| Authentication | Cell lines from the Allen Institute were rigorously authenticated by the supplier. Halotag-YAP1 and CBX2-GFP insertions were authenticated by specific primers. |
| Mycoplasma contamination | Cell cultures were routinely confirmed mycoplasma-negative. |
| Commonly misidentified lines (See ICLAC register) | *Name any commonly misidentified cell lines used in the study and provide a rationale for their use.* |

