## [Peer Review File · Nature Cell Biology]

Mechano-osmotic signals control chromatin state and fate transitions in pluripotent stem cells

Corresponding Author: Professor Sara Wickstrom

Version 0:

Decision Letter:

Revise extended OD

*Please delete the link to your author homepage if you wish to forward this email to co-authors.

Dear Professor Wickström,

Your manuscript, "Mechano-osmotic signals control chromatin state and fate transitions in pluripotent stem cells", has now been seen by 3 referees, who are experts in mechanobiology in development (referee 1); nucleus mechanobiology (referee 2); and embryogenesis (referee 3). As you will see from their comments (attached below) they find this work of potential interest, but have raised substantial concerns, which in our view would need to be addressed with considerable revisions before we can consider publication in Nature Cell Biology.

Nature Cell Biology editors discuss the referee reports in detail within the editorial team, including the chief editor, to identify key referee points that should be addressed with priority, and requests that are overruled as being beyond the scope of the current study. To guide the scope of the revisions, I have listed these points below. We are committed to providing a fair and constructive peer-review process, so please feel free to contact me if you would like to discuss any of the referee comments further.

In particular, it would be essential to:

- A- Clarify the role of nuclear oscillations in cell fate determination (Reviewer#1)
- B- Further test whether hypertonic stress affects nuclear envelope fluctuations (Reviewer #1 paragraph starting with "The authors compare the effect of compression on YAP nuclear localization under two conditions: .. and reviewer#2 pt 1)
- C- Further investigate the function of CBX2 in the proposed model (Reviewer#1 paragraph starting with "The authors propose that mechano-osmotic forces...)
- D- Perform further experiments to determine the physiological relevance of the findings (Reviewer#1 paragraph starting with "The system used to address...), however please note we would not require experiments in a mouse model.

- All other referee concerns pertaining to strengthening existing data, providing controls, methodological details, clarifications of discrepancies between text and data presented in figures (Reviewer #2 pt 2, Reviewer#3 pts 6-10) and measurements made (Reviewer#2 pts 2, 6) and textual changes, should also be addressed.

- Finally please pay close attention to our guidelines on statistical and methodological reporting (listed below) as failure to do so may delay the reconsideration of the revised manuscript. In particular please provide:

We would be happy to consider a revised manuscript that would satisfactorily address these points, unless a similar paper is published elsewhere, or is accepted for publication in Nature Cell Biology in the meantime.

- ensure that it conforms to our format instructions and publication policies (see below and www.nature.com/nature/authors/).

- provide a point-by-point rebuttal to the full referee reports verbatim, as provided at the end of this letter.

- provide the completed Editorial Policy Checklist (found here <https://www.nature.com/authors/policies/Policy.pdf>), and Reporting Summary (found here <https://www.nature.com/authors/policies/ReportingSummary.pdf>). This is essential for reconsideration of the manuscript and these documents will be available to editors and referees in the event of peer review. For more information see <http://www.nature.com/authors/policies/availability.html> or contact me.

Nature Cell Biology is committed to improving transparency in authorship. As part of our efforts in this direction, we are now requesting that all authors identified as 'corresponding author' on published papers create and link their Open Researcher and Contributor Identifier (ORCID) with their account on the Manuscript Tracking System (MTS), prior to acceptance. ORCID helps the scientific community achieve unambiguous attribution of all scholarly contributions. You can create and link your ORCID from the home page of the MTS by clicking on 'Modify my Springer Nature account'. For more information please visit <http://www.springernature.com/orcid>.

Link Redacted

We would like to receive a revised submission within six months. We would be happy to consider a revision even after this timeframe, however if the resubmission deadline is missed and the paper is eventually published, the submission date will be the date when the revised manuscript was received.

We hope that you will find our referees' comments, and editorial guidance helpful. Please do not hesitate to contact me if there is anything you would like to discuss.

Best wishes,

Sabrya Carim

Sabrya Carim, PhD
(she/her/hers)
Associate Editor, Nature Cell Biology
Nature Portfolio

Springer Nature
The Campus, 4 Crinan Street, London N1 9XW, UK
sabrya.carim@springernature.com
<https://orcid.org/0000-0001-9485-1938>

Reviewers' Comments:

Reviewer #1 (Remarks to the Author):

This manuscript aims to investigate the correlation and causal relationship between cell fate decisions and nuclear mechanics. The authors demonstrate that mechano-osmotic changes induce general transcriptional repression and prime chromatin for cell fate transitions by lifting repression of specific genes. They analyze the interplay between mechanical and chemical (growth factor) cues, revealing distinct effects of these signals. While mechanical signals accelerate fate transitions, biochemical cues are necessary for robust and stable induction of specific fates.

The interaction and integration of different types of cellular cues (molecular, physical, etc.) is a topic of intense research, and we are only beginning to understand the complexities of how these signals influence cell behavior. This work presents a compelling argument for the interaction between mechanical and chemical cues in regulating nuclear activity and cell fate decisions, offering a significant contribution to the field.

Most of the data are clear and convincing; however, several issues need to be addressed before this manuscript can be considered for publication in Nature Cell Biology.

The title, abstract, results, and conclusions emphasize the importance of nuclear oscillations in controlling cell fate specification. The result is quite clear: the removal of growth factors triggers nuclear oscillations, and these are well characterized in the manuscript. However, why are these oscillations important and not simply an epiphenomenon of a more dynamic nucleus? In other words, what is the critical mechanical parameter that the cells are sensing? It seems more likely that cells are sensing nuclear deformation or volume changes rather than oscillations, as the authors can reproduce most of the effects via mechanical compression (which does not involve oscillation). If the authors wish to propose nuclear oscillations as a key factor in determining cell fate, they need to address this more directly and rule out other possible alternatives.

The system used to address the original question is highly artificial, as it is mainly in vitro. The physiological relevance of these findings in a more natural, in vivo context remains unclear. For example, in what biological systems could cells control their differentiation based on mechano-osmotic cues? While the authors show some parallels in nuclear deformation between iPSCs and human blastocysts, they do not discuss or explore what potential mechanical inputs might be driving those nuclear changes. Could it be the compaction of the

ICM? Is there any osmotic change in the blastocyst? Additionally, what evidence supports the notion that removing a growth factor induces differentiation *in vivo*? Is there a transient exposure to FGF in cells destined to become the ICM/epiblast?

The authors propose that mechano-osmotic forces trigger a redistribution of CBX2 condensates and genome-wide occupancy, leading to the de-repression of genes involved in osmotic stress response and differentiation. Given CBX2's central role in their model, the authors should perform functional and epistatic experiments with CBX2 to strengthen their claims.

The authors compare the effect of compression on YAP nuclear localization under two conditions: pluripotency and basal, showing that 5 μ m compression does not affect YAP nuclear localization in the basal condition. However, since the authors also report that the nucleus is stiffer in the basal condition, it is possible that this "mild" compression (compared with 3 μ m) is insufficient to deform a stiffer nucleus. To test this, the authors should measure nuclear deformation under these compression regimes. They should also analyze nuclear deformation in hypertonic treatment.

The authors conclude that the absence of pluripotent growth factors induces osmotic stress. However, it is unclear how this conclusion was reached; they show similar mechanical changes after growth factor withdrawal and during osmotic stress, but these similarities do not prove that growth factor withdrawal induces nuclear osmotic stress. A more direct demonstration is required.

Minor points:

Some conclusions are not fully supported by the data:

- In Fig. 1(i, j), the authors conclude that in basal media lacking pluripotency growth factors, compression increases nuclear wrinkling and amplifies nuclear envelope (NE) fluctuations; however, it is unclear whether these differences are significant (effect of nuclear compression on NE fluctuations).
- The values of NE fluctuations across different experiments under the same conditions appear inconsistent. For example, in Fig. 1, NE fluctuations in the basal condition are 280 nm (Fig. 1e), 500 nm (Fig. 1h), and under compression, 200-300 nm (Fig. 1j).
- In Fig. 1(k), due to the high variability in the data and the relatively small changes, it is not clear whether these differences are statistically significant.

Reviewer #2 (Remarks to the Author):

For various stem cell or progenitor systems, it has been frequently noted that cell fate changes often associate with nuclear shape and/or volume, but causality and relations to a nuclear 'mechanophenotype' are under-studied. Kinetics of expression changes and the determining physical or molecular factors are elaborated to some extent in this study, with physical compression of single cells often compared to hypertonic stress in numerous omics assays. Although I likely missed some key descriptions, a number of concerns temper enthusiasm for the present submission.

1. Does hypertonic stress affect nuclear envelope fluctuations? The label for Fig.1q is missing, but more importantly, the authors claim confinement-induced volume loss is not the mechanism of YAP's N/C "activation" because hypertonic stress has no effect. However, the imaging in Supp Figs does not seem quantified and is not clear to this reviewer. More controlled and rigorous experiments to assess the specific activity of YAP under confinement (rather than a generic response) should be done with RFP/GFP-NLS or other constructs to compare results in the same cell as Halo-YAP.
2. Stiffness measurements of nuclei in Fig.1L all seem 10-100 fold higher than are typically reported. Perhaps the indentation depth and nuclear height differences are having an effect.
3. Please clarify how the differential gene expression in Fig.2i was calculated. The authors write "strong IEG induction was not apparent upon 30 min compression in basal medium" but Fig.2L shows 4 highlighted IEGs that seem to respond similarly. Quantitation of these is needed.
4. Rescue of compression effects with HYPO-tonic swelling media would add needed rigor. Likewise, the authors write "we mimicked the cytoskeletal confinement using a compression bioreactor", but they should combine compression and CSK inhibition and re-do some of the key experiments to more rigorously establish the level of claimed mimicry. Should cite in suitable context Newsha Koushki et al (PNAS 2023) for a number of issues, including "using osmotic pressure, we demonstrated that nuclear compression even without active myosin or filamentous actin regulates YAP localization."
5. Fig.4d image labels aren't the same as plot labels, and Fig.4d doesn't clearly indicate to this reviewer that "removal of pluripotency factors as well as axial compression increased CBX2 condensation".
6. Lastly, the authors should make more clear what methods or controls ensure that profiling of the compressed cells is not an artifact of (i) more contact or adhesion with another surface (i.e. upper surface), (ii) less access to nutrients, (iii) transient responses rather than steady state, (iv) etc.

Reviewer #3 (Remarks to the Author):

This paper addresses the question of how cell shape changes and growth factor signals regulate the differentiation of human iPSCs. During germ layer differentiation of hiPSCs, nuclei became flattened and reduced their volume. Removal of pluripotency-maintaining factors altered the mechanical properties of the nuclei, which were enhanced by mechanical compression or hypertonic osmotic stress. Mechano-osmotic stress rapidly altered chromatin accessibility and histone modifications, priming chromatin for spontaneous differentiation. The distinct effects of mechanical compression and osmotic stress suggest that the mechanical deformation of the nucleus involves both mechanical and osmotic stress components. Pluripotency-maintaining growth factor signals determine the mechano-osmotic state of the nucleus, and the removal of pluripotency factors enhances the osmotic component of enhancer deformation. The effects of mechano-osmotic signals are modulated by pluripotency factor signals. In the absence of pluripotency factors, mechano-osmotic signals promote exit from pluripotency, while the pluripotency state is maintained in the presence of these factors. Finally, the

authors showed that mechanical compression increased the condensation of CBX2 at the nuclear periphery, sequestering CBX2/PRC1 from key differentiation genes and derepressing them. Based on these results, the authors proposed a model in which chromatin priming by mechano-osmotic signals lowers the energy barrier for a signaling-factor-driven cell state change.

The detailed analysis of how nuclear morphology changes affect chromatin status and their connection to pluripotency factor signaling and cell differentiation is novel and important. The experiments are generally of high quality, and the proposed model is conceptually novel and significant for the fields of developmental biology and stem cell biology. However, in several cases, it is difficult to follow the descriptions in the text. Some descriptions are unclear, do not match the figures, or the figures lack appropriate explanations. Clear mistakes are also present.

Specific comments:

1. Lines 104-106: The text states that "Morphometric analyses revealed that differentiation of these hiPSCs from primed pluripotency into the three germ layers was accompanied by a reduction in nuclear volume and nuclear flattening (Fig. 1a-c; Supplementary Fig. S1a-d)." However, the data only show a reduction in nuclear volume for the three germ layers and flattening in the ectoderm. Quantification data showing nuclear flattening in the mesoderm and endoderm are required.
2. Definition of osmotic stress is unclear. In lines 180-183, the text says, "Taken together, these findings indicate that withdrawal of pluripotency-maintenance growth factors leads to immediate changes in the nuclear mechanophenotype: increase in nuclear stiffening and induction of osmotic stress, which together increase macromolecular crowding of the nucleoplasm." The authors showed the induction of nuclear stiffening by hypertonic shock. This does not suggest the induction of osmotic stress due to the withdrawal of pluripotency factors. A clearer explanation is required to conclude that osmotic stress is induced by the removal of pluripotency growth factors.
3. Lines 137-138: "We further noted the presence of dynamic actin structures that originated from the extracellular space." What are these actin structures originating from the extracellular space? Are there any extracellular actins?
4. Lines 141-145: "We disrupted the cytoskeleton using a combination of cytochalasin D (actin) and nocodazole (microtubules) and quantified nuclear fluctuations. These analyses revealed amplification of fluctuations with pharmacological inhibitors of actin and microtubules, suggesting that the perinuclear cytoskeleton restricts and confines the nucleus (Fig. 1h; Supplementary Movie 3)." Data for this experiment are absent. The figure legend describes Fig. 1h as the experiment of ATP depletion. What does "CSK inhibited" refer to in this graph? Since the authors demonstrated the presence of a perinuclear actin ring and actin dynamics, the role of actin should be clarified with cytochalasin D treatment alone.
5. Lines 351-354: "Further, while in the pluripotency condition the long-term transcriptional response to transient compression showed a gene expression signature of FOS and STAT3 transcriptional targets and enrichment for cytoskeletal genes, consistent with the dominance of the mechanoresponse and YAP activity in this condition (Fig. 3c-e)." Fig. 3c-e does not contain data showing enrichment for cytoskeletal genes. Data should be provided.
6. Lines 392-395: "Finally, phosphosites upregulated specifically in response to hypertonic shock were regulators of the actomyosin cytoskeleton and Mitogen-activated protein kinases (MAPK) 1, 3, and 14, corresponding to the activation of ERK1/2 and p38 MAPKs (Fig. 4c)." Figure 4c does not show data for the upregulation of "regulators of the actomyosin cytoskeleton." This data should be provided. Since Figs. 4a-c, e contain several clusters, adding cluster numbers when referring to these figures would help readers locate the appropriate data.
7. Lines 395-397: The text describes, "these results highlight the distinct (YAP) and overlapping (ERK, PRC1) pathways activated by compression and hyperosmotic shock." However, activation of the ERK pathway is specific to hyperosmotic shock.
8. Lines 422-423: The text states, "High resolution imaging of CBX2 immunostaining showed that both removal of pluripotency factors as well as axial compression increased CBX2 condensation (Fig. 4d)." However, Fig. 4d shows that removal of pluripotency factors (Basal) did not increase CBX2 condensation.
9. Lines 430-432: "While removal of pluripotency factors led to reduced CBX2 occupancy at key differentiation genes, the reduction was strongest when a pulse of axial compression was applied (Fig. 4e, f)." It is unclear whether all the genes shown in Fig. 4e are key differentiation genes or only some of them.
10. Lines 434-439: "Upon compression in the basal medium, CBX2 occupancy was substantially reduced at genes repressed by pluripotency factors as well as HOX genes involved in differentiation (Fig. 4e-g)." It is unclear which gene(s) in Fig. 4g are pluripotency-repressed genes. The gene name(s) should be described in the text.

Minor comments:

1. Line 203, Fig. 1q should be Fig. 1p.
2. Line 315, Fig. 2d-f may be Fig. 2l.
3. In Figure 4d, what does FC/FT (ratio) represent?

Methods should be written concisely, but should contain all elements necessary to allow interpretation and replication of the results. As a guideline, Methods sections typically do not exceed 3,000 words. The Methods should be divided into subsections listing reagents and techniques. When citing previous methods, accurate references should be provided and any alterations should be noted. Information must be provided about: antibody dilutions, company names, catalogue numbers and clone numbers for monoclonal antibodies; sequences of RNAi and cDNA probes/primers or company names and catalogue numbers if reagents are commercial; cell line names, sources and information on cell line identity and authentication. Animal studies and experiments involving human subjects must be reported in detail, identifying the committees approving the protocols. For studies involving human subjects/samples, a statement must be included confirming that informed consent was obtained. Statistical analyses and information on the reproducibility of experimental results should be provided in a section titled "Statistics and Reproducibility".

All Nature Cell Biology manuscripts submitted on or after March 21 2016 must include a Data availability statement at the end of the Methods section. For Springer Nature policies on data availability see <http://www.nature.com/authors/policies/availability.html>; for more information on this particular policy see <http://www.nature.com/authors/policies/data/data-availability-statements-data-citations.pdf>. The Data availability statement should include:

- Accession codes for primary datasets (generated during the study under consideration and designated as "primary accessions") and secondary datasets (published datasets reanalysed during the study under consideration, designated as "referenced accessions"). For primary accessions data should be made public to coincide with publication of the manuscript. A list of data types for which submission to community-endorsed public repositories is mandated (including sequence, structure, microarray, deep sequencing data) can be found here <http://www.nature.com/authors/policies/availability.html#data>.
- Unique identifiers (accession codes, DOIs or other unique persistent identifier) and hyperlinks for datasets deposited in an approved repository, but for which data deposition is not mandated (see here for details <http://www.nature.com/sdata/data-policies/repositories>).
- At a minimum, please include a statement confirming that all relevant data are available from the authors, and/or are included with the manuscript (e.g. as source data or supplementary information), listing which data are included (e.g. by figure panels and data types) and mentioning any restrictions on availability.

- If a dataset has a Digital Object Identifier (DOI) as its unique identifier, we strongly encourage including this in the Reference list and citing the dataset in the Methods.

We recommend that you upload the step-by-step protocols used in this manuscript to protocols.io. More details can found at <https://www.protocols.io/help/publish-articles>.

All imaging data should be accompanied by scale bars, which should be defined in the legend. Cropped images of gels/blots are acceptable, but need to be accompanied by size markers, and to retain visible background signal within the linear range (i.e. should not be saturated). The boundaries of panels with low background have to be demarked with black lines. Splicing of panels should only be considered if unavoidable, and must be clearly marked on the figure, and noted in the legend with a statement on whether the samples were obtained and processed simultaneously. Quantitative comparisons between samples on different gels/blots are discouraged; if this is unavoidable, it should only be performed for samples derived from the same experiment with gels/blots were processed in parallel, which needs to be stated in the legend.

- Some programs can generate Postscript by 'printing to file' (found in the Print dialogue). If using an application not listed above, save the file in Postscript format or email our Art Editor, Allen Beattie for advice (a.beattie@nature.com).

The total number of Supplementary Figures (not including the "unprocessed scans" Supplementary Figure) should not exceed the number of main display items (figures and/or tables (see our Guide to Authors and March 2012 editorial <http://www.nature.com/ncb/authors/submit/index.html#suppinfo>; <http://www.nature.com/ncb/journal/v14/n3/index.html#ed>). No restrictions apply to Supplementary Tables or Videos, but we advise authors to be selective in including supplemental data.

GUIDELINES FOR EXPERIMENTAL AND STATISTICAL REPORTING

REPORTING REQUIREMENTS – To improve the quality of methods and statistics reporting in our papers we have recently revised the reporting checklist we introduced in 2013. We are now asking all life sciences authors to complete two items: an Editorial Policy Checklist (found here <https://www.nature.com/authors/policies/Policy.pdf>) that verifies compliance with all required editorial policies and a reporting summary (found here <https://www.nature.com/authors/policies/ReportingSummary.pdf>) that collects information on experimental design and reagents. These documents are available to referees to aid the evaluation of the manuscript. Please note that these forms are dynamic 'smart pdfs' and must therefore be downloaded and completed in Adobe Reader. We will then flatten them for ease of use by the reviewers. If you would like to reference the guidance text as you complete the template, please access these flattened versions at <http://www.nature.com/authors/policies/availability.html>.

Version 1:

Decision Letter:

Our ref: NCB-LE55330A

9th June 2025

Dear Dr. Wickström,

Thank you for submitting your revised manuscript "Mechano-osmotic signals control chromatin state and fate transitions in pluripotent stem cells" (NCB-LE55330A) and for your patience with the delayed review process. It has now been seen by a subset of the original referees and their comments are below. Despite our best efforts, we have been unable to receive the comments of reviewer #2, but I have checked the responses to Reviewer#2's original points and we have discussed in the editorial team. The reviewers find that the paper has improved in revision, and therefore we'll be happy in principle to publish it in Nature Cell Biology, pending minor revisions to satisfy the referees' final requests and to comply with our editorial and formatting guidelines.

Please ensure to address reviewer#3's textual points in the revised article. Please also ensure that reviewer#2 pt6 (from the previous round of review) is also clarified in the text.

We are now performing detailed checks on your paper and will send you a checklist detailing our editorial and formatting requirements in about two weeks. Please do not upload the final materials and make any revisions until you receive this additional information from us.

Thank you again for your interest in Nature Cell Biology Please do not hesitate to contact me if you have any questions.

Best wishes,

Sabrya Carim, PhD
(she/her/hers)
Senior Editor, Nature Cell Biology
Nature Portfolio

Springer Nature
The Campus, 4 Crinan Street, London N1 9XW, UK
sabrya.carim@springernature.com
<https://orcid.org/0000-0001-9485-1938>

Reviewer #1 (Remarks to the Author):

The authors have done a great job addressing my previous comments—and, in my opinion, those of the other reviewer—by including important new experiments and clarifications. They have produced a significant paper that tackles the crucial problem of cell fate determination, focusing on the interplay between nuclear mechanics, shape and volume, and biochemical signalling.

Reviewer #3 (Remarks to the Author):

In the revised version of the manuscript, the authors appropriately addressed all of my comments. I am satisfied with the changes made by the authors and support publication after incorporation of the minor comments listed below.

Comments

1. Line 282-285: The text states, "Differential gene expression analyses further confirmed that genes involved in the regulation of the actomyosin cytoskeleton including known YAP target genes (AMOTL1, TAGLN, CCN2), ... were upregulated at 30 min (Extended Data Fig. S3e)." However, these YAP target genes are not indicated in the figure. Please label these genes in the figure.
2. Line 286-288: "... genes involved in pluripotency such as SOX2, OCT4, and DUSP7 were largely unchanged upon 30 min of compression (Extended Data Fig. S3e, f)." However, Extended Data Fig. S3e shows DUSP7 as being upregulated. Please revise the text appropriately.
3. Figure 3I. "Puripotency" should be "Pluripotency".

Version 2:

Decision Letter:

Dear Dr Wickstrom,

I hope you are well. As mentioned previously, after discussion within the team including Dr. Sabrya Carim, I am happy to be handling these next steps with your submission at Nature Cell Biology.

I am pleased to inform you that your manuscript, "Mechano-osmotic signals control chromatin state and fate transitions in pluripotent stem cells", has now been accepted for publication in Nature Cell Biology.

Due to the importance of these deadlines, we ask that you please let us know now whether you will be difficult to contact over the next month. If this is the case, we ask you provide us with the contact information (email, phone and fax) of someone who will be able to check

the proofs on your behalf, and who will be available to address any last-minute problems.

Please note that *Nature Cell Biology* is a Transformative Journal (TJ). Authors may publish their research with us through the traditional subscription access route or make their paper immediately open access through payment of an article-processing charge (APC). Authors will not be required to make a final decision about access to their article until it has been accepted. [Find out more about Transformative Journals](https://www.springernature.com/gp/open-research/transformative-journals)

Authors may need to take specific actions to achieve compliance with funder and institutional open access mandates. If your research is supported by a funder that requires immediate open access (e.g. according to [Plan S principles](https://www.springernature.com/gp/open-science/plan-s-compliance) or the [NIH public access policy](https://www.springernature.com/gp/open-science/us-federal-agency-compliance)) then you should select the gold OA route, and we will direct you to the compliant route where possible. Because authors warrant under our subscription licensing terms that they haven't committed to licensing any version of their article under a licence inconsistent with the terms of our agreement – including the applicable embargo period – publication under the subscription model isn't suitable for authors whose funders require no embargo.

If you have not already done so, we strongly recommend that you upload the step-by-step protocols used in this manuscript to protocols.io (<https://protocols.io>), an open online resource that allows researchers to share their detailed experimental know-how. All uploaded protocols are made freely available and are assigned DOIs for ease of citation. Protocols and Nature Portfolio journal papers in which they are used can be linked to one another, and this link is clearly and prominently visible in the online versions of both. Authors who performed the specific experiments can act as primary authors for the Protocol as they will be best placed to share the methodology details, but the Corresponding Author of the present research paper should be included as one of the authors. By uploading your Protocols onto protocols.io, you are enabling researchers to more readily reproduce or adapt the methodology you use, as well as increasing the visibility of your protocols and papers. You can also establish a dedicated workspace to collect your lab Protocols. Further information can be found at <https://www.protocols.io/help/publish-articles>.

Nature Cell Biology encourages authors presenting evidence for cell, biological, molecular, and genetic interactions to consider communicating these findings using Biofactoid (<https://biofactoid.org/>). This tool helps users share a searchable representation of interactions (e.g. binding, gene expression, post-translational modification) between genes, gene products, or chemicals. Information added to Biofactoid, with author attribution, is shared on social media and public databases, such as Pathway Commons, where it can be discovered and analyzed in the context of a large and growing corpus of knowledge.

With kind regards,

Daryl

Daryl Jason Verzosa David, PhD

Senior Editor, Nature Cell Biology
Advisory Editor, npj Biological Physics and Mechanics
Nature Portfolio

Heidelberger Platz 3, 14197 Berlin, Germany
Email: daryl.david@nature.com
ORCID: <https://orcid.org/0000-0002-9253-4805>

** Visit the Springer Nature Editorial and Publishing website at http://editorial-jobs.springernature.com?utm_source=ejp_NCB_email&utm_medium=ejp_NCB_email&utm_campaign=ejp_NCB for more information about our career opportunities. If you have any questions please click [here](mailto:editorial.publishing.jobs@springernature.com).

Reviewers' Comments:

Reviewer #1 (Remarks to the Author):

The interaction and integration of different types of cellular cues (molecular, physical, etc.) is a topic of intense research, and we are only beginning to understand the complexities of how these signals influence cell behavior. This work presents a compelling argument for the interaction between mechanical and chemical cues in regulating nuclear activity and cell fate decisions, offering a significant contribution to the field.

Most of the data are clear and convincing; however, several issues need to be addressed before this manuscript can be considered for publication in Nature Cell Biology.

We thank the reviewer for this positive assessment of our work and finding the data compelling and significant. We are grateful for the constructive comments and expert suggestions that helped us to further strengthen the manuscript.

The title, abstract, results, and conclusions emphasize the importance of nuclear oscillations in controlling cell fate specification. The result is quite clear: the removal of growth factors triggers nuclear oscillations, and these are well characterized in the manuscript. However, why are these oscillations important and not simply an epiphenomenon of a more dynamic nucleus? In other words, what is the critical mechanical parameter that the cells are sensing? It seems more likely that cells are sensing nuclear deformation or volume changes rather than oscillations, as the authors can reproduce most of the effects via mechanical compression (which does not involve oscillation). If the authors wish to propose nuclear oscillations as a key factor in determining cell fate, they need to address this more directly and rule out other possible alternatives.

We appreciate this comment and fully agree that while it is clear that growth factor removal triggers both fluctuations and deformation, it is important to more clearly define the most relevant signal that is being sensed by the cells. As we had observed that nuclear deformation by both mechanical compression and osmotic pressure accelerated pluripotency exit, we reasoned that if nuclear fluctuations were the driving force for the state transition, they should also be induced by osmotic pressure alone. We investigated this by subjecting cells to 30-minute hypertonic shock followed by washout and recovery with or without pluripotency growth factors. While this treatment caused volume loss of the nucleus, it also transiently ceased nuclear fluctuations (new Extended Data Figure 2d). After hyperosmotic stress was removed, nuclear fluctuations resume with more fluctuations in the absence of growth factors.

We further analyzed the source of these nuclear fluctuations and show that treating iPSC colonies with CalyculinA to increase contractility strongly enhanced nuclear fluctuations and triggered nuclear volume loss and osmotic stress (new Fig. 2d; Extended Data Figure 1e, f).

Thus, we conclude that nuclear fluctuations, while being part of the biological cascade that is triggered by nuclear deformation during hPSC colony compaction, are not essential for the cell fate effects. Instead, osmotic stress and the resulting macromolecular crowding/condensate remodeling are the key mechanism of epigenetic fate regulation by mechanical forces. We have clarified this conclusion in the manuscript.

The system used to address the original question is highly artificial, as it is mainly in vitro.

The physiological relevance of these findings in a more natural, in vivo context remains unclear. For example, in what biological systems could cells control their differentiation based on mechano-osmotic cues? While the authors show some parallels in nuclear deformation between iPSCs and human blastocysts, they do not discuss or explore what potential mechanical inputs might be driving those nuclear changes. Could it be the compaction of the ICM? Is there any osmotic change in the blastocyst? Additionally, what evidence supports the notion that removing a growth factor induces differentiation in vivo? Is there a transient exposure to FGF in cells destined to become the ICM/epiblast?

While the reviewer is raising an important point, investigating causality of mechanical signals in *in vivo* models remains highly challenging due to the complexity of *in vivo* biological systems and the many feedback loops that exist between mechanical and biochemical signaling. This is why using direct mechanical perturbations in iPSC cells are well established as a biological system for fate transitions and is thus valuable in our opinion

We would also like to emphasize that while FGF signaling is critical for the maintenance of primed pluripotency *in vitro* and, importantly, the removal of FGF triggers pluripotency exit (see for example Xu et al, Stem Cells 2005 PMID 15749926; Dvorak et al., Stem Cells 2005 PMID 15955829; Levenstein et al., Stem Cells 2006 PMID 16282444; Chen et al., Nat Methods 2011 PMID 21478862), FGF does not play the same role in the ICM. Thus, while FGF is the relevant signal in the primed iPSCs, we did not intend to convey that the mechano-osmotic priming is a universal effect of FGF signaling across various differentiation decisions. Rather, we propose that the mechano-osmotic nuclear effects directly relate to the fate transition itself and, depending on the cellular context, can be triggered by a number of signaling pathways/growth factors.

To further experimentally address these two central points, we have now performed a large number of additional experiments using a wide range scenarios that model *in vivo* cell fate transitions:

1. We have established human 3D blastoid models from naïve iPSCs where we observe, as in human embryos, GATA6-positive inner cell mass cells have lower nuclear volumes. We further observe that these cells have higher level of activity of the osmosensitive kinase p38 (new Fig. 1d).

2. We then proceeded to understand if this osmotic stress response is a shared feature of compaction-associated differentiation responses and also of primed pluripotency. To this end, we have included analyses of the 2D gastruloid system of primed hPSCs differentiation and show that prior to fate transitions, the micropatterned colonies undergo large scale compaction that is again associated with nuclear deformation that precede differentiation (new Fig. 1h-l). Intriguingly, we observe a similar osmotic stress signal characterized by p38 activation. Thus, an osmotic stress response is shared in different *in vitro* models of embryonic cell fate transitions.

3. We then addressed whether the osmotic stress response is directly controlled by compaction and is not caused by indirect effects from signaling. To do so, we triggered compaction by upregulating contractility/RhoA activity using CalyculinA treatment. As expected, Calyculin triggered strong compaction of the primed iPSC colony to induce a osmotic stress response specifically at the colony edges where nuclear deformation is most substantial (new Extended Data Fig. 1e-f).

4. Finally, to understand if this osmotic stress response not only precedes but is indeed causative for differentiation, we performed a rescue experiment of mechanically compressed cells with hypo-osmotic medium. Intriguingly, brief hypo-osmotic treatment post compression attenuates spontaneous differentiation of compressed cells (new Extended Data Fig. 6c).

Collectively these experiments demonstrate that cell fate transitions in different systems of pluripotency are associated with osmotic stress response and stress can be induced by increase in cellular compaction and nuclear deformation. This osmotic stress response then helps facilitate cell fate transitions. We are grateful for the reviewer for helping us to sharpen this key conclusion of the study.

The authors propose that mechano-osmotic forces trigger a redistribution of CBX2 condensates and genome-wide occupancy, leading to the de-repression of genes involved in osmotic stress response and differentiation. Given CBX2's central role in their model, the authors should perform functional and epistatic experiments with CBX2 to strengthen their claims.

We have now depleted CBX2 from iPSCs using CRISPRi approach and show that removal of CBX2 accelerates spontaneous exit from pluripotency (new Extended Data Fig. 7d, e), thus directly demonstrating the critical role for CBX2 in restricting iPSC differentiation. This experiment is consistent with the sequencing experiments showing loss of CBX2 occupancy genome wide in cells in basal medium and, in particular, in response to compression. We have further included live imaging of endogenously tagged CBX2 to show how growth factor removal, osmotic stress and compression all trigger CBX2 condensate remodeling, explaining the displacement of CBX2 from its targets resulting in their de-repression (new Fig. 5d; new Supplementary Movie 7). These experiments further strengthen the notion that CBX2-PRC1 is important in silencing differentiation genes in primed iPSCs.

The authors compare the effect of compression on YAP nuclear localization under two conditions: pluripotency and basal, showing that 5 μ m compression does not affect YAP nuclear localization in the basal condition. However, since the authors also report that the nucleus is stiffer in the basal condition, it is possible that this "mild" compression (compared with 3 μ m) is insufficient to deform a stiffer nucleus. To test this, the authors should measure nuclear deformation under these compression regimes. They should also analyze nuclear deformation in hypertonic treatment.

We apologize that we were not sufficiently clear in our conclusion was intended to be similar to what the reviewer is proposing; due to the mechanical changes that are induced upon FGF removal, a larger deformation is required to activate YAP in these stiffer cells. We have edited the text to make this point clearer. As suggested by the reviewer we have now also included the analyses of nuclear deformation in these conditions (new Fig. 2m).

The reviewer may have missed this but the original manuscript already included quantification of nuclear deformation in the hypertonic condition (Extended Data Fig 2c in the revised manuscript), showing nuclear envelope wrinkling and volume loss.

The authors conclude that the absence of pluripotent growth factors induces osmotic stress. However, it is unclear how this conclusion was reached; they show similar mechanical changes after growth factor withdrawal and during osmotic stress, but these similarities do not prove that growth factor withdrawal induces nuclear osmotic stress. A more direct demonstration is required.

We have based this conclusion on the strong upregulation of the osmo-sensitive metallothionein genes in the basal medium condition compared to the pluripotency medium (Extended Data Fig 7a and Extended Data Table 4 in the revised manuscript). To even further strengthen this aspect, we have now included extensive analyses of p38 activation as another readout for osmotic stress and show that, as already highlighted in our answers to previous points, under physiological conditions of compaction and differentiation, including growth factor removal, we observe a subtle but robustly reproducible activation of p38 (new Extended Data Fig. 2b).

Minor points:

Some conclusions are not fully supported by the data:

- *In Fig. 1(I, j), the authors conclude that in basal media lacking pluripotency growth factors, compression increases nuclear wrinkling and amplifies nuclear envelope (NE) fluctuations; however, it is unclear whether these differences are significant (effect of nuclear compression on NE fluctuations).*

We have now revisited the statistical analyses to include statistical analyses also for the effect of compression and not just the media conditions. Compression induces a statistically significant change in nuclear envelope fluctuations in both media conditions. The most substantial difference both in terms of significance and biological effect remains the increased fluctuations in basal media conditions after 30 min of compression (Fig. 2b in the revised manuscript).

- *The values of NE fluctuations across different experiments under the same conditions appear inconsistent. For example, in Fig. 1, NE fluctuations in the basal condition are 280 nm (Fig. 1e), 500 nm (Fig. 1h), and under compression, 200-300 nm (Fig. 1j).*

We are grateful for the reviewer for noticing that the fluctuation values for the inhibitor treatments were indeed slightly higher than in the other datasets. A discrepancy in the absolute value of fluctuations was generated by incorrect scale transfer when switching between different microscope objectives. We have corrected this (Fig. 2d in revised manuscript).

- *In Fig. 1(k), due to the high variability in the data and the relatively small changes, it is not clear whether these differences are statistically significant.*

The variability of the data arises from effects of the cell cycle, where nuclei increase in size in G2, leading to large standard deviation in population means. Thus, when we plot means across hundreds of cells the data appears less impressive although the changes are very robust and highly reproducible. We have included statistical analyses that support the conclusion of a more rapid and substantial volume loss in basal medium (Fig. 2g in the revised manuscript).

Reviewer #2 (Remarks to the Author):

For various stem cell or progenitor systems, it has been frequently noted that cell fate changes often associate with nuclear shape and/or volume, but causality and relations to a nuclear 'mechanophenotype' are under-studied. Kinetics of expression changes and the determining physical or molecules factors are elaborated to some extent in this study, with physical compression of single cells often compared to hypertonic stress in numerous omics assays. Although I likely missed some key descriptions, a number of concerns temper enthusiasm for the present submission.

We appreciate the reviewers' constructive comments and expert suggestions that helped us to further strengthen the manuscript.

1. Does hypertonic stress affect nuclear envelope fluctuations? The label for Fig. 1q is missing, but more importantly, the authors claim confinement-induced volume loss is not the mechanism of YAP's N/C "activation" because hypertonic stress has no effect. However, the imaging in Supp Figs does not seem quantified and is not clear to this reviewer. More controlled and rigorous experiments to assess the specific activity of YAP under confinement (rather than a generic response) should be done with RFP/GFP-NLS or other constructs to compare results in the same cell as Halo-YAP.

We have now included quantification of YAP nuclear localization upon hypertonic stress to support the conclusion that hypertonic stress does not activate YAP in iPSCs unlike mechanical deformation (new Extended Data Fig. 2g).

We have also quantified nuclear fluctuations in response to hypertonic stress and see that this arrests nuclear envelope fluctuations that rapidly recover upon change into isotonic medium, with again more abundant fluctuations in the basal medium condition compared to FGF-containing medium (new Extended Data Fig. 2d). We have further included additional data on the role of the cytoskeleton and show that enhancing contractility amplifies nuclear fluctuations, whereas inhibiting F-actin polymerization attenuates but does not completely arrest fluctuations (new Fig 2d). In contrast, ATP depletion arrests fluctuations (new Fig. 2d). Thus, we conclude that the fluctuations reflect the force balance across the nuclear envelope (as was suggested by others previously; Chu et al., PNAS 2017 PMID 28900009), where the perinuclear cytoskeleton generates active forces that confine and deform the nucleus and chromatin, and that are counteracted by osmotic dynamics (volume loss and volume recovery).

We have also followed the reviewers' suggestion to compare GFP-NLS nuclear localization in response to compression in the HALO-YAP cells and show that while YAP nuclear entry is triggered by compression, NLS GFP nuclear intensity mildly decreases (new Extended Data Fig. 2h, i), suggesting that the effects of compression are specific to YAP.

2. Stiffness measurements of nuclei in Fig. 1L all seem 10-100 fold higher than are typically reported. Perhaps the indentation depth and nuclear height differences are having an effect. We thank the reviewer for the careful examination of the data and the higher nuclear stiffness values in our measurements compared to previous reports. We have carefully examined our methodology and data as well as performed additional experiments to address this concern. Our analysis reveals key points that we have subsequently clarified in the manuscript:

1. Our analysis of force-indentation curves showed indentation depths between 500-1500 nm, which is less than 10% of the typical 15 μ m nuclear height in iPSCs. In this depth range it is highly unlikely for substrate effects to artificially increase our elastic modulus measurements. Thus, use of a contact mechanics model incorporating substrate effects would therefore not be suitable in this context (Dimitriadis et al., Biophys J 2002 PMID 11964265; Gavara and Chadwick Nat Nanotechnol 2012 PMID 23023646). The exception is the hypertonic shock condition (Extended Data Fig. 2e), where the ~50% decrease in nuclear height may indeed lead to some substrate-related contribution.
2. We routinely use this exact experimental setup for measuring nuclear stiffness in a range of cell types and typically obtain values that are in the range of 1-20 kPa. We now also performed a direct side-by-side comparison experiment with mouse ESCs and hiPSCs.

Whereas we measured stiffnesses of 1kPa for the mESCs, we reproduced the 20-50 kPa values that we report for hiPSCs in the current manuscript, further increasing our confidence in these measurements.

3. Given these differences between the two cell types, we reasoned that our measurements in the iPSCs may reflect a composite stiffness that includes contributions from both the nucleus and the very dense perinuclear actin network that is present in these hiPSC cells. To test this hypothesis, we disrupted the actin cytoskeleton with cytochalasin D, and indeed observed that this reduced the measured stiffness to ~2 kPa (**new Extended Data Fig. 2f**), which aligns with previously reported values for iPSC nuclear stiffness.
4. Most importantly, when measuring isolated nuclear stiffness after actin, depolymerization, we still observe a significant increase in stiffness (from 2 kPa to 4 kPa) within 5 min upon removing pluripotency factors. This confirms our central conclusion that growth factor removal rapidly alters nuclear mechanics, independent of cytoskeletal effects, and serves as an additional control for these experiments (**new Extended Data Fig. 2f**).

3. Please clarify how the differential gene expression in Fig.2i was calculated. The authors write "strong IEG induction was not apparent upon 30 min compression in basal medium" but Fig.2L shows 4 highlighted IEGs that seem to respond similarly. Quantitation of these is needed.

The differential gene expression is calculated using DESeq2 (size factors estimated using ERCC spike-ins), with Z-scores of expression changes plotted in **Fig. 31** (Fig. 2l in the previous version). To address this point and clarify the conclusions from the differential gene expression analysis between the conditions we have added a new volcano plot (**new Extended Data Fig. 5c**) that specifically shows the differential expression analysis of genes in the basal condition compared to compression in the basal condition. This statistical analysis demonstrates that changes in IEGs are not statistically significant. We have further modified our conclusion to state that "IEG induction was less apparent upon 30 min compression in basal medium". We have also expanded our methods section to provide more detail about how the differential expression analysis was performed.

4. Rescue of compression effects with HYPO-tonic swelling media would add needed rigor. Likewise, the authors write "we mimicked the cytoskeletal confinement using a compression bioreactor", but they should combine compression and CSK inhibition and re-do some of the key experiments to more rigorously establish the level of claimed mimickry. Should cite in suitable context Newsha Koushki et al (PNAS 2023) for a number of issues, including "using osmotic pressure, we demonstrated that nuclear compression even without active myosin or filamentous actin regulates YAP localization. "

We have now followed the reviewers' suggestion and provide data that hypotonic media rescues the effect of compression on iPSC differentiation (**new Extended Data Fig. 6c**), further strengthening the conclusion that osmotic stress drives the effects of confinement on different. We further show that increasing contractility using Calyculin triggers nuclear deformation and osmotic stress, similarly to compression (**new Extended Data Fig. 1e, f**), Finally, have added the excellent suggestion of Newsha Koushki et al (PNAS 2023) reference (p.5 last paragraph, p8 last paragraph). We thank the reviewer for these expert suggestions.

5. Fig.4d image labels aren't the same as plot labels, and Fig.4d doesn't clearly indicate to

this reviewer that "removal of pluripotency factors as well as axial compression increased CBX2 condensation".

We apologize for the inconsistency in labeling that we have now corrected. The constructive criticism of the reviewer prompted us to try to better capture and quantify the dynamics of the proposed redistribution of CBX2 condensates. To this end we generated an endogenous GFP reporter iPSC line for CBX2 using CRSIPR. This endogenously tagged CBX2 nicely recapitulated the condensate pattern of CBX2 in the nucleus. Intriguingly, live imaging analyses of CBX2 dynamics in response to various treatments indicated that while medium change from pluripotency medium to fresh pluripotency medium did not influence CBX2 dynamics, a change into basal medium as well as compression or hyperosmotic stress rapidly dissolved the existing CBX2 condensates. This was followed by reassembly of condensates, with the CBX2 becoming more aggregated specifically in compressed cells, (**new Fig. 5d; new Supplementary Movie 7**). As pointed out by the reviewer, these observations also explain the initial findings reported in the original submission with no major differences in condensate size in basal medium when compared to pluripotency medium. For clarity, we now additionally quantify condensate number instead of intensity ratios to better document the change in condensate dynamics also in the basal medium condition (**new Extended Data Fig. 7b**). Collectively this new data strengthens the conclusion that hyperosmotic stress triggered by growth factor removal and/or nuclear deformation results in remodeling and redistribution of the CBX2 condensates.

6. Lastly, the authors should make more clear what methods or controls ensure that profiling of the compressed cells is not an artifact of (i) more contact or adhesion with another surface (i.e. upper surface), (ii) less access to nutrients, (iii) transient responses rather than steady state, (iv) etc.

While the compression system is obviously artificial and might trigger effects highlighted by the reviewer, the “omics” experiments that we performed did not provide any indications of such potential confounding effects.

(i) If there would be a dominant effect of an upper surface (compression stamp) contacting the apical membrane, we would not expect to observe the differential responses seen between compression in pluripotency versus basal media conditions. Further, the substantial similarity in responses between the hyperosmotic stress response, which involves no surface contact or nutrient access issues, make it unlikely that more adhesion or contact with another surface would play a critical role in the impact of compression on differentiation.

(ii) Importantly, there was no observable effect on the cell cycle or signs of cell death (**Extended Data Fig. 3b**), contrary to what nutrient depletion would have caused. To further address this, we now also quantitatively compared gene expression patterns related to nutrient stress/starvation-responsive genes as well as hypoxia-induced genes between compressed and uncompressed cells and observe no differences upon compression (**new Extended Data Fig. 3b**).

(iii) Finally, regarding the reviewer’s question of transient vs steady state responses, it is not fully clear to us what the reviewer means. Our data and manipulations clearly show that the osmotic effects of compression on nuclear volume are transient, with a rapid recovery of nuclear volume, as has been described for osmotic stress previously (see for example Ni et al., Cell Reports 2024 PMID 39579355; Roffay et al., PNAS 2021 PMID: 34785592). However, the gene-regulatory effects of this transient osmotic stress are long-lasting specifically in the basal media conditions, as observed by altered transcriptional profiles 24h

after a 30 min transient compression, and by altered differentiation trajectories 96h after just a 30 min compression (see Fig. 4). Importantly, our new experiments in response to the comments from Reviewer 1 highlight that a similar transient osmotic stress response can be observed under physiological conditions of ICM or 2D colony compaction, or with inhibitor treatments to enhance contractility (new Fig 1d, k, l; Extended Data Fig. 1e, f). Thus, we conclude that short-term nuclear deformation triggers a transient osmotic stress response that has long-term consequences by accelerating exit from pluripotency.

Reviewer #3 (Remarks to the Author):

The detailed analysis of how nuclear morphology changes affect chromatin status and their connection to pluripotency factor signaling and cell differentiation is novel and important. The experiments are generally of high quality, and the proposed model is conceptually novel and significant for the fields of developmental biology and stem cell biology. However, in several cases, it is difficult to follow the descriptions in the text. Some descriptions are unclear, do not match the figures, or the figures lack appropriate explanations. Clear mistakes are also present.

We thank the reviewer for this positive assessment of our work and finding the data novel, important and of high quality. We greatly appreciate the feedback on the lack of clarity in some cases and apologize for mistakes in the manuscript. We have carefully checked and corrected mistakes, and edited the manuscript for clarity as well as performed a number of additional experiments to further strengthen the manuscript.

Specific comments:

1. Lines 104-106: The text states that “Morphometric analyses revealed that differentiation of these hiPSCs from primed pluripotency into the three germ layers was accompanied by a reduction in nuclear volume and nuclear flattening (Fig. 1a-c; Supplementary Fig. S1a-d).” However, the data only show a reduction in nuclear volume for the three germ layers and flattening in the ectoderm. Quantification data showing nuclear flattening in the mesoderm and endoderm are required.

We have included the data on nuclear flattening for mesoderm and endoderm (new Extended Data Fig. 1b, c).

2. Definition of osmotic stress is unclear. In lines 180-183, the text says, “Taken together, these findings indicate that withdrawal of pluripotency-maintenance growth factors leads to immediate changes in the nuclear mechanophenotype: increase in nuclear stiffening and induction of osmotic stress, which together increase macromolecular crowding of the nucleoplasm.” The authors showed the induction of nuclear stiffening by hypertonic shock. This does not suggest the induction of osmotic stress due to the withdrawal of pluripotency factors. A clearer explanation is required to conclude that osmotic stress is induced by the removal of pluripotency growth factors.

We appreciate this helpful feedback. Our conclusions were based on the data showing that removal of growth factors triggered nuclear volume loss and nuclear stiffening, similar to what is observed upon hyperosmotic shock (Fig. 2a, h and Extended Data Fig. 2c, e in the revised manuscript). We further showed that expression of the osmo-sensitive metallothioneine genes is strongly induced by withdrawal of growth factors (Extended Data Fig. 7a in the revised manuscript). To further strengthen this conclusion, we have analyzed the osmotic stress responses in more detail using a panel of new experiments:

1. We have established 3D blastoids from human naïve PSCs where we observe the same correlation as in the human embryos that GATA6-positive inner cell mass cells have lower nuclear volumes. We further observe that these cells have higher levels of phosphorylation on the osmo-sensitive kinase p38, indicating its enhanced activity (new Fig. 1d).
2. We then proceeded to understand if this osmotic stress response is a common feature of compaction-associated differentiation responses and also of primed pluripotency. For this we utilize the 2D gastruloid system of primed hPSC differentiation and show that prior to differentiation, the micropatterned colonies undergo large scale compaction that is associated with nuclear deformation that precedes differentiation (new Fig. 1h-l). Intriguingly, we observe a similar osmotic stress signal characterized by p38 activation.
3. To confirm that this osmotic stress response is downstream of the growth factor removal-induced nuclear mechanophenotype, we confirm a transient upregulation of p38 activity upon withdrawal of pluripotency growth factors (new Extended Data Fig. 2b)
4. To understand if the osmotic stress response is directly controlled by compaction and not by indirect effects from signaling, we trigger compaction by upregulating contractility/RhoA activity using CalyculinA treatment. As expected, Calyculin triggers strong compaction of the primed iPSC colony and a transient osmotic stress response, specifically at the colony edges where the nuclear deformation is most substantial (new Extended Data Fig. 1e, f).
5. Finally, to understand if this osmotic stress response is causative for differentiation, we perform a rescue experiment of mechanically compressed cells with hypo-osmotic medium. Intriguingly, brief hypo-osmotic treatment post compression attenuates spontaneous differentiation of compressed cells (new Extended Data Fig. 6d).

Collectively this new panel of experiments demonstrate that compaction of hPSCs triggers nuclear deformation, volume loss and hyperosmotic stress, and this hyperosmotic stress accelerates exit from pluripotency.

3. Lines 137-138: “We further noted the presence of dynamic actin structures that originated from the extracellular space.” What are these actin structures originating from the extracellular space? Are there any extracellular actins?

We apologize for this unclear formulation. We certainly did not intend to convey that actin is extracellular. To better characterize the nature of these invaginations we have now performed additional analyses of the invaginations using a marker for the plasma membrane (MemGlow) together with F-actin (FastAct). These new experiments show that the actin-containing invaginations originate from intercellular junctions and are associated with the plasma membrane (new Fig. 2c). More rarely we see that these invaginations are devoid of actin and are blebs derived from dividing cells that then compress neighboring nuclei (new Fig. 2c). These two observations point to high pressure within the colonies as the source for the dynamic deformations. Consistently, we further now demonstrate that increasing actomyosin contractility by Calyculin A treatment enhances nuclear fluctuations, supporting the conclusion that active forces from actomyosin contractility contribute to nuclear fluctuations (new Fig. 2d). We have edited the conclusion accordingly and now state “Indeed, live imaging of actin dynamics revealed two key structures: a taught, perinuclear actin ring that encapsulates the nuclear envelope, and dynamic intercellular cavities resembling “microlumens” that appeared to actively deform the nucleus at the same spatiotemporal

scales observed for nuclear fluctuations (Fig. 2c; Supplementary Movie 3). Occasionally, membrane blebs, frequently associated with mitotic cells, were also found to correlate with nuclear deformations (Fig. 2c; Supplementary Movie 3)."

4. Lines 141-145: *"We disrupted the cytoskeleton using a combination of cytochalasin D (actin) and nocodazole (microtubules) and quantified nuclear fluctuations. These analyses revealed amplification of fluctuations with pharmacological inhibitors of actin and microtubules, suggesting that the perinuclear cytoskeleton restricts and confines the nucleus (Fig. 1h; Supplementary Movie 3)." Data for this experiment are absent. The figure legend describes Fig. 1h as the experiment of ATP depletion. What does "CSK inhibited" refer to in this graph? Since the authors demonstrated the presence of a perinuclear actin ring and actin dynamics, the role of actin should be clarified with cytochalasin D treatment alone.*

We apologize for the unclarity in the labeling and in the conclusions. "CSK" was an abbreviation for Cytoskeleton, referring to disruption of both actin and microtubules (cytochalasinD + nocodazole). We have now replaced CSK label with cytochalasinD + nocodazole for clarity.

We have further modified experiment according to the reviewer's suggestion, focusing on actomyosin dynamics. These new experiments show that increasing contractility by treating cells with CalyculinA that enhances actomyosin contractility accelerates fluctuations and triggers nuclear deformation/osmotic stress (new Fig. 2d). Depleting polymerized actin networks with cytochalasin D mildly reduces nuclear fluctuations, whereas disrupting both microtubule and actin cytoskeletons accelerate fluctuations (new Fig. 2d). As already shown in the previous version of the manuscript, ATP depletion completely halts all fluctuations. Collectively these findings lead us to conclude that fluctuations are a compound consequence of both active actomyosin deformation and actin-dependent confinement around the nucleus and energy-dependent dynamics that occur inside the nucleus, as previously reported (Chu et al., PNAS 2017 PMID 28900009).

5. Lines 351-354: *"Further, while in the pluripotency condition the long-term transcriptional response to transient compression showed a gene expression signature of FOS and STAT3 transcriptional targets and enrichment for cytoskeletal genes, consistent with the dominance of the mechanoresponse and YAP activity in this condition (Fig. 3c-e)." Fig. 3c-e does not contain data showing enrichment for cytoskeletal genes. Data should be provided.*

To support this statement, we have included a heatmap of the genes specifically upregulated in the CompPluri condition compared to CompBasal condition (new Extended Data Fig. 6a). While there are a number of cytoskeletal regulators in this group of genes specifically upregulated in the CompPluri condition (Myh6, Cldn18, Tll11, Sfn1), given the small number of genes upregulated, we realize that using the term "enrichment" appears too strong.

We have thus altered the description of this dataset to state that "Further, in the pluripotency condition the long-term transcriptional response to transient compression was less pronounced (85 significantly upregulated genes vs 179 significantly upregulated genes; p-value <0.05, log₂FC>1) showing a gene expression signature of FOS and STAT3 transcriptional targets and genes involved in regulation of the cytoskeleton and lipid composition (Fig. 4c-e; Extended Data Fig. S6a), consistent with the enhancer profiling."

6. Lines 392-395: *“Finally, phosphosites upregulated specifically in response to hypertonic shock were regulators of the actomyosin cytoskeleton and Mitogen-activated protein kinases (MAPK) 1, 3, and 14, corresponding to the activation of ERK1/2 and p38 MAPKs (Fig. 4c).”* Figure 4c does not show data for the upregulation of “regulators of the actomyosin cytoskeleton.” This data should be provided. Since Figs. 4a-c, e contain several clusters, adding cluster numbers when referring to these figures would help readers locate the appropriate data.

We apologize for the unclarity here and thank the reviewer for the useful suggestion to include cluster numbers in the text when describing the specific patterns of regulation. We have now edited the text to include these.

The statement on cytoskeletal genes was referring to the GO term analyses that show enrichment of the term “Activation of GTPase activity” to be enriched in cluster 3 that is specific for hypertonic stress. We have now edited the text to make this clearer (replaced regulators of actomyosin cytoskeleton with Activation of GTPase activity) and provided examples of these proteins, which are mostly cytoskeletal regulators (Vimentin, Pak1, Myo9B, Dock2). The full dataset is included as **Supplementary Table 5**)

7. Lines 395-397: *The text describes, “these results highlight the distinct (YAP) and overlapping (ERK, PRC1) pathways activated by compression and hyperosmotic shock.”* However, activation of the ERK pathway is specific to hyperosmotic shock.

We thank the reviewer for this comment. Indeed, there was an unfortunate mistake in the text as compression in the pluripotency medium does not trigger robust ERK1/2 activation at the time scales studied.

8. Lines 422-423: *The text states, “High resolution imaging of CBX2 immunostaining showed that both removal of pluripotency factors as well as axial compression increased CBX2 condensation (Fig. 4d).”* However, Fig. 4d shows that removal of pluripotency factors (Basal) did not increase CBX2 condensation.

The constructive criticism of this reviewer and Reviewer 2 prompted us to set up experiments that better capture the dynamics of the proposed redistribution of CBX2 condensates. To this end we generated an endogenous fluorescent reporter iPSC line for CBX2 using Crispr. The endogenously tagged CBX2 nicely recapitulated the condensate pattern of CBX2 in the nucleus. Intriguingly, live imaging analyses of CBX2 dynamics in response to various treatments indicated that while medium change from pluripotency medium to fresh pluripotency medium did not influence CBX2 dynamics, a change into basal medium as well as compression or hyperosmotic stress rapidly dissolved the existing CBX2 condensates. This was followed by reassembly of new condensate with variable dynamics, with the CBX2 becoming more aggregated specifically in compressed cells (**new Fig. 5d and Supplementary Movie 7**). This is also consistent with the previous findings that did not show major differences in condensate size in basal medium when compared to pluripotency medium, as also pointed out by the reviewer. For clarity, we now quantify condensate number instead of intensity ratios to better document the altered condensate dynamics also in the basal medium condition (**new Extended Data Fig. 7b**).

Collectively this new data strengthens the conclusion that hyperosmotic stress triggered by growth factor removal and/or nuclear deformation results in remodeling and redistribution of the CBX2 condensates, while the condensates on average are indeed not larger in the basal medium condition compared to the presence of pluripotency factors.

9. Lines 430-432: “While removal of pluripotency factors led to reduced CBX2 occupancy at key differentiation genes, the reduction was strongest when a pulse of axial compression was applied (Fig. 4e, f).” It is unclear whether all the genes shown in Fig. 4e are key differentiation genes or only some of them.

All of the genes marked have been shown to be implicated in differentiation. We have now further marked key transcription factors that regulate differentiation with an asterisk.

10. Lines 434-439: “Upon compression in the basal medium, CBX2 occupancy was substantially reduced at genes repressed by pluripotency factors as well as HOX genes involved in differentiation (Fig. 4e-g).” It is unclear which gene(s) in Fig. 4g are pluripotency-repressed genes. The gene name(s) should be described in the text.

We have added the gene names into the text.

Minor comments:

1. Line 203, Fig. 1q should be Fig. 1p.

2. Line 315, Fig. 2d-f may be Fig. 2l.

We thank the reviewer for pointing out these mistakes, we have ensured correct figure labeling throughout the revised manuscript.

3. In Figure 4d, what does FC/FT (ratio) represent?

It represented the ration of condensed (C) to total (T) intensity ratio of CBX2 signal. We have now replaced this quantification with quantification of condensate number as explained above (Extended Data Fig. 7b in the revised manuscript).

Remarks to the Author:

The authors have done a great job addressing my previous comments—and, in my opinion, those of the other reviewer—by including important new experiments and clarifications. They have produced a significant paper that tackles the crucial problem of cell fate determination, focusing on the interplay between nuclear mechanics, shape and volume, and biochemical signalling.

We are grateful for this positive assessment of our work

Reviewer #2:

None

Reviewer #3:

Remarks to the Author:

In the revised version of the manuscript, the authors appropriately addressed all of my comments. I am satisfied with the changes made by the authors and support publication after incorporation of the minor comments listed below.

We thank the reviewer for this positive assessment and the additional comments that we have now addressed,

Comments

1. Line 282-285: The text states, “Differential gene expression analyses further confirmed that genes involved in the regulation of the actomyosin cytoskeleton including known YAP target genes (AMOTL1, TAGLN, CCN2), ... were upregulated at 30 min (Extended Data Fig. S3e).” However, these YAP target genes are not indicated in the figure. Please label these genes in the figure.

We have corrected AMOTL1 to AMOTL2 and added these three genes into the figure

2. Line 286-288: “... genes involved in pluripotency such as SOX2, OCT4, and DUSP7 were largely unchanged upon 30 min of compression (Extended Data Fig. S3e, f).” However, Extended Data Fig. S3e shows DUSP7 as being upregulated. Please revise the text appropriately.

We have amended the text to read “genes involved in pluripotency such as SOX2, OCT4, and DUSP7 were largely unchanged or even mildly upregulated upon 30 min of compression (Extended Data Fig. S3e, f).”

3. Figure 3I. “Puripotency” should be “Pluripotency”.

We thank the reviewer for noticing this typo that we have now corrected